# GAUSSIAN CERTIFIED UNLEARNING IN HIGH DIMENSIONS: A HYPOTHESIS TESTING APPROACH

**Aaradhya Pandey**[1]**, Arnab Auddy**[2]**, Haolin Zou**[3]**, Arian Maleki**[3]**, Sanjeev Kulkarni**[1]
[1]Princeton University    [2]The Ohio State University    [3]Columbia University
ap9898@princeton.edu, auddy.1@osu.edu, hz2574@columbia.edu
mm4338@columbia.edu, kulkarni@princeton.edu

## ABSTRACT

Machine unlearning seeks to efficiently remove the influence of selected data while preserving generalization. Significant progress has been made in low dimensions, where the dimension of the parameter $p$ is much smaller than the sample size $n$, but high dimensions, including proportional regimes $p \sim n$, pose serious theoretical challenges as standard optimization assumptions of $\Omega(1)$ strong convexity and $O(1)$ smoothness of the per-example loss $f$ rarely hold simultaneously in proportional regimes $p \sim n$. In this work, we introduce $\varepsilon$-Gaussian certifiability, a canonical and robust notion well-suited to high-dimensional regimes, that optimally captures a broad class of noise adding mechanisms. Then we theoretically analyze the performance of a widely used unlearning algorithm based on one step of the Newton method in the high-dimensional setting described above. Our analysis shows that a single Newton step, followed by a well-calibrated Gaussian noise, is sufficient to achieve both privacy and accuracy in this setting. This result stands in sharp contrast to the only prior work that analyzes machine unlearning in high dimensions Zou et al. (2025), which relaxes some of the standard optimization assumptions for high-dimensional applicability, but operates under the notion of $\varepsilon$-certifiability. That work concludes that at least two steps are required to ensure both privacy and accuracy. Our result leads us to conclude that the discrepancy in the number of steps arises because of the sub optimality of the notion of $\varepsilon$-certifiability and its incompatibility with noise adding mechanisms, which $\varepsilon$-Gaussian certifiability is able to overcome optimally.

## 1 INTRODUCTION AND THEORETICAL CONTRIBUTIONS

Modern ML models (from healthcare tools to systems like ChatGPT) are trained on data that often includes sensitive personal information, creating risks of memorization and leakage. Laws such as Union (2016); of California Department of Justice (2018); of Canada (2022) enforce a "right to be forgotten," requiring removal of both the data and its statistical influence on trained models. Because full retraining is costly, machine unlearning develops efficient methods to remove data traces while preserving model generalization. Recent years have seen rapid progress, fueled by substantial empirical results and rigorous theory Cao & Yang (2015), Bourtoule et al. (2021), Chen et al. (2021), Gupta et al. (2021), Ullah et al. (2021), Nguyen et al. (2022), Chundawat et al. (2023), Warnecke et al. (2023), Tarun et al. (2024), Wang et al. (2024), Liu et al. (2025).

In addition to substantial progress in the practical development of machine unlearning algorithms, their theoretical understanding has also advanced significantly in recent years. For example, Guo et al. (2020) and Sekhari et al. (2021) studied *Newton-based* unlearning algorithms and concluded that a single Newton update, followed by appropriately calibrated noise, is sufficient to ensure privacy for users requesting data removal while preserving the accuracy of the resulting estimates. In addition, Neel et al. (2021), Allouah et al. (2025b) examined the performance of approximate unlearning algorithms based on *gradient descent* and *stochastic gradient descent*. Moreover, there have been recent works Ginart et al. (2019), Qiao et al. (2025), Hu et al. (2025), Shen et al. (2025) developing algorithms for online unlearning to handle continuous unlearning requests.

To fix ideas, consider a supervised learning problem of predicting a response $y$ based on $p$-dimensional feature $x$ based on $n$ training samples $\mathbf{z}_i = (x_i, y_i) \in \mathbb{R}^{p+1}$ for $i \in [n]$. A popular parametric

version of the above problem is to consider the following optimization problem:

$$\hat{\boldsymbol{\beta}} = \arg\min_{\boldsymbol{\beta}} \frac{1}{n} \sum_{i=1}^{n} f(\boldsymbol{\beta}, \mathbf{z}_i) = \arg\min_{\boldsymbol{\beta}} \sum_{i=1}^{n} f(\boldsymbol{\beta}, \mathbf{z}_i),$$

where $f(\boldsymbol{\beta}, z_i)$ is the per-example cost with $n$ datapoints.

An important caveat in these theoretical papers is they *implicitly assume* the number of model parameters $p$ is significantly smaller than the number of observations $n$, by putting *standard optimization assumptions on the loss, which fails to hold in high-dimensional proportional regimes. We prove this point later in this section*. So, these results are not useful for many AI applications, wherein the number of model parameters $p$ is very large and comparable to the sample size $n$. We refer to this as the high-dimensional setting, in contrast to the low-dimensional setting considered in Guo et al. (2020), Sekhari et al. (2021), Neel et al. (2021), and Allouah et al. (2025b).

In the remaining of this section, we first show why the standard optimization assumptions of the per-example loss $f$ in Allouah et al. (2025b); Sekhari et al. (2021) break down in high dimensions. Then we compare our theoretical conclusions with Zou et al. (2025), as it is most closely aligned with ours in high dimensions. We then describe our proposed method and its performance.

**Failure of the assumptions in Allouah et al. (2025b), Sekhari et al. (2021).** In their theory Allouah et al. (2025b) assumes[1] that $f$ satisfies the standard optimization assumptions. More precisely, they assume $\boldsymbol{\beta} \to f(\boldsymbol{\beta}, \mathbf{z}_i)$ is simultaneously $\mu$-strongly convex and $L$-smooth, i.e.,

$$\mu I_p \preceq \nabla^2 f(\boldsymbol{\beta}, \mathbf{z}_i) \preceq L I_p \text{ for all } \mathbf{z}_i$$

for $L = O(1)$ and $\mu = \Omega(1)$. These two cannot hold simultaneously in high-dimensional proportional regime $p \sim n$. In fact these are even violated in simple examples such as linear regression. To prove this, we consider one of the simplest estimators based on ridge regularized least square loss: $\sum_{i=1}^{n} f(\boldsymbol{\beta}, \mathbf{z}_i) = \lambda\|\boldsymbol{\beta}\|^2 + \sum_{i=1}^{n}(y_i - \boldsymbol{x}_i^\top \boldsymbol{\beta})^2$. A computation of the Hessian gives $\nabla^2 \left(\sum_{i=1}^{n} f(\boldsymbol{\beta}, \mathbf{z}_i)\right) = 2X^\top X + 2\lambda I_p$. In high dimensional proportional regime $p \sim n$ that is studied in this paper, the condition $L = O(1)$ smoothness constant of the per-example loss $f$ inevitably requires the scaling $\|\boldsymbol{x}_i\|_2 \sim 1$ (with high probability) for all $i \in [n]$, which in turn requires $\lambda = O(1)$. This, however, completely destroys the $\mu = \Omega(1)$ strong convexity of the per-example loss $f$, as $\lambda_{min}\left(\nabla^2 f(\boldsymbol{\beta}, \mathbf{z}_i)\right) = 2\frac{\lambda}{n}$. More details can be found in Section D of the appendix.

**Comparing with conclusions of Zou et al. (2025).** In high dimensions, the privacy-accuracy interplay is subtler than in low dimensions. Zou et al. (2025) is the only theoretical work we know of that has studied machine unlearning when $p \sim n$ by *relaxing some of the standard optimization assumptions on the loss*, relative to previous works with stringent assumptions Allouah et al. (2025b),Sekhari et al. (2021). Our rigorous result that **one noisy Gaussian-Newton step suffices** (14) to unlearn an RERM (1) in high dimensions under $(\phi_n, \varepsilon)$-GPAR 3.2 stands in contrast with Zou et al. (2025). Their theoretical analysis relies on the $(\phi, \varepsilon)$-PAR certifiability (section 3.1) and shows that even for removing a single data point, **at least two Newton steps are required** to ensure the accuracy. The sharp contrast between the analytical results highlights the tightness of our GPAR framework 3.2 in high dimensions. The *suboptimality of $\varepsilon$-certifiability stems from its incompatibility with natural noise addition strategies*, thereby necessitating the injection of disproportionately large noise to meet the certifiability criteria, an effect that is particularly pronounced in high-dimensional regimes.

**Theoretical contributions of this paper.** We *introduce* the canonical notion of $(\phi, \varepsilon)$-Gaussian certifiability (3.2) that is particularly well suited for high dimensions, and *analyze* the performance of a widely used machine unlearning algorithm in the underexplored proportional regime $p \sim n$. This requires us to relax the stringent standard optimization assumptions on the per-example loss $f$ in all previous works, including Guo et al. (2020), Sekhari et al. (2021), Neel et al. (2021), Allouah et al. (2025b). In this setting, our work makes the following *novel theoretical contributions*:

- **Certifiability.** Inspired by the hypothesis testing interpretation of differential privacy Wasserman & Zhou (2010), Kairouz et al. (2015), Liu et al. (2019), Balle et al. (2020), Bu et al. (2020), Dong et al. (2021), Awan & Dong (2022), Awan & Vadhan (2023), Dong et al. (2022), we introduce a *practical*, *optimally achievable*, *statistically interpretable*, and *robust*

---

[1]Sekhari et al. (2021) assumes a variation of standard optimization assumptions on per-example loss $f$ whose conclusions blow up in high dimensions as well. This important case is treated in Section D of the appendix.

notion of certifiability, called $\varepsilon$-Gaussian certifiability. This notion is particularly *well-suited for developing theory* in high-dimensional situations. In fact, as proved in Dong et al. (2021), a broad class of noise-adding mechanisms converge in behavior to $\varepsilon$-Gaussian certifiability as the dimensionality increases. This makes $\varepsilon$-Gaussian certifiability the *canonical* framework of certifiability in high dimensions.

- **Performance.** Similar to Guo et al. (2020); Sekhari et al. (2021), and Zou et al. (2025), we then theoretically investigate a class of unlearning algorithms that perform a few steps of Newton's method, rather than fully re-optimizing the empirical risk. In contrast to Guo et al. (2020), Sekhari et al. (2021), Allouah et al. (2025b), we explore the theoretical behavior of these methods in *high-dimensional* proportional setting of $p \sim n$, by *relaxing the standard optimization assumptions of* $\Omega(1)$ *strong convexity and* $O(1)$ *smoothness of the chosen per-example loss*. We prove that a *single* Newton update with carefully calibrated independent Gaussian noise, simultaneously attains $\varepsilon$-Gaussian certifiability and incurs only a vanishing degradation in generalization error. Note that while Sekhari et al. (2021); Guo et al. (2020) also concluded that a *single Newton step is sufficient* for approximate removal of a single datapoint, but due to *their restrictive assumptions on the loss*, their results fail completely in high-dimensions, even in the simplest setting of ridge regularized least squares.

- **Removing multiple data points.** Our theory encompasses the simultaneous removal of data from $m$ users and show that, even when the number of removal requests $m$ grows with the sample size, accurate unlearning is still possible in high-dimensional settings provided that $m = o(n^{1/4}/\mathrm{polylog}(n))$.

Table 1: Summary of prior certified unlearning notions and per-example loss assumptions, compared to our proposed $(\varphi, \varepsilon)$-Gaussian certifiability framework (Def. 2). Our notion is optimally achievable with practical Gaussian noise mechanisms in the proportional high-dimensional regime, and it relies on relaxed convex per-example loss assumptions that remain valid in this setting (Sec. 4.2).

| Related works | Certifiability notion (optimality) | High dimensional applicability |
|---|---|---|
| Guo et al. (2020) | $(\varepsilon, \delta)$-unlearning (suboptimal) | Assumptions fail in high dimensions |
| Sekhari et al. (2021) | $(\varepsilon, \delta)$-unlearning (suboptimal) | Assumptions fail in high dimensions |
| Allouah et al. (2025b) | $(q, \varepsilon)$-Rényi (suboptimal) | Assumptions fail in high dimensions |
| Zou et al. (2025) | $(\phi, \varepsilon)$-unlearning (suboptimal) | Extends to high dimensions |
| Proposed in this paper | $(\phi, \varepsilon)$-Gaussian (optimal) | Extends to high dimensions |

## 2 THE MACHINE UNLEARNING PROBLEM: PRIVACY AND ACCURACY

Let $\mathcal{D}_n$ denote the dataset, $A(\mathcal{D}_n)$ denote the machine learning model trained on $\mathcal{D}_n$, and $\mathcal{M} \subset [n]$ the indices of points to be removed, with corresponding data $\mathcal{D}_{\mathcal{M}}$. The *problem of machine unlearning* requires a procedure $\bar{A}$ to produce an update $\bar{A}_{\backslash \mathcal{M}} := \bar{A}(A(\mathcal{D}_n), \mathcal{D}_{\mathcal{M}}, T(\mathcal{D}_n))$ that closely approximates the *ideal* model $A(\mathcal{D}_n \setminus \mathcal{D}_{\mathcal{M}})$ retrained from scratch. We assume that $\bar{A}$ has access to the model output $A(\mathcal{D}_n)$, the dataset of removal requests $\mathcal{D}_{\mathcal{M}}$ and some auxiliary gradient or Hessian information $T(\mathcal{D}_n)$ as in Sekhari et al. (2021) and Zou et al. (2025).

We focus on *approximate unlearning* (e.g. Guo et al. (2020), Sekhari et al. (2021), Neel et al. (2021), Allouah et al. (2025b)) where, $\bar{A}$ achieves *statistical proximity* to the retrained model as opposed to *exact unlearning* Cao & Yang (2015), Bourtoule et al. (2021),Basu Roy Chowdhury et al. (2025), Kuo et al. (2025), which requires literal equality $\bar{A}_{\backslash \mathcal{M}} = A(\mathcal{D}_n \setminus \mathcal{D}_{\mathcal{M}})$ (almost surely or in distribution). Exact recomputation is often computationally infeasible, and since our *focus is on extending the theory of the proposed unlearning framework to a large class of loss functions in high-dimensional problems of interest*, approximate removal methods offer a practical and scalable alternative. Note that the machine unlearning mechanism $\bar{A}_{\backslash \mathcal{M}}$ is expected to satisfy the following two properties:

**Protect user privacy.** The output of the unlearning procedure $\bar{A}_{\backslash \mathcal{M}}$ must leak *no information* about the removed subset $\mathcal{D}_{\mathcal{M}}$, even under arbitrary post-processing of the unlearned output.

**Preserve model accuracy.** The unlearned model $\bar{A}_{\backslash \mathcal{M}}$ must incur only a *vanishing change* in generalization capabilities compared to the ideal retrained from scratch model $A(\mathcal{D}_n \setminus \mathcal{D}_{\mathcal{M}})$.

### 2.1 GENERALIZED LINEAR MODEL (OF DATA) AND EMPIRICAL RISK MINIMIZATION

We now describe our high-dimensional setup as a step towards formalizing privacy and accuracy.

**GLM Data.** We assume our data $\mathcal{D}_n := \{(\boldsymbol{x}_i, y_i)\}_{i=1}^n$ with $\boldsymbol{x}_i \in \mathbb{R}^p$ denoting the *features* and $y_i \in \mathbb{R}$ denoting the *response*. Further, $(\boldsymbol{x}_i, y_i)$ are i.i.d. samples from $dP_{\boldsymbol{\beta}^*}(\boldsymbol{x}, y) = dp(\boldsymbol{x})dq(y|\boldsymbol{x}^\top \boldsymbol{\beta}^*)$ with *density* $p : \mathbb{R}^p \to \mathbb{R}_{\geq 0}$ and *kernel* $q(\cdot|\boldsymbol{x}^\top \boldsymbol{\beta}^*) : \mathcal{Y} \to \mathbb{R}_{\geq 0}$ with *unknown* $\boldsymbol{\beta}^* \in \mathcal{B}_p := \mathbb{R}^p$.

**High-dimensional focus.** In many AI applications, the number of model parameters, $p$, is often comparable to the sample size, $n$. So, we aim to study models under the *proportional high dimensional setting* in which both $n, p$ are large while $n/p = \gamma$. This sets the interpretation of all asymptotic expressions as the directional limit $n \to \infty, p = n/\gamma$. For example, a sequence of random variables $W_n = o_P(1)$ means $W_n$ converges to zero in probability under the asymptotic setting above.

**RERM training.** The goal of a learning procedure $A$ is to estimate the parameter $\boldsymbol{\beta}^*$ from training data. A widely used approach is *regularized empirical risk minimization* (R-ERM), defined as:

$$\hat{\boldsymbol{\beta}} = A(\mathcal{D}_n) := \underset{\boldsymbol{\beta} \in \mathbb{R}^p}{\arg\min} \quad L(\boldsymbol{\beta}) := \underset{\boldsymbol{\beta} \in \mathbb{R}^p}{\arg\min} \quad \lambda r(\boldsymbol{\beta}) + \sum_{i=1}^n \ell(y_i \mid \boldsymbol{x}_i^\top \boldsymbol{\beta}), \tag{1}$$

where $\ell$ is a (individual) loss function and $r$ is a regularization term with factor $\lambda$, which we assume is $O(1)$ throughout. A statistically interpretable choice of the individual loss $\ell(y|z)$ is the negative log likelihood $-\log q(y|z)$, and the choice of the regularizer $r$ is often based on some prior information on $\boldsymbol{\beta}^*$. Popular choices include ridge ($r = \|\cdot\|_2^2$), LASSO ($r = \|\cdot\|_1$), and their variants.

The objective of unlearning is to remove the influence of a subset $\mathcal{M}$ of observations, of size $|\mathcal{M}| = m$, that have requested removal from the training data. The *ideal unlearned estimator* would be the one retrained from scratch on the remaining data $\mathcal{D}_{\setminus \mathcal{M}} := \mathcal{D}_n \setminus \mathcal{D}_{\mathcal{M}}$:

$$\hat{\boldsymbol{\beta}}_{\setminus \mathcal{M}} := A(\mathcal{D}_{\setminus \mathcal{M}}) := \underset{\boldsymbol{\beta} \in \mathbb{R}^p}{\arg\min} \quad L_{\setminus \mathcal{M}}(\boldsymbol{\beta}) := \underset{\boldsymbol{\beta} \in \mathbb{R}^p}{\arg\min} \quad \lambda r(\boldsymbol{\beta}) + \sum_{i \notin \mathcal{M}} \ell(y_i|\boldsymbol{x}_i^\top \boldsymbol{\beta}). \tag{2}$$

To process deletions efficiently, without full retraining, and obscure residual information, we update $\hat{\boldsymbol{\beta}}$ via the randomized procedure below. [2]

$$\tilde{\boldsymbol{\beta}}_{\setminus \mathcal{M}} := \bar{A}(\hat{\boldsymbol{\beta}}, \mathcal{D}_{\mathcal{M}}, T(\mathcal{D}_n), \boldsymbol{b}), \text{ where } \boldsymbol{b} \text{ is a random noise.} \tag{3}$$

Given a trained model, $\hat{\boldsymbol{\beta}}$, note that $\boldsymbol{b}$ is the only source of randomization during unlearning. In addition to the perturbation $\boldsymbol{b}$, the unlearning procedure $\bar{A}$ has access to the set $\mathcal{D}_{\mathcal{M}}$ corresponding to the data to be removed, and limited access to the outputs of the learning algorithm $A$ on $\mathcal{D}_n$. For example, (i) the original estimator $\hat{\boldsymbol{\beta}}$, and (ii) some auxiliary information $T(\mathcal{D}_n)$ such as *gradient* or *Hessian* of the objective function on $\mathcal{D}_n$, at $\hat{\boldsymbol{\beta}}$.

## 3 FORMAL DEFINITIONS OF CERTIFIABILITY AND ACCURACY

As discussed in Section 2, machine unlearning algorithms are expected to satisfy two key criteria: (1) *protect user privacy* and (2) *preserve model accuracy*. The purpose of this section is to formalize the notions corresponding to these objectives by introducing: (1) $(\phi, \varepsilon)$-Gaussian certifiability as a canonical measure of privacy, and (2) Generalized error divergence as a measure of accuracy. We will formally establish later that both notions are particularly well suited to high-dimensions.

### 3.1 GENERAL CERTIFIABILITY THROUGH THE LENS OF HYPOTHESIS TESTING

Let us take the perspective of an adversary who, given an output $\check{\boldsymbol{\beta}}$ from an unlearning algorithm $\bar{A}$, performs the following hypothesis testing problem:

$$H_0 : \check{\boldsymbol{\beta}} \sim \mathcal{P}_{\text{re}} \quad \text{vs.} \quad H_1 : \check{\boldsymbol{\beta}} \sim \mathcal{P}_{\text{un}} \tag{4}$$

where $\mathcal{P}_{\text{re}}$ and $\mathcal{P}_{\text{un}}$ are the conditional distributions of $\bar{A}(\hat{\boldsymbol{\beta}}_{\setminus \mathcal{M}}, \emptyset, T(\mathcal{D}_{\setminus \mathcal{M}}), \boldsymbol{b})$ given the data $\mathcal{D}_{\setminus \mathcal{M}}$ and $\bar{A}(\hat{\boldsymbol{\beta}}, \mathcal{D}_{\mathcal{M}}, T(\mathcal{D}), \boldsymbol{b})$ given the data $\mathcal{D}$ respectively. Indeed, if the adversary has sufficient evidence to reject $H_0$, it would mean that $\check{\boldsymbol{\beta}}$ has retained some information about $\mathcal{D}_{\mathcal{M}}$, defeating the purpose of machine unlearning. Note that in all settings, the adversary may make mistakes in accepting or rejecting the null. Hence, to consistently evaluate the quality of the adversary's decision, we have to evaluate the probability of Type I error (i.e. the error of rejecting $H_0$ when the truth is $H_0$), and Type II error (i.e. the error of not rejecting $H_0$ when the truth is $H_1$).

---

[2]For more details on the claim that randomization is *necessary* to hide residual information and preserve privacy of individuals, please refer to the literature of differential privacy Dwork (2006).

**Trade-off functions:** The above leads us to the notion of trade-off curve defined below and studied extensively in the Statistics and information theory Lehmann et al. (1986), Poor (1994). To introduce this notion, let us simplify the notations of the above problem to the following. Suppose that a random vector $W$ is given, and we would like to perform the following test:

$$H_0 : W \sim P \quad vs. \quad H_1 : W \sim Q.$$

**Definition 1.** *(Trade-off function) Given two probability distributions $P, Q$ on a measurable space $(\mathcal{W}, \mathcal{F}_\mathcal{W})$, we define the* trade-off function *as the map $T(P, Q) : [0, 1] \to [0, 1]$ as*

$$T(P, Q)(\alpha) := \inf_\phi \left\{ \beta_\varphi := \mathbb{E}_Q[1 - \varphi] \; \Big| \; \alpha_\varphi := \mathbb{E}_P[\varphi] \le \alpha, \; \varphi : \mathcal{W} \to [0, 1] \text{ measurable} \right\}. \quad (5)$$

*In words, for any given type I error $\alpha$, the trade-off function returns the smallest possible value of type II error $\beta_\varphi$ over all possible test functions $\varphi \colon \mathcal{W} \to [0, 1]$.*

Using the notion of trade-off functions, and inspired by the notion of $f$-differential privacy introduced in Dong et al. (2022), we aim to define a notion of certifiability that generalizes and unifies the existing techniques. Intuitively speaking, once we decide on a trade-off function $f : [0, 1] \to [0, 1]$, we can certify $\bar{A}$ if

$$T(\mathcal{P}_{\mathrm{re}}, \mathcal{P}_{\mathrm{un}})(\alpha) \ge f(\alpha) \quad \text{for all } \alpha \in [0, 1] \tag{6}$$

To make our certifiability criteria less restrictive, we require the above definition to be satisfied with high probability over datasets $\mathcal{D}$. This leads us to the following definition of general $f - $ *certifiability*.

**Definition 2** ($f$-certifiability). *Given $\phi > 0, m \in [n]$ and $\mathcal{P}_{\mathrm{un}}, \mathcal{P}_{\mathrm{re}}$ as define in (4), and a trade-off curve $f : [0, 1] \to [0, 1]$, we say unlearning algorithm $\bar{A}$ satisfies $\phi$-certifiable with respect to $f$ if*

$$\mathbb{P}\left[ \inf_{|\mathcal{M}| \le m} \min(T(\mathcal{P}_{\mathrm{re}}, \mathcal{P}_{\mathrm{un}})(\alpha), T(\mathcal{P}_{\mathrm{un}}, \mathcal{P}_{\mathrm{re}})(\alpha)) \ge f(\alpha) \quad \text{for all } \alpha \in [0, 1] \right] \ge 1 - \phi \tag{7}$$

*where the probability $\mathbb{P}$ is solely over the randomness of the data $\mathcal{D}$.*

Different choices of $f$ yield different notions of certifiability. Observe that by choosing different trade-off functions, we can cover a wide range of notions of privacy in the literature Guo et al. (2020), Sekhari et al. (2021), Zou et al. (2025). For instance, by setting $f(\alpha)$ in the following way:

$$f_{\varepsilon, \delta}(\alpha) := \max\{0, 1 - \delta - e^\varepsilon \alpha, e^{-\varepsilon}(1 - \delta - \alpha)\} \text{ for parameters } \varepsilon \ge 0, 0 \le \delta \le 1 \tag{8}$$

we will obtain a notion similar to $(\varepsilon\text{-}\delta)$-certifiability condition discussed in Sekhari et al. (2021).

## 3.2 $(\phi, \varepsilon)$-GAUSSIAN CERTFIABILITY- A CANONICAL CHOICE IN HIGH DIMENSIONS

We study machine unlearning in high dimensions, where choosing an appropriate $f$ is the key to formalizing certifiability. In this regime, a natural (indeed canonical) choice of $f$ emerges from two facts. 1) In practice, privacy is achieved by adding noise, and 2) an intrinsically high-dimensional phenomenon implies that, *with high probability (up to a tolerance $\phi$), most 1d projections of an isotropic high-dimensional (log-concave) distribution are approximately Gaussian*[3] Diaconis & Freedman (1984), Klartag (2007). A precise instantiation of this idea appears in the differential privacy literature Dong et al. (2021), which motivates us to consider the following choice of the trade-off curve $f_{G,\varepsilon}$ in Definition 2 in comparison to previously proposed notions such as $f_{\varepsilon,\delta}$.

$$f_{G,\varepsilon}(\alpha) := T(N(0, 1), N(\varepsilon, 1))(\alpha) = \Phi(\Phi^{-1}(1 - \alpha) - \varepsilon) \tag{9}$$

where $\Phi(\cdot)$ is the cdf of a standard Gaussian variable $\Phi(t) = \mathbb{P}[N(0, 1) \le t]$ for all $t \in \mathbb{R}$.

$f_{G,\varepsilon}(\alpha)$ possesses several attractive properties. The following lemma lists one such property of Gaussian trade-off functions known as *dimension-freeness*. This will be used later in this paper.

**Lemma 1** (Dimension freeness). *For any $\boldsymbol{\mu}_1, \boldsymbol{\mu}_2 \in \mathbb{R}^p$ and $\sigma > 0$ let $\varepsilon := \frac{1}{\sigma}\|\boldsymbol{\mu}_1 - \boldsymbol{\mu}_2\|_2$. Then*

$$T(\boldsymbol{\mu}_1 + \sigma N(0, \mathbb{I}_p), \boldsymbol{\mu}_2 + \sigma N(0, \mathbb{I}_p)) \equiv T(N(0, 1), N(\varepsilon, 1)). \tag{10}$$

---

[3]Think of all the 1d projections of the uniform distribution on a high dimensional sphere as an example.

We call the criterion of Definition 2, with $f_{G,\varepsilon}(\cdot)$ replacing $f(\cdot)$ as $(\phi, \varepsilon)$-GPAR, to abbreviate $(\phi, \varepsilon)$-Gaussian Probabilistically certified Approximate data Removal or $(\phi, \varepsilon)$-Gaussian certifiability. We enlist some more properties of $(\phi, \varepsilon)$-GPAR that make it emerging in high-dimensional settings:

**Universality of Gaussian noise.** Dong et al. (2021) shows that a broad class of isotropic log-concave noise-adding mechanisms (to achieve privacy) converge in behavior to $(\phi, \varepsilon)$-Gaussian certifiability as $p \to \infty$. This makes Gaussian mechanism the natural noise-adding mechanism in high dimensions.

**Gaussian certifiability as the tightest notion.** Now, once we use the canonical Gaussian noise for randomizing the estimates, a consequence of *Blackwell ordering* says that the Gaussian trade-off curve (9) is the tightest way to capture the Gaussian mechanism Dong et al. (2022).

**Suboptimality of all other notions of certifiability.** All the previously proposed notions of unlearning, including $\varepsilon$-$\delta$ variations of Guo et al. (2020), Sekhari et al. (2021), Zou et al. (2024), Rényi certifiability of Allouah et al. (2025b), are not tightly achievable under the Gaussian mechanism except for Gaussian certifiability Dong et al. (2022). This establishes the natural emergence of the $\varepsilon$-GPAR framework with tolerance $\phi$ as superior over any other framework in high dimensions.

**Appearance in practice, in related experiments.** In the context of machine unlearning, Gaussian trade-off curves (9) (the core of Gaussian certifiability) have been experimentally shown to match empirical trade-off curves remarkably well for the removal of poisoned data Pawelczyk et al. (2025).

In Section 4, we will use the $(\phi, \varepsilon)$-GPAR framework to theoretically evaluate the user privacy protection capabilities of machine unlearning algorithms that are based on the Newton method.

### 3.3 GENERALIZATION ERROR DIVERGENCE: A MEASURE OF ACCURACY

In the previous subsection, we introduced a canonical notion of certifiability for unlearning. We now turn to the second criterion, which aims to evaluate model accuracy. Without such a criterion, an unlearning algorithm could output pure noise, achieving perfect user privacy at the cost of severely degraded model performance. To formally evaluate the generalization capabilities of our unlearning procedure $\bar{A}$, we consider the metric Generalization error divergence (GED).

**Definition 3.** *Given a dataset $\mathcal{D}_n$ of $n$ i.i.d. samples $(\boldsymbol{x}_i, y_i)$ as introduced in Section 2.1, let $(\boldsymbol{x}_0, y_0)$ be a new sample i.i.d. with the observations in $\mathcal{D}_n$. Let $\ell(y|\boldsymbol{x}^\top \boldsymbol{\beta})$ be a measure of error between $y$ and $\boldsymbol{x}^\top \boldsymbol{\beta}$. Then the Generalization Error Divergence (GED) of the learning-unlearning pair $(A, \bar{A})$ on the dataset $\mathcal{D}_n$ with a subset $\mathcal{M} \subset [n]$ of data removal requests is defined as:*

$$\text{GED}_\ell(A, \bar{A}; \mathcal{M}, \mathcal{D}_n) := \mathbb{E}\left(\left[\left|\ell(y_0|\boldsymbol{x}_0^\top A(\mathcal{D}_{\backslash \mathcal{M}})) - \ell(y_0|\boldsymbol{x}_0^\top \bar{A}(A(\mathcal{D}_n), \mathcal{D}_\mathcal{M}, T(\mathcal{D}_n), \boldsymbol{b}))\right|\right] |\mathcal{D}_n\right),$$
(11)

*where we condition on the randomness of the data set $\mathcal{D}_n$ and average over the randomness of the unlearning algorithm $\bar{A}$, as well as that of the test data point $(\boldsymbol{x}_0, y_0)$.*

The above metric compares the generalization error of the unlearning algorithm $\bar{A}$ given $\hat{\boldsymbol{\beta}}$ and data removal requests $\mathcal{M}$ with that of the model $A$ trained from scratch on $\mathcal{D}_{\backslash \mathcal{M}}$, on a fresh sample $(\boldsymbol{x}_0, y_0)$. We note that previously Sekhari et al. (2021) used the notion of excess risk of $\bar{A}$ against that of the true minimizer of population risk as a measure of accuracy of machine unlearning algorithms. But, as we will show in Section D of appendix that this notion does not behave well under the proportional high-dimensional setting. We refer to Zou et al. (2025) for more.

## 4 NEWTON-BASED MACHINE UNLEARNING: THEORETICAL GUARANTEES

Having settled down on both the evaluation criteria, we study the performance of unlearning procedure $\bar{A}$ based on one Newton step and perturbing the result with Gaussian noise to achieve $(\phi, \varepsilon)$-GPAR.

### 4.1 NEWTON-BASED UNLEARNING MECHANISM

Newton method is an iterative method for solving optimization problems based on a second-order Taylor approximation of the given optimizing function Boyd & Vandenberghe (2004).

**Definition 4.** *(Newton-Raphson method) Suppose that $h : \mathbb{R}^p \to \mathbb{R}^p$ is a differentiable function with an invertible Jacobian matrix $G$ anywhere on $\mathbb{R}^p$ and that $h$ has a root $h(\boldsymbol{\beta}^*) = 0$. Starting from an initial point $\boldsymbol{\beta}^{(0)} \in \mathbb{R}^p$, the Newton method is the following iterative procedure: for step $t \geq 1$,*

$$\boldsymbol{\beta}^{(t)} := \boldsymbol{\beta}^{(t-1)} - \boldsymbol{G}^{-1}(\boldsymbol{\beta}^{(t-1)})h(\boldsymbol{\beta}^{(t-1)}).$$
(12)

Using the Newton method Sekhari et al. (2021) suggested the following unlearning algorithm:
**Approximation step.** starting from $\hat{\boldsymbol{\beta}}$, we run one Newton step with $h = \nabla L_{\backslash \mathcal{M}}$ to obtain:

$$\hat{\boldsymbol{\beta}}^{(1)}_{\backslash \mathcal{M}} = \hat{\boldsymbol{\beta}} - \boldsymbol{G}(L_{\backslash \mathcal{M}})^{-1}(\hat{\boldsymbol{\beta}}) \nabla L_{\backslash \mathcal{M}}(\hat{\boldsymbol{\beta}}), \text{ where } \boldsymbol{G}(L_{\backslash \mathcal{M}}) \text{ is the Hessian of } L_{\backslash \mathcal{M}} \text{ in (2).} \quad (13)$$

**Randomization step.** add a Gaussian noise $\boldsymbol{b} \sim N(\boldsymbol{0}, \sigma^2 \mathbb{I}_p)$ to $\hat{\boldsymbol{\beta}}^{(1)}_{\backslash \mathcal{M}}$ to have the unlearned output

$$\text{Noisy one step Newton output: } \bar{A}(A(\mathcal{D}_n), \mathcal{D}_{\mathcal{M}}, T(\mathcal{D}_n), \boldsymbol{b})) = \tilde{\boldsymbol{\beta}}_{\backslash \mathcal{M}} := \hat{\boldsymbol{\beta}}^{(1)}_{\backslash \mathcal{M}} + \boldsymbol{b}. \quad (14)$$

First, the choice of $\sigma$ will be discussed in our theoretical results. Next, note that since $\hat{\boldsymbol{\beta}}^{(1)}_{\backslash \mathcal{M}}$ differs from $\hat{\boldsymbol{\beta}}_{\backslash \mathcal{M}}$, the difference between the two vectors may reveal information about the data to be removed, $\mathcal{D}_{\mathcal{M}}$. Hence, a standard practice is to obscure it by adding random noise $\boldsymbol{b}$. Finally, the aim in the remaining part of this section is to answer the following two questions:

**Privacy.** How to choose $\sigma$ to ensure that $\tilde{\boldsymbol{\beta}}_{\backslash \mathcal{M}}$ satisfies $(\phi, \varepsilon)$-GPAR when $n, p$ large with $n/p \to \gamma$?
**Accuracy.** Can we calculate the generalization error divergence of $\tilde{\boldsymbol{\beta}}_{\backslash \mathcal{M}}$ given the choice of $\sigma$ obtained above? Does the GED go to zero as $n, p \to \infty$ while $n/p \to \gamma$?

### 4.2 TECHNICAL ASSUMPTIONS ON THE LOSS FUNCTIONS AND THE DATA

Loss functions. We describe the assumptions on the unregularized individual loss $l$ and the regularizer $r$ (1). To ensure that the RERM introduced in (1) has a unique solution (from strong convexity arguments) and Newton unlearning makes sense theoretically, the following assumption are made:

(A1) **Separability of $r$.** $r : \mathbb{R}^p \to \mathbb{R}_+$ is separable. More precisely, $r(\boldsymbol{\beta}) = \sum_{k \in [p]} r_k(\beta_k)$.

(A2) **Smoothness of $(l, r)$.** $\ell : \mathbb{R} \times \mathbb{R} \to \mathbb{R}_+$ and $r : \mathbb{R}^p \to \mathbb{R}_+$ are both thrice differentiable.

(A3) **Convexity of $(l, r)$.** $\ell$ and $r$ are proper convex, and $r$ is $\nu$-strongly convex for $\nu = \Theta(1) > 0$.

(A4) **Polynomial growth.** $\exists$ constants $C, s > 0$ such that for all $y, z$ (denote $\dot{\ell}(y, z) := \partial_y \ell(y, z)$)

$$\max\{\ell(y, z), |\dot{\ell}(y, z)|, |\ddot{\ell}(y, z)|\} \leq C(1 + |y|^s + |z|^s) \text{ and } \nabla^2 r(\boldsymbol{\beta}) = \mathbf{diag}[\ddot{r}_k(\beta_k)]_{k \in [p]} \quad (15)$$

is $C_{rr}(n)$-Lipschitz (in Frobenius norm) in $\boldsymbol{\beta}$ for some $C_{rr}(n) = O(\text{polylog}(n))$.

Note these assumptions on $(\ell, r)$ are from Zou et al. (2025), and they weaken many of the standard optimization assumptions of $\Omega(1)$ strong convexity and $O(1)$ smoothness of per example loss $f$ in Allouah et al. (2025b) or variations thereof in Sekhari et al. (2021). Convexity of $\ell$ and $\nu$-Strong convexity of $r$ above actually means that per-example loss $\boldsymbol{\beta} \to f(\boldsymbol{\beta}, \mathbf{z}) = l(y, x^T \boldsymbol{\beta}) + \frac{\lambda}{n} r(\boldsymbol{\beta})$ is $\frac{\lambda \nu}{n}$ strongly convex, which is significant relaxation of $\Omega(1)$ strong convexity of $f$, capturing the concrete case of the ridge regularized $r(\boldsymbol{\beta}) = \|\boldsymbol{\beta}\|^2$ quadratic loss $l(y, z) = (y - z)^2$. Allouah et al. (2025b) and Sekhari et al. (2021) both fail to capture the simplest example of high dimensional statistics in their respective frameworks.

We require the following on the data that generates our features $\boldsymbol{x}_i$ and responses $y_i$.

(B1) **Subgaussian features.** $\boldsymbol{x}_i$ are mean zero sub-Gaussian vectors with covariance $\boldsymbol{\Sigma}$ denoted as $\boldsymbol{x}_i \overset{iid}{\sim} \mathcal{SG}(\boldsymbol{0}, \boldsymbol{\Sigma})$. We further assume that $\lambda_{\max}(\boldsymbol{\Sigma}) \leq \frac{C_X}{p}$, for some constant $C_X > 0$.

(B2) **Sub-polylogarithmic responses.** We assume that $\mathbb{P}(|y_i| > C_y(n)) \leq q_y(n)$ and $\mathbb{E}|y_i|^{2s} \leq C_{y,s}$ for some $C_y(n) = O(\text{polylog}(n))$, a constant $C_{y,s}$ and $q_n^{(y)} = o(n^{-1})$.

Assumption (B1) on mean-zero sub-Gaussian features is a mild and frequently made in many papers on high dimensional statistics. See for example Miolane & Montanari (2021); Rahnama Rad & Maleki (2020); Auddy et al. (2024); Zheng et al. (2017); Donoho & Montanari (2016); Bellec et al. (2025). Assumption (B2) is not a stringent assumption either, and most popular models such as logistic regression, Poisson regression, and linear regression satisfy this assumption. Using these assumptions, we can finally prove our main theoretical results.

### 4.3 THEORETICAL GUARANTEES: ENSURING USER PRIVACY AS WELL AS MODEL ACCURACY

One step Newton achieves $(\phi, \varepsilon)$-GPAR. Our first result shows that for a suitable noise variance, the randomized one-step Newton unlearned estimator defined in (14) satisfies $(\phi, \varepsilon)$-GPAR.

**Theorem 2.** *Suppose Assumptions (A1)-(A4) as well as (B1)-(B2) hold. Then there exist poly-logarithmic $C_1(n), C_2(n) = O(\text{polylog}(n))$ for which the randomized one-step Newton unlearning* (14) *when used with a perturbation vector $\boldsymbol{b} \sim N\left(\boldsymbol{0}, \frac{R^2}{\varepsilon^2}\mathbb{I}_p\right)$, achieves $(\phi_n, \varepsilon)$-GPAR with*

$$R = C_1(n)\sqrt{\frac{C_2(n)m^3}{2\lambda\nu n}}, \quad \phi_n = nq_n^{(y)} + 8n^{-3} + ne^{-p/2} + 2e^{-p} \to 0. \tag{16}$$

The proof of this theorem is presented in Sections G, H of the appendix. After the above theoretical guarantee ensuring user privacy, we now calculate the GED of this machine unlearning algorithm.

**Theorem 3.** *Suppose Assumptions (A1)-(A4) as well as (B1)-(B2) hold. Consider the unlearning estimator defined in* (14) *with the noise variance set according to Theorem 2. Then, with probability at least $1 - (n+1)q_n^{(y)} - 14n^{-3} - ne^{-p/2} - 2e^{-p} - e^{-(1-\log(2))p}$,*

$$\text{GED}(\tilde{\boldsymbol{\beta}}_{\backslash\mathcal{M}}, \hat{\boldsymbol{\beta}}_{\backslash\mathcal{M}}) \leq C_1(n)\sqrt{C_2(n)}\left(\frac{1}{\varepsilon} + \frac{1}{\sqrt{p}}\right)\sqrt{\frac{m^3(m+2)}{\lambda\nu n}} \cdot \text{polylog}(n).$$

**The final technical conclusion of the paper.** Combining the above two theorems we reach the conclusion: If we set the variance of the Gaussian noise as suggested by Theorem 2, the machine unlearning algorithm $\tilde{\boldsymbol{\beta}}_{\backslash\mathcal{M}}$ that is based on (14) offers $(\phi_n, \varepsilon)$-GPAR as well as vanishing GED

$$\text{GED}(\tilde{\boldsymbol{\beta}}_{\backslash\mathcal{M}}, \hat{\boldsymbol{\beta}}_{\backslash\mathcal{M}}) = O_p\left(\frac{m^2\text{polylog}(n)}{\sqrt{n}}\right) \quad \text{as soon as} \quad \frac{m^2\text{polylog}(n)}{\sqrt{n}} \to 0 \tag{17}$$

In particular, if $m = o(n^{\frac{1}{4}-\alpha})$ for arbitrary $\alpha > 0$, we have $\text{GED}(\tilde{\boldsymbol{\beta}}_{\backslash\mathcal{M}}, \hat{\boldsymbol{\beta}}_{\backslash\mathcal{M}}) = o_p(1)$. Note that both theorems are valid in high-dimensional settings where $n, p \to \infty$, while $n/p \to \gamma$. Moreover, see appendix F.3 for a discussion on calibrating $\sigma$ without the constants $C_1(n), C_2(n)$, as well as a procedure in practice to determine what they are given a dataset $\mathcal{D}$ and the losses $(\ell, r)$.

## 5 EXPERIMENTS VALIDATING OUR THEORETICAL CONTRIBUTIONS

Given our theoretical focus, and the abundance of empirical studies on Newton-based unlearning Guo et al. (2020), Bui et al. (2024), we present simulations solely to validate our results and to compare with the results of Zou et al. (2025). The incompatibility of $\varepsilon$ certifiability with noise adding mechanisms requires disproportionately large amount of isotropic Laplace noise in Zou et al. (2025), whereas $\varepsilon$-GPAR criterion optimally captures the Gaussian mechanism. Our simulations show that it leads to substantial improvement of the unlearned estimator across different settings.

**Logistic regression model.** For simplicity, we use ridge penalized $r(\boldsymbol{\beta}) = \|\boldsymbol{\beta}\|^2$ logistic loss $\ell(y|\boldsymbol{x}^\top\boldsymbol{\beta})$, with random features $\boldsymbol{x}_i \sim N(\boldsymbol{0}, \frac{1}{n}\mathbb{I}_p)$ and true parameter $\boldsymbol{\beta}_* \sim N(\boldsymbol{0}, \mathbb{I}_p)$. To depict high-dimensionality, we take $n = p$, penalty parameter $\lambda = 0.5$, and show behavior of the error metrics by varying the values of $n, p, m$, and certifiability parameter $\varepsilon$. 1) When $m = 1$, we look at all possible removals of size one justifying robustness of our certifiability. 2) When $m > 1$, instead of all possible removals, we study unlearning across 1000 randomly chosen subsets of size $m$. More details and codes are in the appendix and at `https://anonymous.4open.science/r/unlearning-E14D`.

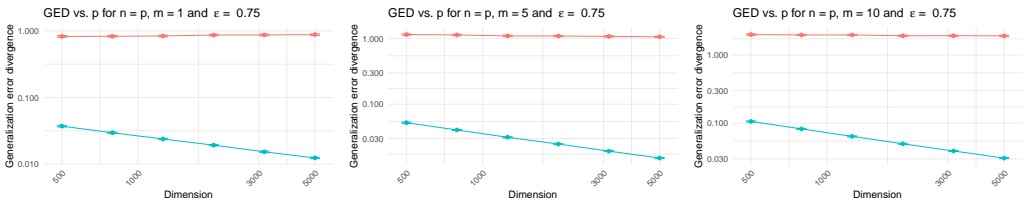

Figure 1: Comparison of unlearned estimators on new test data: mean GED (with 3 SD error bars) across the dimension $p$ (both in $\log$ scale) for Laplace (in red) vs. Gaussian (in cyan). We set $\lambda = 0.5$.

In the first figure we fix $\varepsilon = 0.75$, and examine via GED (11) the generalization performance of unlearning on 100 previously unseen test data. We vary $n = p$ across 6 values from 500 to 5000

equally spaced on a logarithmic scale. Figure 1 plots the average GED values (across 200 replications) against dimension $p$, transforming both X and Y axes to be on a logarithmic scale. The figures are repeated for unlearning sizes $m = 1$ (left), $m = 5$ (middle), and $m = 10$ (right). Observe for $m = 1, 5, 10$, as $p \uparrow$, GED for Laplace perturbed estimator of Zou et al. (2025) is unchanged, whereas our Gaussian perturbed estimator decreases steadily. A linear regression of $\log(\text{mean}(\text{GED}))$ vs. $\log(p)$ confirms this. The lines for Laplace unlearning had slopes of 0.03, -0.03, and -0.01 for $m = 1, 5, 10$ respectively, implying one step Laplace-Newton estimator has a non-vanishing GED as $p \uparrow$. This shows the sub-optimality of $\varepsilon$-certifiability by *requiring at least two Newton steps* to reach vanishing GED. In contrast, the Gaussian unlearning lines had slopes of -0.47,-0.54, -0.51, thus GED decaying at an order of $p^{-0.5}$ supporting Theorem 3. The same performance persists when comparing the performance on the unlearned dataset, where we compute the unlearned error divergence (UED) between the retrained and unlearned estimators (see appendix for more on UED).

We next fix $n = p = 1255$ and examine GED with X axis as $\varepsilon$ with $m = 1, 5, 10$. Figure 2 reports boxplots of GED in this setting across 200 Monte Carlo replications. Our Gaussian perturbed estimator performs increasingly closer to that of the ideal retrained estimator as $\varepsilon \uparrow$. The error for the Laplace-perturbed estimator remains high above its Gaussian counterpart, uniformly over $\varepsilon$, establishing robustness of Gaussian certifiability across all scales $\varepsilon$. Similar behavior is reflected on the unlearned subset (see Section I in appendix).

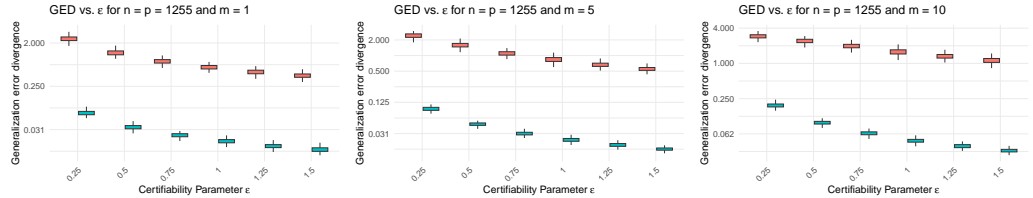

Figure 2: Comparison of GED (plotted in $\log$ scale) across different values of $\varepsilon$ for Laplace noise (in red) vs. Gaussian noise (in cyan). We set $\lambda = 0.5$.

Finally we fix $\varepsilon = 0.75$, examine GED with X axis as $m$ , as $n = p$ varies between $500, 1500, 2500$. We vary $m$ across 6 values from 5 to 50 equally spaced on a log scale. Figure 3 plots the average GED values (200 replications), transforming X and Y axes on a log scale. For all three plots, as $m \uparrow$, GED increases for both the estimators with Laplace high above Gaussian. A linear regression of $\log(\text{mean}(\text{GED}))$ vs. $\log(m)$ confirms this. The lines for Laplace unlearning had slopes of 0.26, 0.22, and 0.24 for $p = 500, 1500, 2500$, whereas Gaussian unlearning had slopes of 1.37,1.42, 1.44, showing that GED values increase at an order of approximately $m^{1.5}$ supporting Theorem 3 but suggests a better dependence on $m$ in practice.

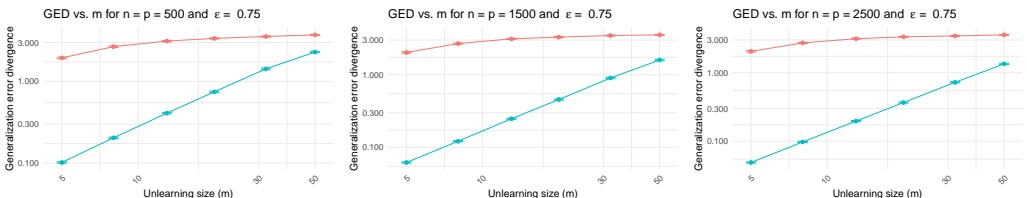

Figure 3: Comparison of mean GED (with 3 SD error bars) across the unlearning size $m$ (both in $\log$ scale) for Laplace noise (in red) vs. Gaussian noise (in cyan). We set $\lambda = 0.5$.

### 5.1 LARGE DIMENSIONAL AND REAL DATA EXPERIMENTS

In this section, we add additional experiments with larger dimension $p$. The problem setting is exactly the same as in the previous section, but $p$ now varies from 1000 to 10000. Figure 4 shows that the same pattern as in Figure 1 earlier persists here. In the above figure we have also added the GED and UED bounds from $(\phi, \varepsilon, \delta)$ unlearning of Sekhari et al. (2021) adapted to our setting. Note that $(\phi, \varepsilon)$ Gaussian certifiability implies $(\phi, \varepsilon, \delta)$ certifiability for $\delta(\varepsilon)$ determined according to Corollary 2.13 of Dong et al. (2022). The above figure shows that, beyond the dimensionality drawbacks pointed

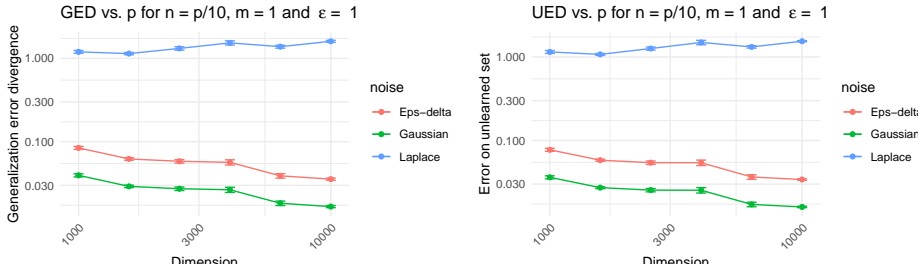

Figure 4: Comparison of mean GED and UED (with 3 SD error bars) across the dimension $p$ when the dimension is large, in that it varies from 1000 to 10000. We set $\lambda = 0.1$.

out earlier, replacing $(\phi, \varepsilon)$ Gaussian certifiability by $(\phi, \varepsilon, \delta(\varepsilon))$ certifiability for the same $\varepsilon$ leads to strictly worse performance while maintaining the same level of certifiability.

We apply our methods to binary classification tasks on real data to check performance in cases where sub-Gaussianity assumptions are violated. The two examples are the low-dimensional MNIST data where we classify digits as '3' or '4'; and the high dimensional IMDb reviews data where we predict whether a text review is positive or negative. While all three certifiability notions are comparable for the low-dimensional MNIST data, superior performance of the Gaussian framework is evident in the high-dimensional IMDb dataset from Figure 5.

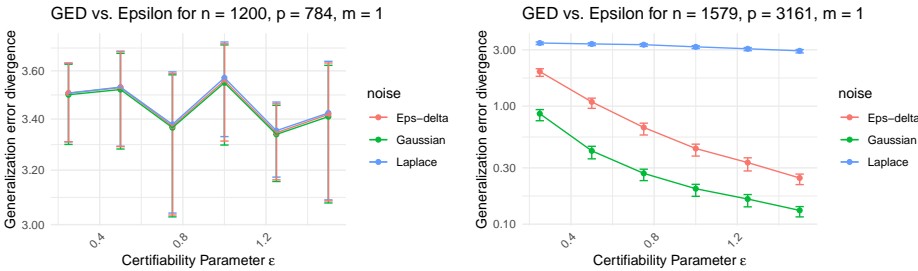

Figure 5: Comparison of mean GED (with 3 SD error bars) across $\varepsilon$ for (left) low-dimensional MNIST data with $p = 784$, $n = 1200$ and (right) high dimensional IMDb data with $p = 3161$, $n = 1579$. We set $\lambda = 0.01$. Experiments were repeated 20 times across random draws of the test set.

## 6 CONCLUSION AND FUTURE DIRECTIONS OF RESEARCH

We introduce $\varepsilon$-Gaussian certifiability and show that our noisy one Gaussian-Newton step ensures both privacy and accuracy in the high-dimensional proportional setting. This contrasts with Zou et al. (2025), which, under the $(\phi, \varepsilon)$-certifiability framework, concludes that at least two Newton steps are necessary, even for removing one data point. Our results indicate that this gap reflects the suboptimality of their $(\phi, \varepsilon)$-certifiability, which $(\phi, \varepsilon)$-Gaussian certifiability resolves optimally.

Our analysis of models with a large number of parameters opens new avenues for tackling several other important challenges. First, it would be interesting to see how much we can give up on the distributional assumptions on the data $\{x_i, y_i\}_i$ in high dimensions, and going beyond the generalized linear model 4.2. Second, it would be of great importance to analyze (un)-learning algorithms for non-convex loss functions, to capture the complicated loss surfaces arising in deep neural networks. Third, although the one-step noisy Newton proposed in this paper is theoretically efficient, but calculating the full Hessian and inverting it can be expensive in very high-dimensional settings. So, it would be interesting to investigate how the conclusions of this paper change under approximate second-order methods, such as Koh & Liang (2017), Park et al. (2023). Moreover, our one-step noisy Newton procedure effectively allows multiple simultaneous deletions of size $m$ with scaling $m^4 = o(n)$ 3, and it would be interesting to strengthen it to match with the scaling $m^3 = o(n)$ empirically observed in Figure 3, and see how the conclusion changes for more sophisticated unlearning scenarios such as distributional unlearning such as class or concept unlearning Allouah et al. (2025a), online unlearning such as handling continuous unlearning requests Qiao et al. (2025).

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

## A  APPENDIX OF GAUSSIAN CERTIFIED UNLEARNING IN HIGH DIMENSIONS: A HYPOTHESIS TESTING APPROACH

In this supplement, we provide motivations  B for some of the procedures in the paper, more background on Gaussian certifiability  C, discussion and comparison of assumptions on the per-example loss functions  D, proofs of our theoretical results  G,  H, implementation details for our experiments  I, and results from additional simulations. For completeness, we will also restate the proposed Newton-based unlearning algorithm  E and theoretical results  F in the paper.

## B  MOTIVATIONS BEHIND OUR APPROACH ON A HIGH LEVEL

In this section, we provide motivations for several aspects of our paper on a high level. First, we provide intuitions for our approach to describe the problem of machine unlearning. Then we discuss the theoretical assumption of (generalized) linearity of the model of data, and applicability of the framework in general. We end with discussing our experimental metrics and the reasoning behind their focus on prediction-level distributional differences GED and UED.

### B.1  MOTIVATION BEHIND THE DEFINITION OF MACHINE UNLEARNING

**On comparing the unlearned $\bar{A}(\hat{\boldsymbol{\beta}}, \mathcal{D}_{\mathcal{M}}, T(\mathcal{D}_n), \boldsymbol{b})$ with $\bar{A}(\hat{\boldsymbol{\beta}}_{\backslash\mathcal{M}}, \emptyset, T(\mathcal{D}_{\backslash\mathcal{M}}), \boldsymbol{b})$ and not with the ideal retrained model $A(\mathcal{D}_{\backslash\mathcal{M}})$, for the criteria of $\varepsilon-$ Gaussian certifiability .**

- Condition on $\mathcal{D}$.
  In considering $\varepsilon-$ Gaussian certifiability, the following was the reason for comparing the output $\bar{A}(\hat{\boldsymbol{\beta}}, \mathcal{D}_{\mathcal{M}}, T(\mathcal{D}_n), \boldsymbol{b})$ of 'the unlearning algorithm on the full dataset with removed set $\mathcal{M}$' and $\bar{A}(\hat{\boldsymbol{\beta}}_{\backslash\mathcal{M}}, \emptyset, T(\mathcal{D}_{\backslash\mathcal{M}}), \boldsymbol{b})$ 'the result of the unlearning algorithm on the dataset without $\mathcal{M}$ with an empty removed set'. Our hypothesis testing procedure is conditional on the data $\mathcal{D}$.

- Why condition on $\mathcal{D}$.
  This conditioning is required as unlearning happens after learning, so it should be captured almost surely or conditionally on $\mathcal{D}$. Therefore, the natural base candidate $A(\mathcal{D}_{\backslash\mathcal{M}})$, 'the learning result on the dataset without $\mathcal{M}$' to compare to, becomes deterministic. More precisely, conditionally on $\mathcal{D}$, the distribution of $A(\mathcal{D}_{\backslash\mathcal{M}})$ becomes a Dirac mass as a probability distribution and hence trivially distinguishable from 'any other' probability distribution, unless the learning algorithm is itself randomized, which we do not know a priori (in general). In our case, the learning algorithm $A$ is just Regularized Empirical Risk Minimization, which is deterministic given the data.

### B.2  MODEL ASSUMPTION AND APPLICABILITY OF THE FRAMEWORK

**On the assumption of (generalized) linearity of the model and applicability of the framework.**

- GLM in theory.
  Within high-dimensional statistics and machine learning, we consider a Generalized linear model $dP_{\boldsymbol{\beta}^*}(\boldsymbol{x}, y) = dp(\boldsymbol{x})dq(y|\boldsymbol{x}^\top \boldsymbol{\beta}^*)$ capturing a large class of statistical models, including linear, Poisson, logit, multinomial regression, single index models, etc. This is indeed a large class of models among high-dimensional machine learning models considered in the theoretical literature.

- $\varepsilon-$Gaussian certifiability in theory.
  Our focus in this paper is on closing the gap by proposing a 'universally applicable' definition of certifiability that is well-suited in high dimensions. As argued in the paper, the proposed notion is the most practical, optimally achievable, statistically interpretable, and robust notion of certifiability that optimally captures a broad class of noise adding mechanisms.

- Practical and theoretical deficiency of previous notions of certifiability.
  None of the previously proposed notions of certifiability, including the proposals of Guo et al. (2020),Sekhari et al. (2021), Zou et al. (2025) on variations of $\varepsilon-$ certifiability, and the

proposal of Allouah et al. (2025b) on a divergence-based certifiability, have all these nice theoretical properties of $\varepsilon-$Gaussian certifiability simultaneously. This essentially stems from the fact that all these previous notions are incompatible (in achieving optimality) with noise adding mechanisms, the practical way of achieving privacy. We refer the reader to Dong et al. (2022) for elaborate discussions on these deficiencies.

- $\varepsilon-$Gaussian certifiability in practice.

  *Gaussian trade-off curves* are the core of our proposed Gaussian certifiability framework, and they appear in practice, Pawelczyk et al. (2025). Here, these authors allow different choices of learning and unlearning algorithms of their choosing than the ones used to establish the theoretical results of the paper.

- Applicability of our Gaussian certifiability framework beyond the proportional regime.

  We believe that because of its intrinsic high dimensional nature, the current $(\phi, \varepsilon)$ Gaussian certifiability framework does provide a canonically solid base for any such high-dimensional machine unlearning setting, proportional ($p \sim n$) and beyond ($p \gg n$) (including modern Neural networks), with varying loss functions (neural architectures), data distributions, learning, and unlearning algorithms. Validating the Gaussian certifiability framework with other learning or unlearning algorithms in regimes beyond the proportional regime is an interesting direction to pursue in the future.

- Applicability of our more general $f$-certifiability framework.

  We believe that our more general $f$-certifiability framework applies to the more sophisticated or structured unlearning scenarios, including class or concept unlearning Allouah et al. (2025a). The algorithmic question would be an interesting direction to pursue in the future.

- Summary of utility and flexibility of our $f$ certifiability framework.

  We believe that in all the situations mentioned above, our $(\phi, \varepsilon)$ Gaussian certifiability framework 2 would serve as the tightest possible notion of certifiability in high dimensions, with changes in data distributions, loss functions, learning and unlearning algorithms. More precisely, Gaussian certifiability serves as the canonical and tightest possible notion of certifiability in high dimensions, as long as the coordinates $\hat{\beta}_i$ of the released statistic $\hat{\beta} \in \mathbb{R}^p$ are generically continuous, instead of being (structured or discrete) constrained such as binary $\hat{\beta}_i \in \{0, 1\}$. Even in such highly non-Gaussian and discrete high dimensional situations, our flexible and more general $f$ certifiability framework 2 is applicable, and we believe that our framework is flexible enough to dictate what different baseline $f$ to choose from depending on the class of problems at hand. Natural examples of non-Gaussian baseline $f$ includes $f = T(Po(1), Po(\varepsilon))$, where $Po(\varepsilon)$ is the Poisson distribution with mean $\varepsilon$. An interesting direction would be to apply the Poisson framework for sparse networks, including problems of graph unlearning with approximately Poisson degree distributions Chen et al. (2022).

- Newton-based unlearning in theory.

  Given the superiority of our proposed $\varepsilon-$ Gaussian certifiability framework in high dimensions, we restrict ourselves to showing its utility for a particular noisy Newton-based algorithm Sekhari et al. (2021) in the Generalized linear model as a foundational step .

- Newton-based unlearning in practice: (from GLM to LLM)

  Newton-based unlearning is popular in practice and has been used for neural networks in existing papers Guo et al. (2020), Bui et al. (2024). We believe that the theoretical insights gained from analyzing these Generalized linear models can help us better understand the high-dimensional behavior of more sophisticated models, including going beyond proportional regimes, and Neural Networks used in LLMs and vision.

## B.3 EXPERIMENTAL METRICS AND PREDICTION LEVEL EVALUATIONS

**On the accuracy metrics, and their focus on prediction-level distributional differences**

We do not rule out other accuracy criteria for machine unlearning problems. However, most of the existing literature has also focused on prediction-level accuracy. This emphasis is because such a criterion is more closely aligned with many AI applications, including image classification and predictive modeling. Secondly, a good prediction accuracy usually implies good 'upper-stream'

accuracies, such as in the training level. This is indeed true in our study, as Theorem 7 in the supplement material indicates. Hence, to make our results comparable with the existing literature and clarify the benefits, we also adopted a notion of prediction level accuracy. For example, see our definition of Generalization Error Divergence in the paper for measuring accuracy. Below, we explain the reason behind picking the experimental metrics GED and UED.

- GED.

  The notion of Generalization Error Divergence focuses on the distributional level generalization error in comparison to previous notions like excess risk of Sekhari et al. (2021). It has been shown in Zou et al. (2025) that such previous notions are not well-suited in high dimensions, which is based on the fact that we can allow unbounded Lipschitz bounds on the loss $\ell$ (indeed with low tolerance) compared to the worst-case bounds, which very much restricts the class of loss functions, particularly in high dimensions, including the ridge regularized empirical risk minimization.

- UED.

  In the experimental section, we evaluated the prediction accuracy on the Unlearned (Error Divergence) set as well, with the motivation that the unlearned set should not have an undue advantage over new data in terms of prediction. Indeed, the error levels are comparable on the unlearned dataset as well as the new test data, thus validating that the unlearning algorithm has successfully removed the influence of the unlearned set from the training data.

## C  GAUSSIAN CERTIFIABILITY AND CONNECTIONS WITH RELATED NOTIONS

Having outlined the motivations for our approach on a high level, we now provide intuitions for some of the more technical aspects of the assumptions made in the paper. First, we describe in detail the relations between our Gaussian certifiability framework and other previously proposed frameworks. Second, we describe from a dual perspective the emergence of Gaussian certifiability over any other framework of certifiability in high dimensions.

### C.1  THE GAUSSIAN CERTIFIABILITY FRAMEWORK: RELATIONS WITH OTHER NOTIONS

**On comparison of Gaussian certifiability with variations of $\varepsilon$-certifiability**

In short, our definition of Gaussian certifiability (with tolerance $\phi$) implies (for a certain collection of pairs of $\epsilon - \delta$ better known as 'privacy profile'), a collection of $\epsilon - \delta$ extension of the unlearning framework proposed in Zou et al. (2025) with tolerance $\phi$ as well as a collection of the certifiability definition of Sekhari et al. (2021) with tolerance level $\phi = 0$. This follows immediately from Proposition 6 and Corollary 1 of Dong et al. (2022).

**On comparison of Gaussian certifiability with Rényi divergence based certifiability**

In short, our definition of Gaussian certifiability (with tolerance $\phi$) also implies the Rényi certifiability framework of Allouah et al. (2025b) of all orders '$\alpha$' (with tolerance $\phi$) for a certain range of the other parameter. This follows immediately from Corollary B.6 in the appendix of Dong et al. (2022). So, the Gaussian certifiability framework also subsumes the Rényi divergence-based frameworks.

**Intricate zoo of different notions of certifiability and Gaussian certifiability at its center**

Having discussed that Gaussian certifiability (with tolerance $\phi$) implies all previously proposed such notions, including their privacy profile versions, we would like to highlight three key points here.

- Lossless conversions.

  First, from the Dong et al. (2022) results, Gaussian certifiability with tolerance $\phi$ is in fact equivalent to a collection of $\epsilon - \delta$ unlearning extension of the definition proposed in Zou et al. (2025) with tolerance $\phi$. Therefore, the conversion is lossless, which is not the case for conversions from Rényi certifiability to $\epsilon - \delta$ certifiability and vice versa.

- Blackwell ordering.

  Secondly, Rényi divergence (even including all orders '$\alpha$') does not determine Blackwell ordering (see Proposition B.7 in the appendix of Dong et al. (2022)). For us, it means

that Rényi divergence-based certifiability does not behave tightly under post-processing of the output of the unlearning algorithm. This is a necessary requirement for an unlearning algorithm. In contrast, trade-off functions exactly capture Blackwell ordering (see Theorem 2 of Dong et al. (2022)) and is the core of our definition of Gaussian certifiability, thereby confirming optimal behavior of our certifiability under any arbitrary post-processing.

- About tolerance $\phi$.

  Finally, as mentioned in the paper, we allow the tolerance $\phi$ to be positive, and it is of utmost importance, especially in high dimensions, and is absent in Sekhari et al. (2021), Dong et al. (2022), Allouah et al. (2025b). This allows us derivatives of the unregularized individual loss $l$ (as well as the regularizer $r$) to have polynomial growth, in particular, unbounded (albeit with small tolerance $\phi$), whereas Sekhari et al. (2021), Guo et al. (2020), Allouah et al. (2025b) requires worst-case uniform bounds on them (simultaneously with a $\Omega(1)$ strong convexity assumption), which break down in high dimensions even for ridge-regularized empirical risk minimization.

- Choosing the tolerance $\phi$ in practice.

  As mentioned above, the only practical requirement is that $\phi$ has to be positive and is completely up to the user specification, and does not affect empirical performance or computational stability above a threshold $\phi_n$ (see Theorem 2 for the value of $\phi_n$).

  Example. The dominating term for the computation of the threshold $\phi_n$ (see Theorem 4) under the proportional regime (consider the responses known to be bounded, so that $q_n^{(y)} = 0$) is given by $\frac{8}{n^3}$, so the smallest possible allowed choice of $\phi$ for $p \sim n = 10^5$ is given by approximately $8 \times 10^{-15}$, which is effectively negligible. More generally, our results prove that the minimal threshold $\phi_n = nq_n^{(y)} + 8n^{-3} + ne^{-p/2} + 2e^{-p} \to 0$ as $n, p \to \infty$, with an explicit dependence on $(n, p)$ and growth of the response $y$ (see Theorem 4), which is known for a given model of the problem (such as linear regression, binary classification or more generally whenever it is known that responses remain bounded), and the threshold of smallest $\phi$ for a fixed $(n, p)$ can be determined quite easily. Any $\phi$ larger than the threshold $\phi_n$ won't affect computational performance or stability.

## C.2 GAUSSIAN CERTIFIABILITY AS THE CANONICAL FRAMEWORK OF CERTIFIABILITY

As mentioned above, the theoretical as well as practical implications of the $(\phi, \varepsilon)$-GPAR framework are significant compared to all previously proposed notions of certifiability, especially in high dimensions. In summary, in (practice) every real-world scenario, where we achieve privacy by adding 'isotropic log-concave' noise, GPAR beats every other notion of privacy, in high dimensions.

**On Gaussian certifiability as the canonical framework of certifiability in high dimensions**

Dual perspective. A detailed answer to the above question lies in an interesting dual perspective. It is the realization that the starting point in the theoretical investigation of unlearning (privacy) is not to begin with a criterion of privacy, but with how we achieve it in practice.

- Step 1.

  The most natural and sought after way to achieve unlearning (privacy/certifiability) is to add noise to the (deterministic) algorithm output, and Gaussian noise appears as a natural candidate in high dimensions because of the following intrinsically high dimensional fact that *with high probability (up to a tolerance $\phi$), most 1d projections of an isotropic high-dimensional (log-concave) distribution are approximately Gaussian* Diaconis & Freedman (1984), Klartag (2007)[Theorem 1.1].

- Step 2.

  In the context of differential privacy, this translates into the fact that up to a tolerance $\phi$, in high dimensions, a large collection of (log-concave) noise adding mechanisms becomes comparable to the Gaussian noise addition mechanism Dong et al. (2021)[Theorem 3.1]. This makes the Gaussian perturbation mechanism the canonical noise adding mechanism in high dimensions.

- Step 3.

So, up to a tolerance $\phi$, the question reduces to devising an unlearning framework that can tightly capture the Gaussian mechanism. It turns out trade-off functions capture it in the tightest way possible through Blackwell's theorem Dong et al. (2022)[Theorem 2].

- Step 4.

  More precisely, all the previously proposed notions of unlearning, including $\epsilon - \delta$ variations of Guo et al. (2020), Sekhari et al. (2021), Zou et al. (2024), Rényi unlearning of Allouah et al. (2025b), are not tightly achievable under the Gaussian mechanism except for Gaussian certifiability Dong et al. (2022)[Theorem 1, Equation 6]. So, this establishes the natural emergence of the $(\phi, \varepsilon)$-GPAR framework with tolerance $\phi$ as superior over any other framework in high dimensions.

# D    ASSUMPTIONS ON THE UNREGULARIZED INDIVIDUAL LOSS L, REGULARIZER R AND COMPARISON WITH PREVIOUS WORKS

Before delving into precise comparisons with previous works such as Sekhari et al. (2021), Allouah et al. (2025b), we address the motivations behind some of the more generic assumptions on our unregularized individual loss $l$ and the regularizer $r$ in the proportional regime.

$$\hat{\boldsymbol{\beta}} = A(\mathcal{D}_n) := \arg\min_{\boldsymbol{\beta} \in \mathbb{R}^p} \quad L(\boldsymbol{\beta}) := \arg\min_{\boldsymbol{\beta} \in \mathbb{R}^p} \quad \lambda r(\boldsymbol{\beta}) + \sum_{i=1}^{n} \ell(y_i \mid \boldsymbol{x}_i^\top \boldsymbol{\beta}), \tag{18}$$

## D.1    ON CONVEXITY AND SMOOTHNESS OF UNREGULARIZED INDIVIDUAL LOSS L AND THE REGULARIZER R

Convexity of the pair of losses $(\ell, r)$ and strong convexity of the regularizer $r$ are assumed to ensure that there exists a unique minimum of the total loss function $L$ above (18). Smoothness of the pair $(\ell, r)$ is assumed so that the Newton method proposed in the paper makes sense theoretically.

However, we expect the theorems presented in this paper to extend to certain non-convex loss functions, potentially under stronger constraints on the size of the removal set. More specifically, as long as the local minimizer of the total loss function after data removal remains within the basin of attraction of the local minimizer for the full dataset, and the total loss function is strongly convex within that basin, a similar proof strategy should be applicable. The main remaining challenge, however, is to verify whether these assumptions hold for widely used AI models such as transformers or multilayer perceptrons (MLPs). This is certainly an interesting question for future research and one of the directions we plan to explore further. We also note that all prior theoretical works on certified unlearning cited in our paper, including Guo et al. (2020), Sekhari et al. (2021), Allouah et al. (2025b), Zou et al. (2025) assume convexity of the pair $(\ell, r)$.

## D.2    ON THE REGULARIZATION PARAMETER BEING O(1)

- Step 1- Natural growth of norm $\|\boldsymbol{\beta}^*\|_2 = B = \Omega(\sqrt{p})$ of the true parameter $\boldsymbol{\beta}^*$.

  In the high-dimensional proportional regime $p \sim n$, it is natural to assume that the $l_2$ norm $\|\boldsymbol{\beta}^*\|_2$ of the true underlying parameter $\boldsymbol{\beta}^*$ is atleast of order $\sqrt{p}$[4] One reason for such an assumption comes from a standard volume estimate of the ball of radius $B$ in dimension $p$ Vershynin (2018)[Exercise 4.29]

  $$\text{Vol}_p(B_p(B)) = \frac{\pi^{p/2}}{\Gamma\left(\frac{p}{2} + 1\right)} B^p \sim \frac{1}{\sqrt{\pi p}} \left(\frac{2\pi e \, B^2}{p}\right)^{p/2}, \tag{19}$$

  This immediately implies that unless $B > \sqrt{\frac{p}{2\pi e}}$, volume of the ball goes to zero. This translates to the fact that unless $B > \sqrt{\frac{p}{2\pi e}}$, volume of our possibilities for $\boldsymbol{\beta}^*$ shrinks to nothing as $p \to \infty$, and therefore not an interesting regime. Moreover, a stricter but more intuitive assumption on $\boldsymbol{\beta}^*$ causing this $l_2$ growth is that the coordinates of the underlying

---

[4]This is exactly assumption 3 of Sekhari et al. (2021) where the parameter $B$ should be treated as of order $\Omega(\sqrt{p})$. This is one such quantity that contributes to the blowup in all estimates of their paper.

parameter $\boldsymbol{\beta}^*$ are of order one individually, as is often assumed for precise high-dimensional asymptotics Miolane & Montanari (2021)

$$\text{weak convergence } \frac{1}{p} \sum_{i=1}^{p} \delta_{\boldsymbol{\beta}_i^*} \xrightarrow{d} \mu \text{ and } \lim_{p \to \infty} \frac{1}{p} \sum_{i=1}^{p} \boldsymbol{\beta}_i^{*2} = \int x^2 d\mu(x) = \Theta(1) \quad (20)$$

for some Borel probability measure $\mu$ on $\mathbb{R}$ with finite and positive second moment.

- Step 2- balance between noise and regularizer.
  Consider the simplest case of ridge regularized $r(\beta) = \|\beta\|^2$ quadratic loss $l(y, x^T\beta) = (y - x^T\beta)^2$[5]. Observe that the total loss has the form

$$L(\beta) = \|y - X\beta\|^2 + \lambda\|\beta\|^2 \quad (21)$$

Now, given that the true minimizer $\boldsymbol{\beta}^*$ has order one coordinates, we would also want to have order one coordinates for the minimizer $\hat{\boldsymbol{\beta}}$ of the empirical total loss $L$. In this case we even have

$$\hat{\boldsymbol{\beta}} = (X^T X + \lambda I_p)^{-1} X^T y. \quad (22)$$

Now, under our normalization of each row $x_i$ of $X$ is i.i.d. (sub)-Gaussian $\mathcal{SN}(0, \boldsymbol{\Sigma})$ with $\lambda_{\max}(\boldsymbol{\Sigma}) \leq \frac{C}{p}$, we immediately have that $\|X^T X\| = \Theta(1)$ with high probability Vershynin (2018)[Exercise 4.43]. Therefore, one would want $\lambda = \Theta(1)$ to not develop significant bias either towards the Hessian $X^T X$ of the actual loss, nor towards the Hessian $\lambda I_p$ of the regularization term[6].

## D.3 ON COMPARISON WITH SEKHARI ET AL. (2021)

In this section, we show that the results of Sekhari et al. (2021) blow up in our high-dimensional proportional regime (18), and motivates our definition of Generalization Error Divergence. First, observe that Sekhari et al. (2021) studies a one-step Newton method with Gaussian perturbation as a machine unlearning algorithm that is similar to our work.

- Step 1- mapping the optimization problem and loss functions.
  To map their result to our setting, we first set the common ground for the optimization problem of the estimator $\hat{\beta}$ and define the per-example loss $f$ in terms of the unregularized individual loss $\ell$ and regularizer $r$

$$\hat{\beta} = \arg\min_{\beta} \frac{1}{n} \sum_{i=1}^{n} f(\beta, \mathbf{z}_i) = \arg\min_{\beta} \sum_{i=1}^{n} f(\beta, \mathbf{z}_i), \text{ where} \quad (23)$$

$$f(\beta, \mathbf{z}) := \ell(y; \boldsymbol{x}^\top \boldsymbol{\beta}) + \frac{\lambda}{n} r(\boldsymbol{\beta}), \quad \text{and} \quad F(\boldsymbol{\beta}) := \mathbb{E}[f(\boldsymbol{\beta})]. \quad (24)$$

- Step 2- mapping the assumptions.
  Sekhari et al. (2021)[Theorem 4, Second equation 24] assumes that $f$ is worst case $L = O(1)$-Lipschitz, worst case $\lambda = \Theta\left(\frac{1}{n}\right)$-strongly convex, and worst case $M = O(1)$-Hessian Lipschitz, and show the following analogue of Generalization error divergence result of ours (Theorem 3) after one noisy-Newton step (substitute $\|\boldsymbol{\beta}^*\|_2 = B = \Omega(\sqrt{p})$).

$$\mathbb{E}\left| F(\tilde{\boldsymbol{\beta}}_{\backslash \mathcal{M}}) - \min_{\boldsymbol{\beta}} F(\boldsymbol{\beta}) \right| = O\left( \frac{\lambda B^2}{2} + \frac{\sqrt{p} M m^2 L^3}{\lambda^3 n^2 \varepsilon} \sqrt{\log\left(\frac{1}{\varepsilon}\right)} + \frac{m L^2}{\lambda n} \right) \quad (25)$$

$$= O\left( 1 + \frac{n\sqrt{n} M m^2 L^3}{\varepsilon} \sqrt{\log\left(\frac{1}{\varepsilon}\right)} + m L^2 \right) \quad (26)$$

---

[5] The same computation holds more generally in our setting with the same scaling for $n$ and $p$ up to constants.

[6] This example makes it clear that according to the normalization in Sekhari et al. (2021), the regularization parameter also called $\lambda$ should be taken to be of order $\Theta\left(\frac{1}{n}\right)$ throughout their paper. This is another such quantity that contributes to the blowup of all their results.

Under our high-dimensional proportional regime $p \sim n$, and from the arguments in D.2 it follows immediately that $\lambda = \Theta\left(\frac{1}{n}\right)$, $M = \Theta(1)$, and $L = \Theta(1)$. This simplifies (25) to:

$$\mathbb{E}\left|F(\tilde{\boldsymbol{\beta}}_{\setminus \mathcal{M}}) - \min_{\boldsymbol{\beta}} F(\boldsymbol{\beta})\right| = O\left(1 + \frac{n\sqrt{n}Mm^2L^3}{\varepsilon}\sqrt{\log\left(\frac{1}{\varepsilon}\right)} + mL^2\right) \quad (27)$$

which diverges as $n, p \to \infty$ with $n/p \to \gamma$, making the result inapplicable in high-dimensions.

- Summary of the failure of Sekhari et al. (2021) results and motivation for GED.

  To summarize, the bound above (27) essentially concludes that one-step noisy Newton method of Sekhari et al. (2021) under the notion of $(\epsilon, \delta)$-unlearning does not achieve vanishing change in generalization capabilities (in fact it diverges), and therefore remains inconclusive in our high-dimensional setting. We emphasize that failure of the above measure of accuracy (called *excess risk*) (25) in high dimensions motivates us to consider a different way to measure accuracy, namely through the Generalization error divergence. which we show goes to zero even in the high-dimensional proportional regime.

- Remark 1- On the regularization factor $\lambda$ (an important normalization).

  Observe the $\frac{\lambda}{n}$ is used in (24) instead of $\lambda$, since in Sekhari et al. (2021) $\lambda\|\beta\|^2$ has been added $n$ times in total and once to every example Sekhari et al. (2021)[Algorithm 4], whereas we only add it once to the entire pure loss part $\sum_{i=1}^{n} l(y_i, x_i^T\beta)$. According to the arguments of D.2 $\lambda = \Theta(1)$ in our normalization of $\|X^TX\| = \Theta(1)$. This translates to $\lambda = \Theta\left(\frac{1}{n}\right)$ in their normalization, causing blowup in (25).

  These changes in normalization of $\lambda$, although ineffective in low-dimension, turn out to be quite important in high-dimension. Finally, if some readers are still not convinced by our arguments in D.2 in favor of $\lambda = \Theta(1)$ in our normalization, observe that Sekhari et al. (2021)[Corollary 2] also finds the 'optimal choice' of $\lambda$ is of constant order in our normalization or of order $\frac{1}{n}$ in their normalization.

- Remark 2- On the worst case Lipschitz constant $L$ of loss $f$ (a weakening).

  This worst case $L = O(1)$ Lipschitzness of the per-example loss $f$ fails even for the simplest ridge regularized $r(\beta) = \|\beta\|^2$ quadratic case $\beta \to l(y|x^T\beta) = (y - x^T\beta)^2$, since quadratic functions are never worst case Lipschitz. A weakening of this stringent assumption is given in our paper (A4) by allowing the Lipschitz constant '$\dot{\ell}$' of the individual loss $\ell$ to have polynomial growth, which we will prove happens with low tolerance probability $\phi$. The key difference is that our proofs will be intricate and different in comparison to the standard proofs of such results using usual optimization assumptions.

- Remark 3- On the worst case Hessian Lipschitz constant $M$ of loss $f$ (another weakening).

- $M$ does not appear in the simplest case of ridge regularized least squares, but, we weaken this stringent worst case Hessian Lipschitzness assumption by allowing polynomial growth of '$\dddot{\ell}$' in (A4).

## D.4 ON COMPARISON WITH ALLOUAH ET AL. (2025)

In this section, we show in detail that the standard optimization assumptions of $\Omega(1)$ strong convexity and $O(1)$ smoothness of the per-example loss $f$ as made in Allouah et al. (2025b), fails completely in high dimensional proportional setting of ours. This in fact true in the simplest case of ridge regularized $r(\boldsymbol{\beta}) = \|\boldsymbol{\beta}\|^2$ quadratic (unregularized individual) loss $l(y|x^T\boldsymbol{\beta}) = (y - x^T\boldsymbol{\beta})^2$ case.

- Step 1-mapping the optimization problem and loss functions:

  To map their result to our setting, we first set the common ground for the optimization problem of the estimator $\hat{\boldsymbol{\beta}}$ and define the per-example loss $f$ in terms of the unregularized individual loss $\ell$ and regularizer $r$

$$\hat{\beta} = \arg\min_{\beta} \frac{1}{n}\sum_{i=1}^{n} f(\beta, \mathbf{z}_i) = \arg\min_{\beta} \sum_{i=1}^{n} f(\beta, \mathbf{z}_i), \text{ where} \quad (28)$$

$$f(\beta, \mathbf{z}) := \ell(y; \boldsymbol{x}^\top \boldsymbol{\beta}) + \frac{\lambda}{n} r(\boldsymbol{\beta}). \quad (29)$$

- Step 2– Ridge plus quadratic loss.

  Denote $\mathbf{z}_i = (x_i, y_i) \in \mathbb{R}^{p+1}$. Allouah et al. (2025b) assumes that per-example loss $f$ satisfy standard optimization assumptions: $\boldsymbol{\beta} \to f(\boldsymbol{\beta}, \mathbf{z}_i)$ is simultaneously

$$\text{worst case } \mu = \Omega(1) \text{ strongly convex} \iff \nabla^2 f(\boldsymbol{\beta}, \mathbf{z}_i) \succeq \mu I_p \text{ for all } \mathbf{z}_i, \qquad (30)$$

$$\text{and worse case } L = O(1) \text{ smooth} \iff \nabla^2 f(\boldsymbol{\beta}, \mathbf{z}_i) \preceq L I_p \text{ for all } \mathbf{z}_i. \qquad (31)$$

  Now, consider the ridge regularized case $r(\boldsymbol{\beta}) = \|\boldsymbol{\beta}\|^2$ with (unregularized individual) quadratic loss $l(y|x^T\boldsymbol{\beta}) = (y - x^T\boldsymbol{\beta})^2$ of our high dimensional setting $p \sim n$ (18). Then we have

$$\hat{\boldsymbol{\beta}} = \arg\min_{\boldsymbol{\beta}} \frac{1}{n} \sum_{i=1}^{n} f(\boldsymbol{\beta}, \mathbf{z}_i) = \arg\min_{\boldsymbol{\beta}} \sum_{i=1}^{n} f(\boldsymbol{\beta}, \mathbf{z}_i), \text{ with } f(\boldsymbol{\beta}, \mathbf{z}_i) = (y_i - x_i^T\boldsymbol{\beta})^2 + \frac{\lambda}{n}\|\boldsymbol{\beta}\|^2 \tag{32}$$

$$\nabla^2 f(\boldsymbol{\beta}, \mathbf{z}_i) = 2x_i x_i^T + 2\frac{\lambda I_p}{n}, \text{ and } \nabla^2\left(\sum_{i=1}^{n} f(\boldsymbol{\beta}, \mathbf{z}_i)\right) = 2X^T X + 2\lambda I_p. \tag{33}$$

- Step 3- Consequence of Hessian computation.

  Since all the (distinct) eigenvalues of the Hessian of $f$ are $\{2\frac{\lambda}{n}, 2\frac{\lambda}{n} + 2\|x_i\|_2\}$, in the high dimensional proportional regime $p \sim n$, $L = O(1)$ smoothness constant of the per-example loss $f$ inevitably requires the scaling $\|x_i\|_2 \sim 1$ (with high probability) for all $i \in [n]$ 4.2. This in turn implies $\|X^T X\| = O(1)$ (with high probability) Vershynin (2018)[Exercise 4.43] and requires a comparable $\lambda = O(1)$ D.2. But, this completely destroys the $\mu = \Omega(1)$ strong convexity of the individual loss $f$, as $\lambda_{min}\left(\nabla^2 f(\boldsymbol{\beta}, \mathbf{z}_i)\right) = 2\frac{\lambda}{n} \neq \Omega(1)$.

- Summary of the failure of assumptions made in Allouah et al. (2025b) in high dimensions.

  The above bounds essentially convey that the standard optimization assumptions made about the per-exmaple loss $f$ in Allouah et al. (2025b) do not capture the simplest model of ridge regularized least squares in high dimensions, in the context of machine unlearning in high dimensions.

- Remark 1- Our relaxed set of assumptions.

  The failure of the assumptions made in Allouah et al. (2025b) demands a new set of relaxed assumptions for the per-example loss $f$ that are well-suited in high dimensions. In this regard, we work with the weaker set of assumptions F.1. Most importantly, our assumptions on (only requiring) convexity of $\ell$ and $\nu$-Strong convexity of $r$ above actually means that per-example loss $\boldsymbol{\beta} \to f(\boldsymbol{\beta}, \mathbf{z}) = l(y, x^T\boldsymbol{\beta}) + \frac{\lambda}{n}r(\boldsymbol{\beta})$ is only required to be $\frac{\lambda\nu}{n}$ strongly convex, which is significant relaxation of the requirement of $\Omega(1)$ strong convexity of $f$, capturing the concrete case of the ridge regularized $r(\boldsymbol{\beta}) = \|\boldsymbol{\beta}\|^2$ quadratic loss $l(y, z) = (y - z)^2$.

  We end with the observation that on the algorithmic side Allouah et al. (2025b), Neel et al. (2021) focuses on proposing first-order methods of machine unlearning based on *gradient descent* and *stochastic gradient descent*, whereas we focus on a second-order Newton-based method.

## E  NOISY NEWTON-BASED UNLEARNING MECHANISM

Newton method is an iterative method for solving optimization problems based on a second-order Taylor approximation of the given optimizing function Boyd & Vandenberghe (2004).

**Definition 5.** *(Newton-Raphson method) Suppose that $h : \mathbb{R}^p \to \mathbb{R}^p$ is a differentiable function with an invertible Jacobian matrix $G$ anywhere on $\mathbb{R}^p$ and that $h$ has a root $h(\boldsymbol{\beta}^*) = 0$. Starting from an initial point $\boldsymbol{\beta}^{(0)} \in \mathbb{R}^p$, the Newton method is the following iterative procedure: for step $t \geq 1$,*

$$\boldsymbol{\beta}^{(t)} := \boldsymbol{\beta}^{(t-1)} - \boldsymbol{G}^{-1}(\boldsymbol{\beta}^{(t-1)})h(\boldsymbol{\beta}^{(t-1)}). \tag{34}$$

Using the Newton method Sekhari et al. (2021) suggested the following unlearning algorithm: Approximation step . starting from $\hat{\boldsymbol{\beta}}$, we run one Newton step with $h = \nabla L_{\backslash \mathcal{M}}$ to obtain:

$$\hat{\boldsymbol{\beta}}^{(1)}_{\backslash \mathcal{M}} = \hat{\boldsymbol{\beta}} - \boldsymbol{G}(L_{\backslash \mathcal{M}})^{-1}(\hat{\beta})\nabla L_{\backslash \mathcal{M}}(\hat{\boldsymbol{\beta}}), \text{ where } \boldsymbol{G}(L_{\backslash \mathcal{M}}) \text{ is the Hessian of } L_{\backslash \mathcal{M}}. \tag{35}$$

Randomization step. add a Gaussian noise $\boldsymbol{b} \sim N(\boldsymbol{0}, \sigma^2 \mathbb{I}_p)$ to $\hat{\boldsymbol{\beta}}_{\backslash\mathcal{M}}^{(1)}$ to have the unlearned output

$$\textbf{Noisy one step Newton output: } \bar{A}(A(\mathcal{D}_n), \mathcal{D}_\mathcal{M}, T(\mathcal{D}_n), \boldsymbol{b})) = \tilde{\boldsymbol{\beta}}_\mathcal{M} := \hat{\boldsymbol{\beta}}_{\backslash\mathcal{M}}^{(1)} + \boldsymbol{b}. \tag{36}$$

Intuition and the goal. First, the choice of $\sigma$ will be discussed in our theoretical results. Next, note that since $\hat{\boldsymbol{\beta}}_{\backslash\mathcal{M}}^{(1)}$ differs from $\hat{\boldsymbol{\beta}}_{\backslash\mathcal{M}}$, the difference between the two vectors may reveal information about the data to be removed, $\mathcal{D}_\mathcal{M}$. Hence, a standard practice is to obscure it by adding noise $\boldsymbol{b}$.

# F MAIN THEORETICAL RESULT: ASSUMPTIONS AND CONCLUSIONS

## F.1 TECHNICAL ASSUMPTIONS ON THE LOSS FUNCTIONS AND THE DATA

Loss functions. We describe the assumptions on the unregularized individual loss $l$ and the regularizer $r$ (18). To ensure that the RERM introduced in (18) has a unique solution (from strong convexity arguments) and Newton unlearning makes sense theoretically, the following assumption are made:

(A1) **Separability of $r$.** $r : \mathbb{R}^p \to \mathbb{R}_+$ is separable. More precisely, $r(\boldsymbol{\beta}) = \sum_{k \in [p]} r_k(\beta_k)$.

(A2) **Smoothness of $(l, r)$.** $\ell : \mathbb{R} \times \mathbb{R} \to \mathbb{R}_+$ and $r : \mathbb{R}^p \to \mathbb{R}_+$ are both thrice differentiable.

(A3) **Convexity of $(l, r)$.** $\ell$ and $r$ are proper convex, and $r$ is $\nu$-strongly convex for $\nu = \Theta(1) > 0$.

(A4) **Polynomial growth.** $\exists$ constants $C, s > 0$ such that for all $y, z$ (denote $\dot{\ell}(y, z) := \partial_z \ell(y, z)$)

$$\max\{\ell(y, z), |\dot{\ell}(y, z)|, |\ddot{\ell}(y, z)|\} \le C(1 + |y|^s + |z|^s) \text{ and } \nabla^2 r(\boldsymbol{\beta}) = \mathbf{diag}[\ddot{r}_k(\beta_k)]_{k \in [p]} \tag{37}$$

is $C_{rr}(n)$-Lipschitz (in Frobenius norm) in $\boldsymbol{\beta}$ for some $C_{rr}(n) = O(\text{polylog}(n))$.

Relaxed assumptions on $(\ell, r)$ and the ridge example. First, note that these assumptions on $(\ell, r)$ are inspired from Zou et al. (2025), and they weaken many of the standard optimization assumptions of $\Omega(1)$ strong convexity and $O(1)$ smoothness of per example loss $f$ in Allouah et al. (2025b) or variations thereof in Sekhari et al. (2021). Concretely, the ridge regularized $r(\beta) = \|\beta\|^2$ quadratic loss $l(y, z) = (y - z)^2$ satisfies all the assumptions above. But Allouah et al. (2025b) and Sekhari et al. (2021) both fail to capture this example in their respective frameworks, as was detailed earlier.

Data. We require the following on the data that generates our features $\boldsymbol{x}_i$ and responses $y_i$.

(B1) **Subgaussian features.** $\boldsymbol{x}_i$ are mean zero sub-Gaussian vectors with covariance $\boldsymbol{\Sigma}$ denoted as $\boldsymbol{x}_i \overset{iid}{\sim} \mathcal{SG}(\boldsymbol{0}, \boldsymbol{\Sigma})$. We further assume that $\lambda_{\max}(\boldsymbol{\Sigma}) \le \frac{C_X}{p}$, for some constant $C_X > 0$.

(B2) **Sub-polylogarithmic responses.** We assume that $\mathbb{P}(|y_i| > C_y(n)) \le q_y(n)$ and $\mathbb{E}|y_i|^{2s} \le C_{y,s}$ for some $C_y(n) = O(\text{polylog}(n))$, a constant $C_{y,s}$ and $q_n^{(y)} = o(n^{-1})$.

Intuition behind features and responses. Assumption (B1) on mean-zero sub-Gaussian features is a mild and frequently made in many papers on high dimensional statistics. See for example Miolane & Montanari (2021); Rahnama Rad & Maleki (2020); Auddy et al. (2024); Zheng et al. (2017); Donoho & Montanari (2016); Bellec et al. (2025). Assumption (B2) is not a stringent assumption either, and most popular models such as logistic regression, Poisson regression, and linear regression satisfy this assumption. Using these assumptions we can finally prove our main theoretical results.

## F.2 THEORETICAL GUARANTEES: USER PRIVACY AND MODEL ACCURACY

One step Newton achieves $(\phi, \varepsilon)$-GPAR. Our first result shows that for a suitable noise variance, the randomized one-step Newton unlearned estimator defined in (14) satisfies $(\phi, \varepsilon)$-GPAR.

**Theorem 4.** *Suppose Assumptions (A1)-(A4) as well as (B1)-(B2) hold. Then there exist poly-logarithmic $C_1(n), C_2(n) = O(\text{polylog}(n))$ for which the randomized one-step Newton unlearning (14) when used with a perturbation vector $\boldsymbol{b} \sim N\left(\boldsymbol{0}, \frac{R^2}{\varepsilon^2}\mathbb{I}_p\right)$, achieves $(\phi_n, \varepsilon)$-GPAR with*

$$R = C_1(n)\sqrt{\frac{C_2(n)m^3}{2\lambda\nu n}}, \quad \phi_n = n q_n^{(y)} + 8n^{-3} + n e^{-p/2} + 2e^{-p} \to 0. \tag{38}$$

The proof of this theorem is presented in the next section. After the above theoretical guarantee ensuring user privacy, we now calculate the GED of this machine unlearning algorithm.

**Theorem 5.** *Suppose Assumptions (A1)-(A4) as well as (B1)-(B2) hold. Consider the unlearning estimator defined in* (14) *with the noise variance set according to Theorem 2. Then, with probability at least* $1 - (n+1)q_n^{(y)} - 14n^{-3} - ne^{-p/2} - 2e^{-p} - e^{-(1-\log(2))p}$,

$$\text{GED}(\tilde{\boldsymbol{\beta}}_{\backslash \mathcal{M}}, \hat{\boldsymbol{\beta}}_{\backslash \mathcal{M}}) \leq C_1(n)\sqrt{C_2(n)} \left( \frac{1}{\varepsilon} + \frac{1}{\sqrt{p}} \right) \sqrt{\frac{m^3(m+2)}{\lambda \nu n}} \cdot \text{polylog}(n).$$

### F.3 ON THE PROBLEM OF DETERMINING THE POLYLOGARITHMIC FACTORS $C_1(n), C_2(n)$ AND DECIDING ON THE NOISE VARIANCE $\sigma$ IN PRACTICE ON A GIVEN DATASET $\mathcal{D}$

In this section, we discuss the perspective of a practitioner on how one decides on the noise variance $\sigma$ of the Gaussian mechanism $\boldsymbol{b}$ in practice for a given dataset $\mathcal{D}$, following our noisy-Newton-based framework to ensure both certifiability 4 and generalizability 5. This will lead us to discuss the dependence of the certifiability bound (Theorem 4) and the generalization bound (Theorem 5) on the noise variance $\sigma$, and eventually to the problem of determining or estimating the polylogarithmic factors $C_1(n), C_2(n)$ for a given dataset $\mathcal{D}$ and for a specific choice of loss functions $(\ell, r)$.

1. Dependence on $\sigma$.
   From the proof, it will be immediate that the $(\phi, \varepsilon)$ certifiability bound of Theorem 4 is satisfied for all choices of noise variance $\sigma$ above the critical threshold of $\sigma = \frac{R}{\varepsilon}$ with $R$ as mentioned in (38)[7]. Therefore, any amount (small or large) of positive deviations in user-chosen noise variance $\sigma$ above the critical choice would not break the certifiability guarantees of Theorem 4. However, as expected, with increasing noise variance $\sigma$, the generalization error bound in Theorem 5 will deteriorate. But, the dependence of the generalization error bound 5 is linear in $\sigma$ as will be immediate from the proof, in the dominating term of the error involving $\frac{1}{\varepsilon}$. So, the generalization error bound of Theorem 5 will deteriorate only linearly with the noise variance $\sigma$ above its critical threshold. Therefore, the incentive lies with the user to choose the smallest possible value of noise variance $\sigma$, or equivalently estimate the smallest value of $R$ allowing the conclusion of Theorem 4 to hold given the dataset, user's choice of loss functions, and other available problem parameters.

2. Computation of $C_1(n), C_2(n)$.
   From the discussion above, it is clear that to utilize the theoretical guarantees provided by our noisy-Newton based framework for a given dataset $\mathcal{D} = \{x_i, y_i\}_{i=1}^n$, one needs to explicitly compute an upper bound (as small as possible) on the value of $R$, as mentioned in Theorem 5, in terms of parameters available to (or chosen by) the practitioner.

3. Features.
   First, to match the dataset with our framework of assumptions on data distributions, one must center the feature vectors $\{x_i\}_{i=1}^n \rightarrow \{x_i - \bar{x}\}_i$ with $\bar{x} = \frac{1}{n}\sum_{i=1}^n x_i$, and scale them[8] in such a way that they have the scaling of our assumptions (B1). The computation of the empirical (after centering and scaling) covariance matrix $\hat{\Sigma} = \frac{1}{n}\sum_{i=1}^n (x_i - \bar{x})(x_i - \bar{x})^T$, and its corresponding top eigenvalue $\lambda_{max}(\hat{\Sigma})$ can be done explicitly. Although not addressed in the theoretical results of this paper, under the sub-Gaussian assumptions on the features, $\hat{\Sigma}$ is strongly concentrated around its mean $\Sigma$, the population covariance matrix (see remark 4.7.3 of Vershynin (2018)). As a consequence of this and since the eigenvalues of positive definite matrices are Lipschitz functions satisfying $|\lambda_{max}(A) - \lambda_{max}(B)| \leq \|A - B\|$, with probability $1 - 2e^{-p}$ we have $\lambda_{max}(\hat{\Sigma}) \leq \lambda_{max}(\Sigma) + \|\hat{\Sigma} - \Sigma\| \leq \frac{C_X}{p} + C(\sqrt{\frac{1}{\gamma}} + \frac{1}{\gamma})\frac{C_X}{p} = \frac{C_X'}{p}$ for some universal constant $C > 0$. Therefore, the conclusions of Theorems 4 and

---

[7]One can consider smaller levels of noise $\sigma < \frac{R}{\varepsilon}$, and our results would not break down. But, as expected, going to smaller $\sigma$ would require us to analyze the dependence of the probability $\mathbb{P}[\max_{|\mathcal{M}| \leq m} \|\hat{\boldsymbol{\beta}}_{\backslash \mathcal{M}}^{(1)} - \hat{\boldsymbol{\beta}}_{\backslash \mathcal{M}}\| \geq r]$ as a function of $r (\leq R)$, involved in Theorem 7 which will be done elsewhere.

[8]One way to scale is to divide each vector $x_i - \bar{x}$ by the $l_2$ norm of itself to have each of them lie on the unit sphere in dimension $p$. This was the intuition behind our scaling with dimension assumption (B1).

5 will be useful in practice, since the empirical covariance matrix satisfies the same set of assumptions required in our framework with high probability, as is the case for the population covariance. In summary, it's safe in practice to work directly with the empirical covariance matrix $\hat{\Sigma}$, rather than the practically unavailable population version. Finally, deciding whether the features (even after centering and scaling) of the given dataset are actually coming from a sub-Gaussian population or equivalently keeping our conclusions of Theorems 4, 5 without making any distributional assumptions on the features $\{x_i\}_i$, is an interesting question for future study.

4. Response.

For the responses $y$, we have the technical assumption (B2) requiring (probability bound) $\mathbb{P}(|y_i| > C_y(n)) \le q_y(n)$ and (expectation bound) $\mathbb{E}|y_i|^{2s} \le C_{y,s}$ for some $C_y(n) = O(\mathrm{polylog}(n))^9$, a constant $C_{y,s}$ and $nq_n^{(y)} = o(1)$. Both of these technical assumptions are used in our proofs, and most importantly, the quantity $nq_n^{(y)}$ appears in the 'tail' probability of both of our Theorems 4, 5. Indeed, in practice, one can try to verify whether the responses in the given dataset (available to the practitioner) satisfy the above assumptions of our framework by computing the empirical probability bound and empirical expectation bound with respect to the background probability measure $\hat{\mathbb{P}}_n = \frac{1}{n}\sum_{i=1}^n \delta_{y_i}$. Although not addressed in the theoretical results of this paper, since the responses are one-dimensional, there are no high-dimensional consequences of replacing $\mathbb{P}$ with $\hat{\mathbb{P}}_n$ while computations only involve the responses $\{y_i\}_i$. This is the same we do in the classical central limit theorem to find confidence intervals, and one expects strong concentration of functionals (such as moments or tail probabilities) of $\hat{\mathbb{P}}_n$ to be around functionals of $\mathbb{P}$ for a large class of marginal distributions on the response $y$, encompassing sub-Gaussian responses and many others. However, for a large class of responses $y$ for which the responses $\{y_i\}_i$ are already known (to the practitioner) to be bounded by $M$, such as binary or multiclass classification, there is no need for empirical verification with $\hat{\mathbb{P}}_n$ as described above; our framework applies directly, since both conditions are satisfied with $C_y(n) = M + 1$, and $C_{y,s} = M^s$ with $q_n^{(y)} = 0$. As will be shown later that our framework also applies to problems where the practitioner is trying to model a feature-responsive dataset $\{x_i, y_i\}_i$ coming from a high-dimensional linear regression, logistic regression. Even more generally, with the empirical measure, $\hat{\mathbb{P}}_n$ the practitioner can choose the growth constants $C_y(n), C_{y,s}$ of their choice so that they can control the value of $nq_n^{(y)}$ and their desired error probability in our results 4, 5.

5. Loss and the regularizer.

We will soon show that the computation of the constants $C_1(n), C_2(n)$ within the class of separable regularizer (A1), smooth loss and regularizer (A2), and convex loss (A3), depends through the strong convexity constant of (only) the regularizer $r$ (A3), and involve the 'Lipschitz-type' constants for the polynomial growth conditions for both the derivatives of loss and regularizer (A4). For high-dimensional applicability, we would like to make our framework available for a large collection of pairs of loss functions $\ell$ and regularizers $r$, going far beyond the implicitly low-dimensional standard optimization assumptions of Sekhari et al. (2021), Allouah et al. (2025a). These relaxations in assumptions and attempts to accommodate our results simultaneously for a large class of loss-regularizer pairs make the constants $C_1(n), C_2(n)$ necessarily large. However, for a fixed choice of $(\ell, r)$, these constants can be computed, although we do not claim any sharpness of these constants (shown below) up to the constant order $(1 + o(1))$, since determining the constants in high dimensions often require exact asymptotics, whereas our focus was on getting practically useful and directly applicable non-asymptotic bounds. Before writing down a general expression for $C_1(n), C_2(n)$, we describe a procedure to apply in practice to decide on the noise level $\sigma$ and finally compute these constants $C_1(n), C_2(n)$ for linear regression and logistic regression examples.

6. Calibrating the noise level $\sigma$.

---

[9]We will show explicitly that $C_y(n) = \sqrt{\log n}$ with $nq_n^{(y)} = n^{-2}$ for high dimensional linear regression and $C_{y,n} = k + 1$ (no polylog) for binary or multiclass $k$ classification with $q_n^{(y)} = 0$.

From the proof of Theorem 4 (see equation (61)) noise $\boldsymbol{b} \sim N(0, \sigma^2 \mathbb{I}_p)$ is injected in the estimate $\hat{\boldsymbol{\beta}}_{\backslash \mathcal{M}}^{(1)}$, where $\sigma^2 = \frac{R^2}{\varepsilon^2}$, where $R = C_1(n)\sqrt{\frac{C_2(n)m^3}{2\lambda\nu n}}$ as an upper bound to $R' = \max_{|\mathcal{M}| \leq m} R'_{\mathcal{M}}$, where $R'_{\mathcal{M}} = \left\| \hat{\boldsymbol{\beta}}_{\backslash \mathcal{M}} - \hat{\boldsymbol{\beta}}_{\backslash \mathcal{M}}^{(1)} \right\|_2$.

- First calibration method.
  In practice, the unlearning algorithm designer would know what the deletion set $\mathcal{M}$ is (along with the corresponding dataset $\{x_i, y_i\}_{i \in \mathcal{M}}$), so its enough to bound (or empirically estimate) $R'_{\mathcal{M}} = \left\| \hat{\boldsymbol{\beta}}_{\backslash \mathcal{M}} - \hat{\boldsymbol{\beta}}_{\backslash \mathcal{M}}^{(1)} \right\|_2$ in practice for a given deletion request $\mathcal{M}$[10]. Observe that the unlearning algorithm designer does not have access to $\hat{\boldsymbol{\beta}}_{\backslash \mathcal{M}}$ in our unlearning framework. Hence the ideal choice of

$$\text{the noise level } \sigma = \frac{R'_{\mathcal{M}}}{\varepsilon} \text{ is not possible in our framework.} \qquad (39)$$

Although not covered directly by our theoretical results, in practice one can choose a much more computationally explicit noise level $\sigma$ given by [11]

$$\sigma = \frac{R''_{\mathcal{M}}}{\varepsilon}, \text{ where } R''_{\mathcal{M}} = \| \left[ \bar{\boldsymbol{G}}_{\backslash \mathcal{M}}''^{-1} - \boldsymbol{G}_{\backslash \mathcal{M}}^{-1}(\hat{\boldsymbol{\beta}}) \right] \Big( \sum_{i \in \mathcal{M}} \dot{\ell}_i(\hat{\boldsymbol{\beta}})x_i \Big) \|_2, \qquad (40)$$

where each of the quantities are defined below explicitly and is available in practice. Now, we explain the intuition behind the above choice of $\sigma$ given deletion request $\mathcal{M}$. From the first order conditions ($\nabla L(\hat{\boldsymbol{\beta}}) = \nabla L_{\backslash \mathcal{M}}(\hat{\boldsymbol{\beta}}_{\backslash \mathcal{M}}) = 0$) for the total empirical loss $L$ (1) and its deleted version $L_{\backslash \mathcal{M}}$ (2) and applying integral form of Taylor's theorem ($f(x) - f(x_0) = \int_0^1 f'(tx + (1-t)x_0)dt$) to the gradients $\nabla L, \nabla L_{\backslash \mathcal{M}}$ at their respective minimizers $x_0 = \hat{\boldsymbol{\beta}}, \hat{\boldsymbol{\beta}}_{\backslash \mathcal{M}}$, we have the following exact representation

$$\hat{\boldsymbol{\beta}}_{\backslash \mathcal{M}} - \hat{\boldsymbol{\beta}} = \bar{\boldsymbol{G}}_{\backslash \mathcal{M}}^{-1} \Big( \sum_{i \in \mathcal{M}} \dot{\ell}_i(\hat{\boldsymbol{\beta}})x_i \Big), \text{ where } \bar{\boldsymbol{G}}_{\backslash \mathcal{M}} := \int_0^1 \boldsymbol{G}_{\backslash \mathcal{M}}\big(t\hat{\boldsymbol{\beta}} + (1-t)\hat{\boldsymbol{\beta}}_{\backslash \mathcal{M}}\big) \, dt, \qquad (41)$$

where $\boldsymbol{G}_{\backslash \mathcal{M}}(\boldsymbol{\beta})$ denotes the Hessian of total empirical loss (after deletion) $L_{\backslash \mathcal{M}}$ (2). After one Newton step $\hat{\boldsymbol{\beta}}_{\backslash \mathcal{M}}^{(1)}$ on $L_{\backslash \mathcal{M}}$ starting at the trained ML model $\hat{\boldsymbol{\beta}}$ (35) we have

$$\hat{\boldsymbol{\beta}}_{\backslash \mathcal{M}} - \hat{\boldsymbol{\beta}}_{\backslash \mathcal{M}}^{(1)} = \left[ \bar{\boldsymbol{G}}_{\backslash \mathcal{M}}^{-1} - \boldsymbol{G}_{\backslash \mathcal{M}}^{-1}(\hat{\boldsymbol{\beta}}) \right] \Big( \sum_{i \in \mathcal{M}} \dot{\ell}_i(\hat{\boldsymbol{\beta}})x_i \Big) \qquad (42)$$

Observe that the quantity that we are supposed to compute (or estimate) is given by

$$R'_{\mathcal{M}} = \|\hat{\boldsymbol{\beta}}_{\backslash \mathcal{M}} - \hat{\boldsymbol{\beta}}_{\backslash \mathcal{M}}^{(1)}\|_2 = \| \left[ \bar{\boldsymbol{G}}_{\backslash \mathcal{M}}^{-1} - \boldsymbol{G}_{\backslash \mathcal{M}}^{-1}(\hat{\boldsymbol{\beta}}) \right] \Big( \sum_{i \in \mathcal{M}} \dot{\ell}_i(\hat{\boldsymbol{\beta}})x_i \Big) \|_2 \qquad (43)$$

but unfortunately the right hand side of (42) is uncomputable in our unlearning framework, since $\bar{\boldsymbol{G}}_{\backslash \mathcal{M}}$ involves the ideal unlearned estimator $\hat{\boldsymbol{\beta}}_{\backslash \mathcal{M}}$, unavailable to the practitioner. So, in practice, we replace $\hat{\boldsymbol{\beta}}_{\backslash \mathcal{M}}$ with one-step Newton estimator $\hat{\boldsymbol{\beta}}_{\backslash \mathcal{M}}^{(1)}$ in the expression $\bar{\boldsymbol{G}}_{\backslash \mathcal{M}}$ for the computation of $R'_{\mathcal{M}} = \|\hat{\boldsymbol{\beta}}_{\backslash \mathcal{M}} - \hat{\boldsymbol{\beta}}_{\backslash \mathcal{M}}^{(1)}\|_2$, and call that estimate of $R'_{\mathcal{M}}$ as $R''_{\mathcal{M}}$ given by the following[12].

$$R''_{\mathcal{M}} = \| \left[ \bar{\boldsymbol{G}}_{\backslash \mathcal{M}}''^{-1} - \boldsymbol{G}_{\backslash \mathcal{M}}^{-1}(\hat{\boldsymbol{\beta}}) \right] \Big( \sum_{i \in \mathcal{M}} \dot{\ell}_i(\hat{\boldsymbol{\beta}})x_i \Big) \|_2 = \|(H_1^{-1} - H_2^{-1})v\|_2, \qquad (44)$$

---

[10]Our theoretical results (see equation (61)) is covering a worst case situation over all deletion requests, but in practice one knows the deletion request before running the unlearning algorithm, and can utilize that information by choosing a smaller (and more informed) noise level $\sigma$.

[11]We denote $\dot{\ell}_i(\hat{\boldsymbol{\beta}})$ to mean $\partial_2 l(y_i, x_i^T \hat{\boldsymbol{\beta}})$, the first derivative of at the $i$th data point $(x_i, y_i)$ evaluated at $x_i^T \hat{\boldsymbol{\beta}}$.

[12]Recall that $\boldsymbol{G}$ denotes the Hessian of the total empirical loss $L$ and it is related to the Hessian $\boldsymbol{G}_{\backslash \mathcal{M}}$ of the total (deleted) empirical loss $L_{\backslash \mathcal{M}}$ as $L(\boldsymbol{\beta}) = L_{\backslash \mathcal{M}}(\boldsymbol{\beta}) + \sum_{i \in \mathcal{M}} \ell(y_i, x_i^T \boldsymbol{\beta})$, so the computation (evaluations at various $\boldsymbol{\beta}$s) involving $\boldsymbol{G}_{\backslash \mathcal{M}}$ (potentially unavailable) can be reduced to the computation of the Hessian $\boldsymbol{G}$ of the available losses $L$ (the total empirical loss used during training) and the deletion part $\sum_{i \in \mathcal{M}} \ell(y_i, x_i^T \boldsymbol{\beta})$.

$$\text{where } \bar{\boldsymbol{G}}''_{\backslash\mathcal{M}} := \int_0^1 \boldsymbol{G}_{\backslash\mathcal{M}}(t\hat{\boldsymbol{\beta}} + (1-t)\hat{\boldsymbol{\beta}}^{(1)}_{\backslash\mathcal{M}}) \, dt. \tag{45}$$

Now, a priori computing the integral numerically inside $\bar{\boldsymbol{G}}''_{\backslash\mathcal{M}}$ seems difficult, and one can in practice think of applying some general purpose tools to simplify computations, such as Monte-Carlo, by uniformly sampling several points within the unit interval and taking the average as a proxy, or can use quadrature formulas of their choosing.

After having discussed some standard procedures to compute the integral, we now discuss the extra structure that we have for our problem which might also help the practitioner to compute these quantities quickly. First, observe that we are only required to compute a quantity of the form $\|H_1^{-1}v - H_2^{-1}v\|_2$, and there are also numereical schemes that can compute quickly $H^{-1}v$, since $Hw = \nabla^2 f(x)w = \frac{d}{d\delta}\nabla f(x+\delta w)|_{\delta=0}$ is a Hessian matrix-vector product for some function $f$ at a point $x$, so we can get away without actually computing the entire matrix $H$. Second, our Hessian matrices are structured enough in the generalized linear model class that the high dimensional matrix integral reduces to simpler tractable scalar integrals[13] .

$$\boldsymbol{G}_{\backslash\mathcal{M}}(\boldsymbol{\beta}) := \boldsymbol{X}_{\backslash\mathcal{M}}^\top \mathbf{diag}[\ddot{l}_i(\boldsymbol{\beta})]_{i\notin\mathcal{M}}\boldsymbol{X}_{\backslash\mathcal{M}} + \lambda\mathbf{diag}[\ddot{r}_k(\boldsymbol{\beta})]_{k\in[p]} \tag{46}$$

$$\bar{\boldsymbol{G}}''_{\backslash\mathcal{M}} := A + B \text{ where} \tag{47}$$

$$A = \boldsymbol{X}_{\backslash\mathcal{M}}^\top \mathbf{diag}[\int_0^1 \ddot{l}_i((t\hat{\boldsymbol{\beta}} + (1-t)\hat{\boldsymbol{\beta}}^{(1)}_{\backslash\mathcal{M}}))dt]_{i\notin\mathcal{M}}\boldsymbol{X}_{\backslash\mathcal{M}} \tag{48}$$

$$B = \lambda\mathbf{diag}[\int_0^1 \ddot{r}_k((t\hat{\boldsymbol{\beta}} + (1-t)\hat{\boldsymbol{\beta}}^{(1)}_{\backslash\mathcal{M}}))dt]_{k\in[p]} \tag{49}$$

Now for a given loss-regularization pair $(\ell, r)$, the scalar integrals above can often be computed often in exact form or using some standard quadrature formulas. Having described the calibration method of computing the noise level $\sigma$ in practice, we now provide an argument why is that calibration procedure theoretically sound[14]. The problem boils down to a computation of the upper bound on the following difference

$$|R''_{\mathcal{M}} - R'_{\mathcal{M}}| \leq \|\left(\bar{\boldsymbol{G}}_{\backslash\mathcal{M}}^{-1} - \bar{\boldsymbol{G}}''^{-1}_{\backslash\mathcal{M}}\right)\left(\sum_{i\in\mathcal{M}} \dot{\ell}_i(\hat{\boldsymbol{\beta}})x_i\right)\|_2 \tag{50}$$

Going again back to our assumptions (A4) and following the proof of Theorem 7 in Zou et al. (2025) on the high probability Lipschitz bound on the third derivative on the pair $(\ell, r)$ and the tool developed in Zou et al. (2025)[Lemma A.1] help us show that this difference goes to zero with high probability. In summary, the plugin estimator in the fixed point equation (41) could help us choose the noise level $\sigma$ in practice (40).

- Second calibration method.
  Inspired by Zou et al. (2025), we propose to compute $R'$ instead, although computing $R' = \max_{|\mathcal{M}|\leq m} R'_{\mathcal{M}}$, where $R'_{\mathcal{M}} = \left\|\hat{\boldsymbol{\beta}}_{\backslash\mathcal{M}} - \hat{\boldsymbol{\beta}}^{(1)}_{\backslash\mathcal{M}}\right\|_2$. requires evaluating all $\binom{n}{|\mathcal{M}|}$ possible subsets, which is computationally infeasible even for moderate $n$. Hence, instead of exhaustively enumerating all $\binom{n}{|\mathcal{M}|}$ configurations, we select a random subset of size (small) $m_0 = 100$ and compute the maximum $\|\hat{\boldsymbol{\beta}}_{\backslash\mathcal{M}} - \hat{\boldsymbol{\beta}}^{(1)}_{\backslash\mathcal{M}}\|_2$ over this subset (including the exact unlearned estimates). To approximate the maximum, we rescale the result by a factor of $\sqrt{\log\binom{n}{|\mathcal{M}|}/\log m_0}$.

7. Exact expression of $C_1(n), C_2(n)$.
   An inspection of our proof of 4 would reveal that the expression of the implicit constants $C_1(n), C_2(n)$ boils down to the constants arise in the proof of Theorem 7 of Zou et al. (2025). Here we summarize their exact formulas in terms of the following constants:

---

[13]We denote $\ddot{l}_i(\hat{\boldsymbol{\beta}})$ to mean $\partial_{22}l(y_i, x_i^T\hat{\boldsymbol{\beta}})$, the second derivative of at the $i$th data point $(x_i, y_i)$ evaluated at $x_i^T\hat{\boldsymbol{\beta}}$, and the same for the separable regularizer $r$.

[14]A completely rigorous proof of this calibration method would be an interesting direction for future research.

- $\lambda$: the regularization strength,
- $\gamma_0 = n/p$,
- $\nu$ is the strong convexity parameter of the regularizer,
- $C_X$ is the constant in Assumption (B1): $\|\mathbb{E}x_1 x_1^\top\| = \|\Sigma\| \leq p^{-1} C_X$

we have from Zou et al. (2025)

$$C_1(n) = \frac{2\sqrt{3}}{3\lambda^2 \nu^2}[C_{\ell\ell}(n) + \lambda C_{rr}(n)]C_1(C_1 + 1)C_\ell^2(n)C_{xx}(n),$$

$$C_2(n) = (1 + (\sqrt{\gamma_0} + 3)\sqrt{C_X})^2(C_{\ell\ell}(n) + \lambda C_{rr}(n)), \text{ where}$$

- $C_1 = (\sqrt{\gamma_0} + 3)\sqrt{C_X}$ is a high probability bound for $\|X\|$,
- $C_l(n)$ is a high probability bound for $\max_{i\in[n]} |\dot{\ell}_i(\hat{\beta})|$ (the explicit term is $\mathrm{polylog}_3(n)$ below)
- $C_{ll}(n)$ is a (high probability) Lipschitz constant for $\ddot{l}_{\mathcal{M}}(\beta)$ in a neighborhood of $\hat{\beta}_{\backslash\mathcal{M}}$ (the explicit expression is $\max\{\mathrm{polylog}_{10}(n), \mathrm{polylog}_{11}(n)\}$ below),
- $C_{rr}(n)$ is the Lipschitz constant of $\ddot{r}(\beta)$ in Assumption (A4),
- $C_{xx}(n)$ is a high probability upper bound for the following quantity $\max_{|\mathcal{M}|\leq m} \sqrt{\frac{n}{m}}\|\bar{X}_{\backslash\mathcal{M}} G_{\backslash\mathcal{M}}^{-1}(\hat{\beta}_{\backslash\mathcal{M}}) X_{\mathcal{M}}^\top\|_{2,\infty}$ and is given by

$$C_{xx}(n) = \frac{(\sqrt{\gamma_0} + 3)C_X \vee 1}{\lambda\nu}\sqrt{\frac{m(1 + 20\log(n))}{p}}$$

.

The explicit expressions of the $\mathrm{polylog}(n)$ terms are:

$$\mathrm{polylog}_1(n) = 2\sqrt{5(\lambda\nu)^{-1}\gamma_0 C_X(1 + C_y^s(n))\log(n)}$$

$$\mathrm{polylog}_2(n) = 1 + C_y^s(n) + \mathrm{polylog}_1^s(n)$$

$$\mathrm{polylog}_3(n) = \mathrm{polylog}_1(n) + \frac{C_X}{\lambda\nu}\mathrm{polylog}_2(n)$$

$$\mathrm{polylog}_4(n) = 1 + C_y^s(n) + \mathrm{polylog}_3^s(n)$$

$$\mathrm{polylog}_5(n) = \frac{2(\sqrt{\gamma_0} + 3)C_X}{\lambda\nu}\sqrt{m(2m + 4)p^{-1}\log(n)}\mathrm{polylog}_4(n)$$

$$\mathrm{polylog}_6(n) = \mathrm{polylog}_1(n) + \mathrm{polylog}_5(n)$$

$$\mathrm{polylog}_7(n) = 1 + C_y^s(n) + \mathrm{polylog}_6^s(n)$$

$$\mathrm{polylog}_8(n) = \mathrm{polylog}_6(n) + \frac{4C_X}{\lambda\nu}\mathrm{polylog}_7(n)$$

$$\mathrm{polylog}_9(n) = 1 + C_y^s(n) + \mathrm{polylog}_8^s(n)$$

$$\mathrm{polylog}_{10}(n) = 1 + C_y^s(n) + \mathrm{polylog}_9^s(n)$$

$$\mathrm{polylog}_{11}(n) = 1 + C_y^s(n) + |\mathrm{polylog}_8(n) + 2\sqrt{C_X}|^s. \tag{51}$$

Note that the above cumbersome constants are for general classes of problems. For specific problems (see the following example) they could take much simpler forms. Another comment is that in practice we don't know $C_X$, the largest eigenvalue of $p\Sigma$, but a researcher with a specific model instance can estimate it through the sample covariance $C_1$. Details are in the '3.Features' part above.

**Example 3.1 (Linear ridge regression).** Suppose $y_i \mid x_i \sim \mathcal{N}(x_i^\top \beta^*, \sigma^2)$, then we have $y_i \sim \mathcal{N}(0, \tau^2)$ with

$$\tau^2 := \sigma^2 + \frac{1}{p}\|\beta^*\|^2.$$

Its negative log-likelihood is the $\ell_2$ loss:

$$\ell(y, z) = \frac{1}{2}(y - z)^2, \qquad \dot{\ell}(y, z) = z - y, \qquad \ddot{\ell}(y, z) = 1, \qquad \dddot{\ell}(y, z) = 0.$$

If we adopt a ridge penalty then the constants can be further simplified into $\nu = 1, C_{ll}(n) = C_{rr}(n) \equiv 0$. This would inevitably cause the noise scale $R = 0$, and this is actually consistent with the fact that one Newton step for quadratic loss function achieves exact solution $\hat{\beta}_{\setminus \mathcal{M}}$ (broadly comes under the exact unlearning framework).

**Example 3.2 (Logistic regression).** Suppose $y_i \sim \text{Bernoulli}(p_i)$ where

$$p_i = \left(1 + e^{-x_i^\top \beta^*}\right)^{-1}.$$

The negative log-likelihood is then

$$\ell(y, z) = y \log(1 + e^{-z}) + (1 - y) \log(1 + e^z), \qquad y \in \{0, 1\},$$

and in particular $\ell(y, z) \le 2 \log(2) + 2z$.
Obviously $|y_i| \le 1$ and moreover,

$$\left|\dot{\ell}(y, z)\right| = \left|\frac{e^z}{1 + e^z} - y\right| \le 1 + |y| \le 2,$$

$$\left|\ddot{\ell}(y, z)\right| = \left|\frac{e^z}{(1 + e^z)^2}\right| \le 1,$$

$$\left|\dddot{\ell}(y, z)\right| = \left|-\frac{1}{1 + e^z} + \frac{3}{(1 + e^z)^2} - \frac{2}{(1 + e^z)^3}\right| \le 6,$$

so we have $C_l(n) = 2, C_{ll}(n) = 6$. If we use a ridge penalty and assume $p = n, \Sigma = p^{-1}\mathbb{I}_p$ then $\gamma_0 = C_X = 1$ and $C_{rr}(n) = 0$, and we have

- $C_1 = (\sqrt{\gamma_0} + 3)\sqrt{C_X} = 4$
- $C_{xx}(n) = \frac{4}{\lambda}\sqrt{\frac{m}{p}(1 + 20\log(n))}$
- $C_1(n) = \frac{1280\sqrt{3}}{\lambda^3}\sqrt{\frac{m}{p}(1 + 20\log(n))}$.
- $C_2(n) = 150$.

Again, when $C_X$ is unknown to the researcher, it can be estimated from data $X$ as described in '3. Features' above.

## G  PROOFS IN THE CASE OF GAUSSIAN FEATURES

The main goal of this section is to present the proofs of Theorem 2 and Theorem 3 assuming when $x_i \sim N(0, \Sigma)$ with $\lambda_{\max}(\Sigma) \le \frac{C_X}{p}$ for some constant $C_X > 0$. After reviewing some preliminary results that will be used in the proof in Section G.1 we present the proof of Theorem 2 in Section G.2 and the proof of Theorem 3 in Section G.3.

### G.1  PRELIMINARIES

Our first lemma summarizes some of the basic but extremely fruitful properties of the Gaussian trade-off function defined in the main text.

**Lemma 6.** *(Dong et al. (2022)) The Gaussian trade off functions satisfy the following properties:*

- *Monotonicity: For any pair $\varepsilon_1, \varepsilon_2 \ge 0$, $\varepsilon_1 \le \varepsilon_2$ if and only if*

$$f_{G,\varepsilon_1}(\alpha) \ge f_{G,\varepsilon_2}(\alpha) \text{ for all } \alpha \in [0, 1]. \tag{52}$$

- *Closure under suprema: For any collection of $(\varepsilon_i)_{i \in I} \subset \mathbb{R}$ with index set $I$*

$$\inf_{i \in I} f_{G,\varepsilon_i} = f_{G, \sup_{i \in I} \varepsilon_i} \text{ for all } \alpha \in [0, 1] \tag{53}$$

- *Symmetry: For any $\mu_1, \mu_2 \in \mathbb{R}^p$ and $\sigma > 0$*

$$T(\mu_2 + \sigma N(0, \mathbb{I}_p), \mu_1 + \sigma N(0, \mathbb{I}_p)) = T(\mu_1 + \sigma N(0, I_p), \mu_2 + \sigma N(0, I_p)) \tag{54}$$

- *Dimension freeness: For any $\mu_1, \mu_2 \in \mathbb{R}^p$ and $\sigma > 0$ let $\varepsilon := \frac{1}{\sigma}\|\mu_1 - \mu_2\|_2$. Then*

$$T(\mu_1 + \sigma N(0, \mathbb{I}_p), \mu_2 + \sigma N(0, \mathbb{I}_p)) \equiv T(N(0, 1), N(\varepsilon, 1)), \tag{55}$$

Intuition and importance. The proof of this lemma is immediate from the explicit description of $f_{G,\varepsilon}(\alpha) = \Phi(\Phi^{-1}(1 - \alpha) - \varepsilon)$ where $\Phi(\cdot)$ is the cdf of a standard Gaussian variable $\Phi(t) = \mathbb{P}[N(0, 1) \leq t]$ for all $t \in \mathbb{R}$. It also requires applying the Neyman-Pearson lemma or the likelihood ratio test Polyanskiy & Wu (2025). But, the conclusions that they imply are extremely powerful.

Monotonicity. The *monotonicity* condition (52) reduces an apriori difficult functional comparison between two functions $f$ and $g$ at uncountably many points to a comparison of just one parameter $\varepsilon$.

Closure under suprema. The closure under suprema property (53) essentially says two very important things. First, it makes it easy to identify what the suprema of an apriori arbitrary collection of functions $\{f_i\}_i$ is (in fact explicitly). Second, the limiting object is a function of the same kind: it is again a Gaussian trade-off function with a different choice of parameter $\varepsilon$.

Symmetry. The symmetry property (54) (requires Neyman-Pearson lemma) is very interesting because the definition of a trade-off function $T(P, Q)$ 1 is asymmetric in general, between its first and second arguments. But, for a pair of shifted isotropic Gaussians, they match because of the spherical symmetry (orthogonal invariance) of the standard Gaussian density in any dimension.

Dimension freeness. The dimension freeness property (55) (requires Neyman-Pearson lemma) makes the case for Gaussian certifiability in high dimensions stronger than any other notion of certifiability. This is because, almost all the results of classical statistics that are true in low dimensions, fail to hold in high dimensions, because many of the quantities involved in controlling the errors are dimension dependent and blows up when $p \uparrow \infty$. It is often the case that finding a dimension-free quantity or even an inequality that *tensorizes* [15] help us resolve high dimensional issues.

The second preliminary work we will be using in our proof is about the $\ell_2$ difference of the a single step Newton approximation. Define the event $F$ in the following way:

$$F = \left\{ \max_{|\mathcal{M}| \leq m} \|\hat{\boldsymbol{\beta}}^{(1)}_{\backslash \mathcal{M}} - \hat{\boldsymbol{\beta}}_{\backslash \mathcal{M}}\|_2 \geq C_1(n)\sqrt{\frac{C_2(n)m^3}{2\lambda \nu n}} \right\}. \tag{56}$$

Now, Theorem 3.4 of Zou et al. (2025) in the notation and setting of this paper becomes:

**Theorem 7.** *Zou et al. (2025) If the event $F$ is as defined in* (56)*, then*

$$\mathbb{P}(F) \leq \phi_n,$$

*where $\phi_n := nq_n^{(y)} + 8n^{1-c} + ne^{-p/2} + 2e^{-p}$.*

Based on these two results we can now work on the proofs of our main results.

## G.2 PROOF OF THEOREM 2 IN THE GAUSSIAN CASE

Recall that we are supposed to show the following with the prescribed choice of $\phi = \phi_n$ as in (16).

$$\mathbb{P}\left[\inf_{|\mathcal{M}| \leq m} \min(T(\mathcal{P}_{\mathrm{re}}, \mathcal{P}_{\mathrm{un}})(\alpha), T(\mathcal{P}_{\mathrm{un}}, \mathcal{P}_{\mathrm{re}})(\alpha) \geq f_{G,\varepsilon}(\alpha) \quad \text{for all } \alpha \in [0, 1]\right] \geq 1 - \phi. \tag{57}$$

Let $\boldsymbol{b} \sim \sigma N(0, I_p)$ (independent of everthything else) with $\sigma = \frac{R}{\varepsilon}$. Then, observe that

$$\begin{aligned}
\mathcal{P}_{\mathrm{re}} &\stackrel{d}{=} \bar{A}(\hat{\boldsymbol{\beta}}_{\backslash \mathcal{M}}, \phi, T(\mathcal{D}_{\backslash \mathcal{M}}), \boldsymbol{b}) = \hat{\boldsymbol{\beta}}_{\backslash \mathcal{M}} + \boldsymbol{b}, \\
\mathcal{P}_{\mathrm{un}} &\stackrel{d}{=} \bar{A}(\hat{\boldsymbol{\beta}}, \mathcal{D}_{\mathcal{M}}, T(\mathcal{D}), \boldsymbol{b}) = \hat{\boldsymbol{\beta}}^N_{\mathcal{M}} + \boldsymbol{b},
\end{aligned} \tag{58}$$

Note that since the distributions $\mathcal{P}_{\mathrm{re}}$ and $\mathcal{P}_{\mathrm{un}}$ are conditional distributions given the dataset $\mathcal{D}$, we can use the dimension freeness property introduced in Lemma 6, and conclude that

$$T(\hat{\boldsymbol{\beta}}_{\backslash \mathcal{M}} + \boldsymbol{b}, \hat{\boldsymbol{\beta}}^N_{\mathcal{M}} + \boldsymbol{b}) = T(N(0, 1), N(\|\hat{\boldsymbol{\beta}}^N_{\mathcal{M}} - \hat{\boldsymbol{\beta}}_{\backslash \mathcal{M}}\|_2 / \sigma, 1)), \tag{59}$$

---

[15]We are referring dimension free Poincare and Logarithmic Sobolev inequalities of high-dimensional statistics that are extremely important in obtaining concentration bounds in high dimensions van Handel (2016).

$$T(\hat{\boldsymbol{\beta}}_{\mathcal{M}}^N + \boldsymbol{b}, \hat{\boldsymbol{\beta}}_{\backslash\mathcal{M}} + \boldsymbol{b}) = T(N(0,1), N(\|\hat{\boldsymbol{\beta}}_{\mathcal{M}}^N - \hat{\boldsymbol{\beta}}_{\backslash\mathcal{M}}\|_2/\sigma, 1)). \tag{60}$$

Using these two equations we conclude the following chain of equalities, all follow from Lemma 6.

$$\mathbb{P}\left[\inf_{|\mathcal{M}|\leq m} \min(T(\mathcal{P}_{\text{re}}, \mathcal{P}_{\text{un}})(\alpha), T(\mathcal{P}_{\text{un}}, \mathcal{P}_{\text{re}})(\alpha)) \geq f_{G,\epsilon}(\alpha) \quad \text{for all } \alpha \in [0,1]\right]$$

$$\overset{(a)}{=} \quad \mathbb{P}\left[\inf_{|\mathcal{M}|\leq m} T(N(0,1), N(\|\hat{\boldsymbol{\beta}}_{\mathcal{M}}^N - \hat{\boldsymbol{\beta}}_{\backslash\mathcal{M}}\|_2/\sigma, 1)) \geq f_{G,\epsilon}(\alpha) \quad \text{for all } \alpha \in [0,1]\right]$$

$$\overset{(b)}{=} \quad \mathbb{P}\left[T(N(0,1), N(\sup_{|\mathcal{M}|\leq m} \|\hat{\boldsymbol{\beta}}_{\mathcal{M}}^N - \hat{\boldsymbol{\beta}}_{\backslash\mathcal{M}}\|_2/\sigma, 1)) \geq f_{G,\epsilon}(\alpha) \quad \text{for all } \alpha \in [0,1]\right]$$

$$\overset{(c)}{=} \quad \mathbb{P}\left[\sup_{|\mathcal{M}|\leq m} \|\hat{\boldsymbol{\beta}}_{\mathcal{M}}^N - \hat{\boldsymbol{\beta}}_{\backslash\mathcal{M}}\|_2/\sigma \leq \epsilon\right]$$

$$\overset{(d)}{=} \quad \mathbb{P}\left[\sup_{|\mathcal{M}|\leq m} \|\hat{\boldsymbol{\beta}}_{\mathcal{M}}^N - \hat{\boldsymbol{\beta}}_{\backslash\mathcal{M}}\|_2/ \leq R\right]. \tag{61}$$

The above inequalities have been obtained through the following arguments:

- Equality (a): To obtain this equality we have used (59) and (60).
- Equality (b): We used the monotonicity property of $f_{G,\epsilon}(\alpha)$ mentioned in Lemma 6.
- Equality (c): We have used the monotonicity property of $f_{G,\epsilon}(\alpha)$ of Lemma 6 again.
- Equality (d): To obtain equality (d) we have used the fact that $\sigma = R/\varepsilon$.

So, it remains to prove the following with $\phi_n, R$ as prescribed in 2.

$$\mathbb{P}\left[\sup_{|\mathcal{M}|\leq m} \|\hat{\beta}_{\mathcal{M}}^N - \hat{\boldsymbol{\beta}}_{\backslash\mathcal{M}}\|_2 \leq R\right] \geq 1 - \phi. \tag{62}$$

Using Theorem 7 establishes this result.

Remark on Gaussianity assumption on features. Observe that our entire computation above was conditional on the data $\mathcal{D}$, and therefore do not depend on the Gaussianity of the features at all, except while invoking Theorem 7 to establish (62). We will address this issue later in Section H.

### G.3 Proof of Theorem 3 in the Gaussian case

Similar to Zou et al. (2025) we define the following two events:

$$E_5 := \left\{|\boldsymbol{x}_0^\top \hat{\beta}| \leq \sqrt{(\lambda\nu)^{-1} C_X \gamma_0 (1 + C_y^s(n) 2c \log(n))}\right\}$$

$$E_6 := \left\{\max_{|\mathcal{M}|\leq m} |\boldsymbol{x}_0^\top (\hat{\beta} - \hat{\boldsymbol{\beta}}_{\backslash\mathcal{M}})| \leq \frac{2C_X}{\lambda\nu}(\sqrt{\gamma_0} + 3)\sqrt{\frac{m(2m+c)}{p} \log(n)} \text{polylog}_4(n)\right\}.$$

We use the following result from Zou et al. (2025) to establish some bounds on the above events.

**Lemma 8.** *Then for any $\mathcal{D} \in F^c$, we have*

$$\mathbb{P}(E_5^c|\mathcal{D}) \leq \mathbb{P}\left(|\boldsymbol{x}_0^\top \hat{\beta}| \leq \sqrt{\frac{C_X}{p}}\|\hat{\beta}\|\sqrt{2c\log(n)}|\mathcal{D}\right) \leq 2n^{-c}$$

$$\mathbb{P}(E_6^c|\mathcal{D}) \leq \mathbb{P}\left(\exists|\mathcal{M}| \leq m, |\boldsymbol{x}_0^\top (\hat{\beta} - \hat{\boldsymbol{\beta}}_{\backslash\mathcal{M}})| > \sqrt{\frac{C_X}{p}}\|\hat{\beta} - \hat{\boldsymbol{\beta}}_{\backslash\mathcal{M}}\| \cdot 2\sqrt{\log(N) + c\log(n)}\Big|\mathcal{D}\right)$$

$$\leq 2n^{-c},$$

*where $N = \sum_{s=0}^m \binom{n}{s} \leq 2\binom{n}{m} \leq e^{2m\log(n)}$.*

Define the following two events

$$E_7 := \left\{ \forall |\mathcal{M}| \le m, |\boldsymbol{x}_0^\top (\tilde{\boldsymbol{\beta}}_{\backslash \mathcal{M}}^{(t)} - \hat{\boldsymbol{\beta}}_{\backslash \mathcal{M}} + \boldsymbol{b})| \right.$$
$$\left. \le 2\sqrt{C_X} \left( \frac{2}{\varepsilon} + \frac{1}{\sqrt{p}} \right) R_{t,n} \cdot \sqrt{(2m+c)\log(n)} \right\},$$

$$E_8 := \left\{ \|\boldsymbol{b}\| \le \frac{2R_{t,n}\sqrt{2p}}{\varepsilon}, |y_0| \le C_y(n) \right\}.$$

Define $\gamma := 2\sqrt{C_X} \left( \frac{2}{\varepsilon} + \frac{1}{\sqrt{p}} \right) R_{t,n} \cdot \sqrt{(2m+c)\log(n)}$. Note that using the union bound we have

$$\mathbb{P}(E_7^c \cap F^c \cap E_8 | \mathcal{D}, \boldsymbol{b}) \le e^{2m \log n} \mathbb{P} \left\{ |\boldsymbol{x}_0^\top (\tilde{\boldsymbol{\beta}}_{\backslash \mathcal{M}}^{(t)} - \hat{\boldsymbol{\beta}}_{\backslash \mathcal{M}} + \boldsymbol{b})| \ge \gamma | \mathcal{D}, \boldsymbol{b} \right\}$$

$$\le 2e^{2m \log n} e^{-\frac{\gamma^2}{2\|\tilde{\boldsymbol{\beta}}_{\backslash \mathcal{M}} - \hat{\boldsymbol{\beta}}_{\backslash \mathcal{M}} + \boldsymbol{b}\|^2}}. \tag{63}$$

By definition of $F$ from (56), under $F^c \cap E_8$,

$$\|\tilde{\boldsymbol{\beta}}_{\backslash \mathcal{M}} - \hat{\boldsymbol{\beta}}_{\backslash \mathcal{M}} + \boldsymbol{b}\| \le \|\tilde{\boldsymbol{\beta}}_{\backslash \mathcal{M}} - \hat{\boldsymbol{\beta}}_{\backslash \mathcal{M}}\| + \|\boldsymbol{b}\| \le \left( \frac{2\sqrt{2p}}{\varepsilon} + 1 \right) R_{t,n}.$$

Then, following the proof of Theorem 3.2 in Zou et al. (2025) we would have

$$\mathbb{P}((E_7')^c \cap F^c \cap E_8') \le 2n^{-c}.$$

Under $(\cap_{i=5}^6 E_i) \cap E_7 \cap E_8 \cap F^c$, w.p.$\ge 1 - (n+1)q_n^{(y)} - 14n^{1-c} - ne^{-p/2} - 2e^{-p} - e^{-(1-\log(2))p}$, $\forall a \in [0,1]$,

$$|\boldsymbol{x}_0^\top [a(\tilde{\boldsymbol{\beta}}_{\backslash \mathcal{M}} + \boldsymbol{b}) + (1-a)\hat{\boldsymbol{\beta}}_{\backslash \mathcal{M}}]|$$
$$\le |\boldsymbol{x}_0^\top \hat{\boldsymbol{\beta}}_{\backslash \mathcal{M}}| + a|\boldsymbol{x}_0^\top (\tilde{\boldsymbol{\beta}}_{\backslash \mathcal{M}}^{(t)} - \hat{\boldsymbol{\beta}}_{\backslash \mathcal{M}} + \boldsymbol{b})|$$
$$\le \text{polylog}_1(n) + 2\sqrt{C_X} \left( \frac{2\sqrt{2}}{\varepsilon} + \frac{1}{\sqrt{p}} \right) R \cdot \sqrt{(2m+c)\log(n)}$$
$$\le \text{polylog}_2(n),$$

provided $R_{t,n} = o\left( \frac{\varepsilon}{\sqrt{m}\text{polylog}(n)} \right)$ so that the second term is $O(\text{polylog}(n))$. So we have

$$|\dot{\ell}_0((\tilde{\boldsymbol{\beta}}_{\backslash \mathcal{M}} + \boldsymbol{b}) + (1-a)\hat{\boldsymbol{\beta}}_{\backslash \mathcal{M}})|$$
$$\le 1 + C_y^s(n) + |\boldsymbol{x}_0^\top [a(\tilde{\boldsymbol{\beta}}_{\backslash \mathcal{M}} + \boldsymbol{b}) + (1-a)\hat{\boldsymbol{\beta}}_{\backslash \mathcal{M}}]|^s$$
$$\le \text{polylog}_3(n),$$

and

$$|\bar{\dot{\ell}}_0| = \left| \int_0^1 \dot{\ell}_0((\tilde{\boldsymbol{\beta}}_{\backslash \mathcal{M}}^{(t)} + \boldsymbol{b}) + (1-a)\hat{\boldsymbol{\beta}}_{\backslash \mathcal{M}})da \right| \le \text{polylog}_3(n).$$

Finally we have that, under $(\cap_{i=5}^6 E_i) \cap E_7' \cap E_8' \cap F^c$,

$$\text{GED}(\tilde{\boldsymbol{\beta}}_{\backslash \mathcal{M}}, \hat{\boldsymbol{\beta}}_{\backslash \mathcal{M}})$$
$$= |\ell_0(\tilde{\boldsymbol{\beta}}_{\backslash \mathcal{M}} + \boldsymbol{b}) - \ell_0(\hat{\boldsymbol{\beta}}_{\backslash \mathcal{M}})|$$
$$\le |\bar{\dot{\ell}}_0| |\boldsymbol{x}_0^\top (\tilde{\boldsymbol{\beta}}_{\backslash \mathcal{M}} - \hat{\boldsymbol{\beta}}_{\backslash \mathcal{M}} + \boldsymbol{b})|$$
$$\le \text{polylog}_3(n) 2\sqrt{C_X} \left( \frac{2}{\varepsilon} + \frac{1}{\sqrt{p}} \right) R \cdot \sqrt{(2m+c)\log(n)}$$
$$\le \text{polylog}_4(n) 2\sqrt{C_X} \left( \frac{2}{\varepsilon} + \frac{1}{\sqrt{p}} \right) R \cdot \sqrt{(2m+c)\log(n)}.$$

Plugging in the value of $R$ from Theorem 2 finishes the proof.

Remark on Gaussianity assumption on features. Observe that the entire computation above was reduced to getting the correct orderwise bound on the lablled set of the events $E_5, E_6, E_7, E_8, E_7', E_8'$ above and event $F$ 56. Now, all the labelled events $E_5, E_6, E_7, E_8, E_7', E_8'$ involves a fresh feature data $\boldsymbol{x}_0$ essentially in its linear component $\boldsymbol{x}_0^T v$ for some vector $v$. Now, it is well known Vershynin (2018) that for sub-Gaussian vector $\boldsymbol{x}_0 \sim \mathcal{SG}(0, \Sigma)$, the linear projection $\boldsymbol{x}_0^T v$ has the same (orderwise) Gaussian tail beahvior as if $\boldsymbol{x}_0$ was $\mathcal{N}(0, \Sigma)$, since in all these cases $v$ is independent of the fresh sample $\boldsymbol{x}_0$ and therefore, the same orderwise bounds of $E_5, E_6, E_7, E_8, E_7', E_8'$ continue to hold except while invoking Theorem 7 to establish (62) or equivalently the tail of 56. We will address this issue next.

## H    An extension of the proof in the case of sub-Gaussian features

Given the arguments already presented in the proofs of Theorems 2 and 3 in section G, it is enough to establish that the Theorem 7 of Zou et al. (2025) can be extended from $x \sim \mathcal{N}(0, \boldsymbol{\Sigma})$ to $x \sim \mathcal{SG}(0, \boldsymbol{\Sigma})$ with the same assumptions on $\boldsymbol{\Sigma}$. We now describe why is this possible. An inspection of the proof of Theorem 7 of Zou et al. (2025) reveals that their entire proof never uses the following potentially Gaussian facts that fails when one goes from Gaussian to sub-Gaussian vectors. 1) It never uses orthogonal invariance of the feature vectors in the proof. 2) It never uses Gaussian conditioning technique: $(X|Y$ is Gaussian if $(X, Y)$ is jointly Gaussian). Rather it relies on standard concentration bounds collected in Lemma B.7 –B.11 in the appendix of Zou et al. (2025)[arxiv version May 12] which can be all be generalized from Gaussian to sub-Gaussian vectors with exact same (mean, covariance) via a standard computation of its moment-generating functions.[16] Morevoer, their proofs uses the well-known fact that an $m$ by $n$ matrix with centered, normalized independent columns has a bounded operator norm (and its Wishart version) Vershynin (2018)[Exercise 4.43], be it sub-Gaussian or Gaussian. More precisely, each of the results in has an analogous sub-Gaussian extension (with potentially worse constants, which we disregard anyway) and therefore extends verbatim.

## I    Additional Numerical Experiments

In this section, we provide results from additional numerical experiments under the ridge penalized $r(\boldsymbol{\beta}) = \|\boldsymbol{\beta}\|^2$ logistic regression model $\ell(y|\boldsymbol{x}^\top \boldsymbol{\beta})$, with random features $\boldsymbol{x}_i \sim N(\boldsymbol{0}, \frac{1}{n}\mathbb{I}_p)$ and true parameter $\boldsymbol{\beta}_* \sim N(\boldsymbol{0}, \mathbb{I}_p)$, penalty parameter $\lambda = 0.5$ as in the main paper 5. This further supports our theoretical results. These were not kept in the main paper for lack of space.

### I.1    Additional plots for n=p

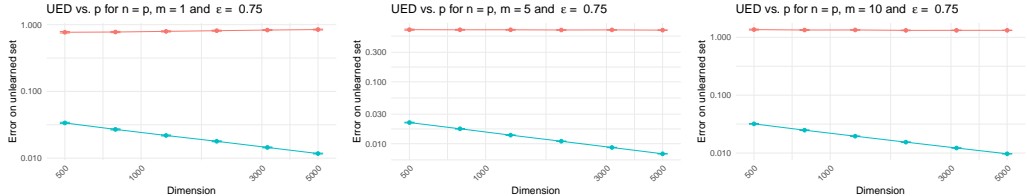

Figure 6:  Comparison on unlearned data: mean UED (with 3 SD error bars) on the unlearned set across $p$ (both in $\log$ scale) for Laplace (in red) vs. Gaussian noise (in cyan). We set $\lambda = 0.5$.

We plot UED: the error divergence of the unlearned estimator on unlearned data, with $n = p = 1255$, as we vary $\varepsilon$ on X axis across 6 different values, and take $m = 1, 5, 10$. The mean UED over 200 replications with standard error bars are presented in Figure 7.

Now, we fix $\varepsilon = 0.75$ and study the change in UED as the unlearning size $m \uparrow$, varying $m$ across 6 values spaced equally between 5 and 50 on the logarithmic scale. The mean UED over 200 replications along with standard error bars are presented in Figure 8.

---

[16]A sub-Gaussian vector $x \sim \mathcal{SG}(0, \boldsymbol{\Sigma})$ by definition satisfies $\mathbb{E}[e^{v^T x}] \leq e^{\frac{1}{2}v^T \boldsymbol{\Sigma} v} \ \forall v \in \mathbb{R}^p$ Vershynin (2018).

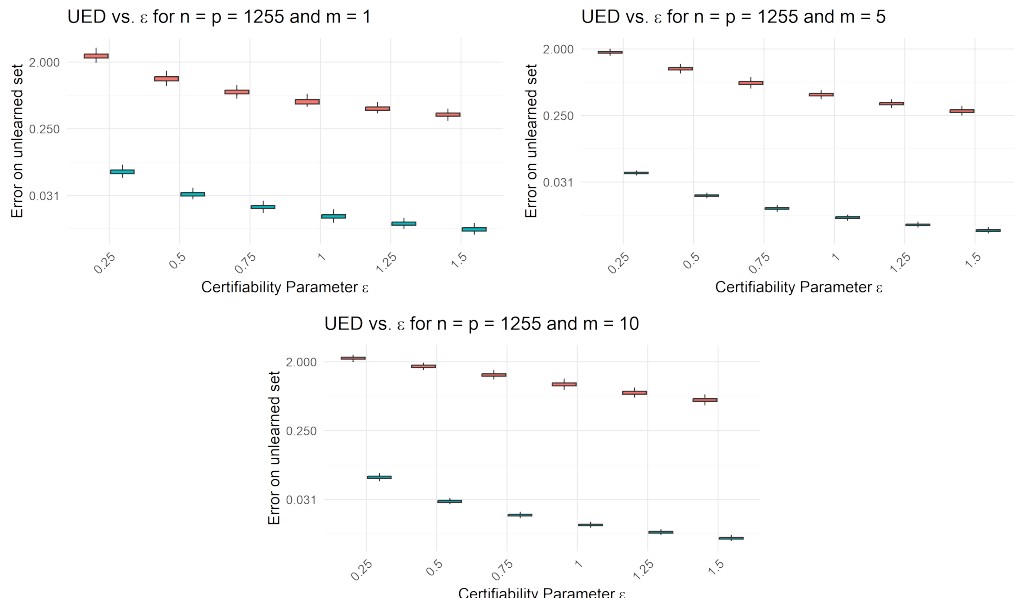

Figure 7: Comparison of the difference in negative log likelihood on the requested removal set (plotted in $\log$ scale) among the retrained estimator, with unlearned estimator with Laplace noise (in red) and Gaussian noise (in cyan). The left, middle, and right figures plot unlearning for removal sizes $m = 1$, $m = 5$, and $m = 10$ respectively. Here $n = p = 1255$.

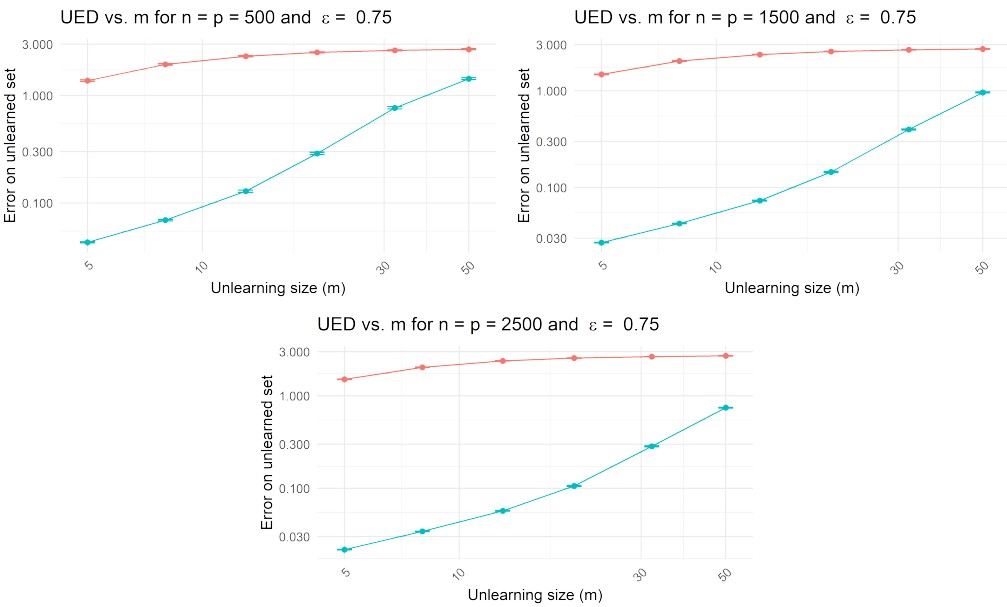

Figure 8: Comparison of UED (plotted in $\log$ scale) across the unlearning size $m$ (plotted in $\log$ scale) for Laplace noise (in red) vs. Gaussian noise (in cyan).

I.2    PLOTS FOR N=P/2

In this subsection we provide results for the analogous experiments in the high dimensional regime where $n = p/2$. It is evident from our results that the conclusions remain the same.

First set (meaning of figures 7 and 8). We plot GED as we vary $p$ on the X axis and $n = p/2$ on a logarithmic scale, while keeping $m$ fixed at $1, 5, 10$ across the three plots. We find the mean GED values across 200 replications decrease with $p$ for Gaussian noise, but stay more or less the same for Laplace noise. Results are plotted in Figure 9 and Figure 10.

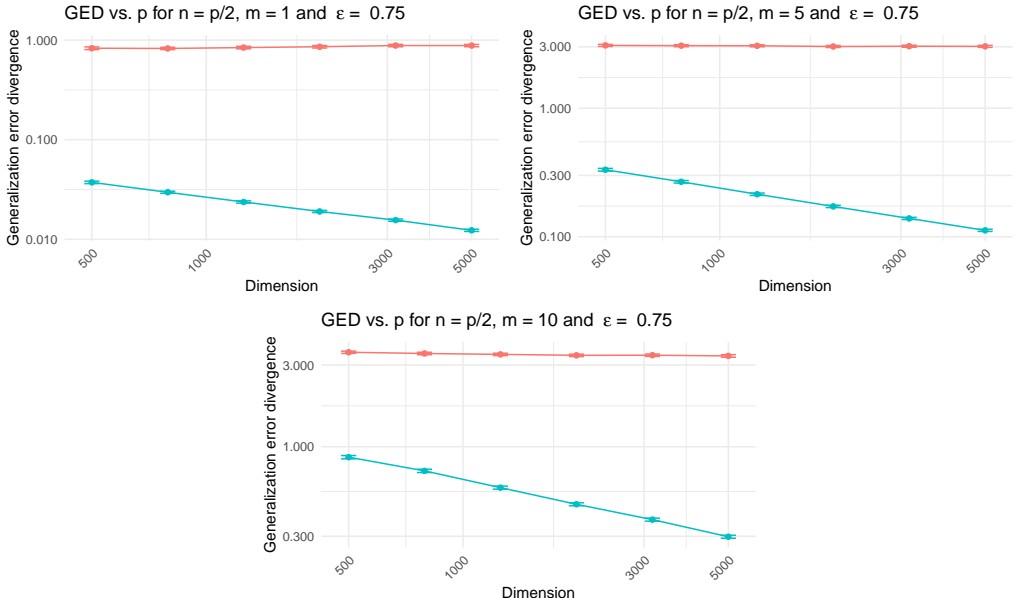

Figure 9: Comparison of unlearned estimators on new test data: mean GED (with 3 SD error bars) across the dimension $p$ (both in $\log$ scale) for Laplace (in red) vs. Gaussian (in cyan). We set $\lambda = 0.2$.

Second set (meaning of figures 9 and 10). Now, we fix $2n = p = 1255$ and examine GED as we vary $\varepsilon$ on the $X$ axis, while $m$ varies between $m = 1$ (left), $m = 5$ (middle), and $m = 10$ (right). Figure 11 and Figure 12 report boxplots of GED in this setting across 200 Monte Carlo replications.

Conclusions from figures 9 and 10. The performance of Gaussian perturbed estimator becomes increasingly closer to that of the ideal retrained estimator as $\varepsilon \uparrow$ on the X axis. The error for the Laplace perturbed estimator remains above its Gaussian counterpart, uniformly over $\varepsilon$. Similar behavior is reflected on the unlearned subset as well. Results are in Figure 11 and Figure 12.

Final set (meaning of figures 11 and 12). In our final set of experiments, we fix the certifiability parameter $\varepsilon = 0.75$, and examine GED as the unlearning size $m$ varies on the X axis, while $n, p$ varies between $2n = p = 500$ (left), $2n = p = 1500$ (middle), and $2n = p = 2500$ (right). We vary $m$ across 6 values from 5 to 50 equally spaced on a logarithmic scale. Figure 13 plots the average GED values (averaged across 200 replications) against dimension $m$, transforming both X and Y axes to be on a logarithmic scale.

Conclusions from figures 11 and 12. For all three plots, as $m$ increases, unlearning accuracy as reflected in GED, worsens for both the estimators. Results are in Figure 13 and Figure 14.

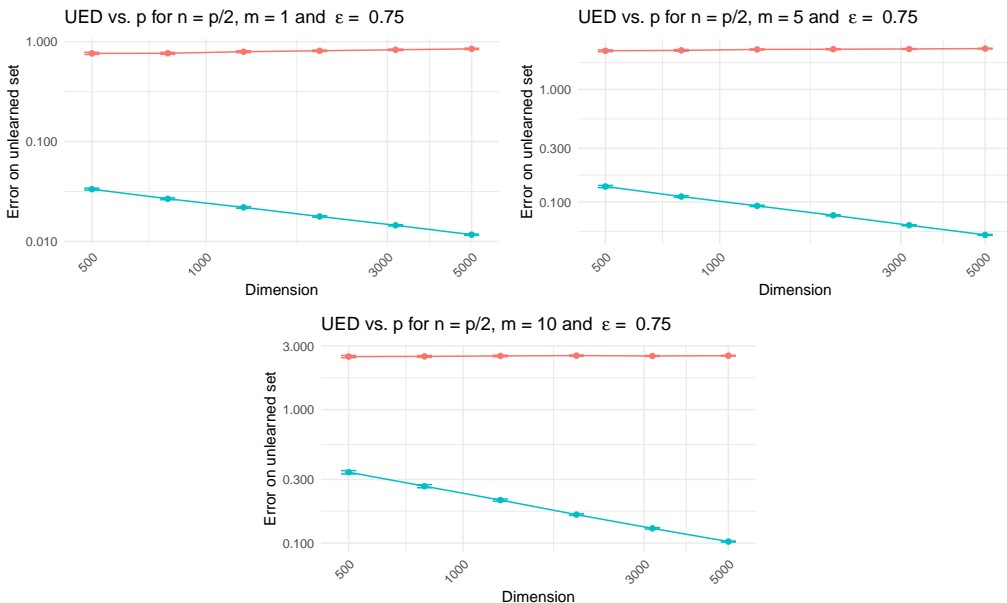

Figure 10: Comparison on unlearned data: mean UED (with 3 SD error bars) on the unlearned set across $p$ (both in $\log$ scale) for Laplace (in red) vs. Gaussian noise (in cyan). We set $\lambda = 0.2$.

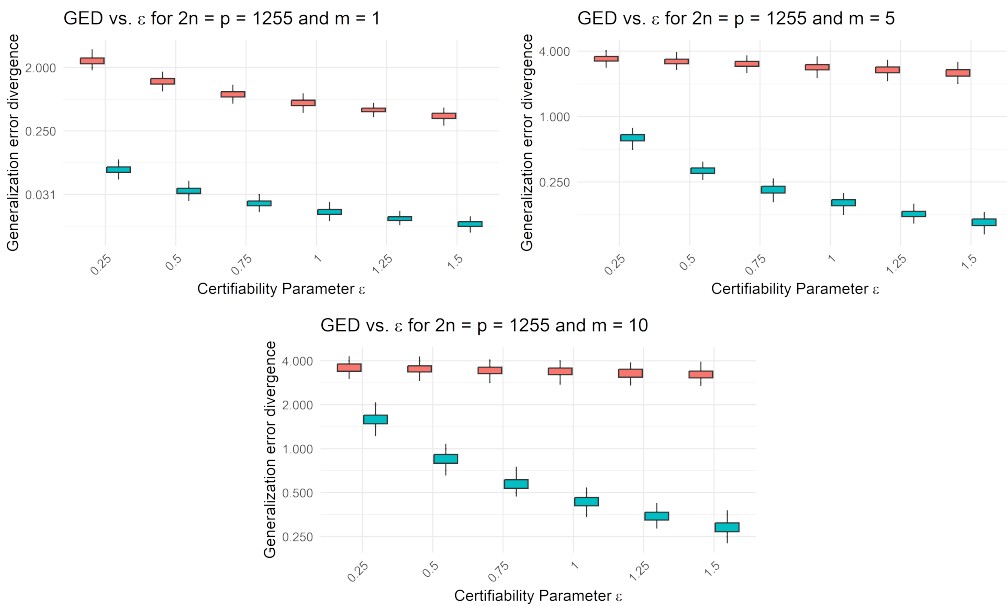

Figure 11: Comparison of GED (plotted in $\log$ scale) across different values of $\varepsilon$ for Laplace noise (in red) vs. Gaussian noise (in cyan). Here $2n = p = 1255$. We set $\lambda = 0.2$.

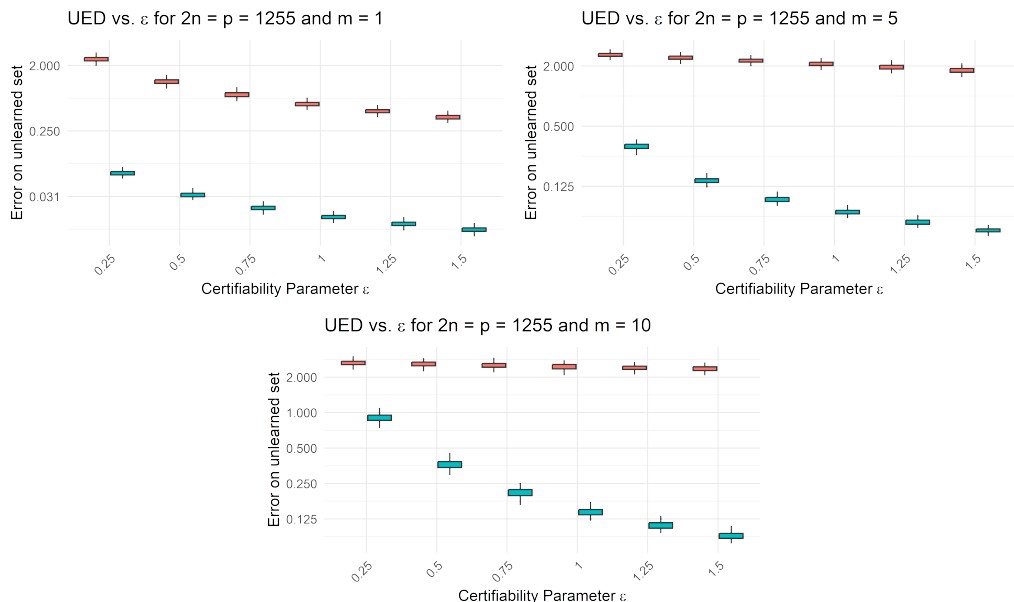

Figure 12: Comparison of the difference in negative log likelihood on the requested removal set (plotted in $\log$ scale) among the retrained estimator, with unlearned estimator with Laplace noise (in red) and Gaussian noise (in cyan). The left, middle, and right figures plot unlearning for removal sizes $m = 1$, $m = 5$, and $m = 10$ respectively. Here $2n = p = 1255$.

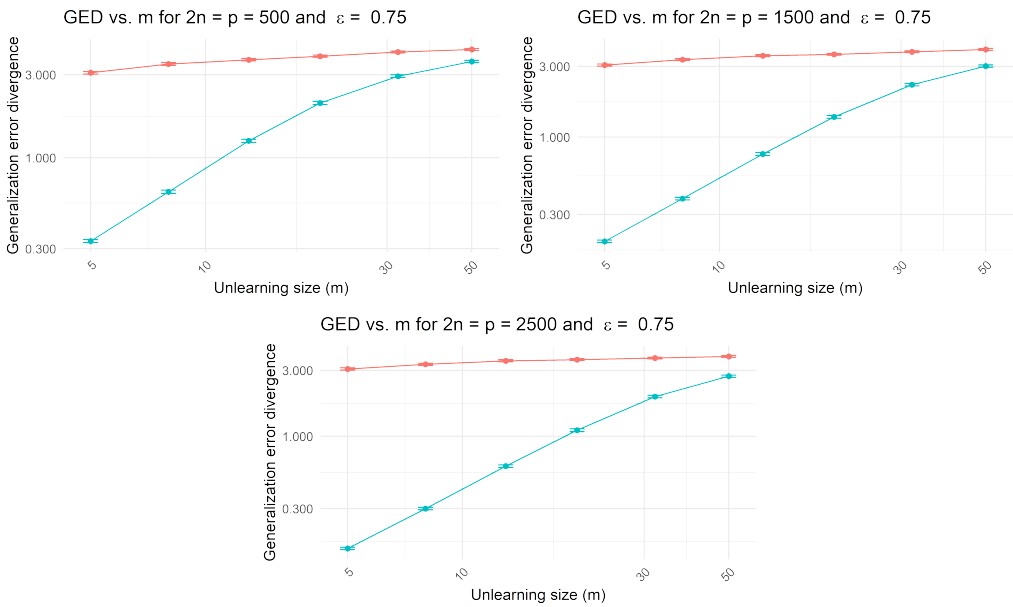

Figure 13: Comparison of mean GED (with 3 SD error bars) across the unlearning size $m$ (both in $\log$ scale) for Laplace noise (in red) vs. Gaussian noise (in cyan). We set $\lambda = 0.2$.

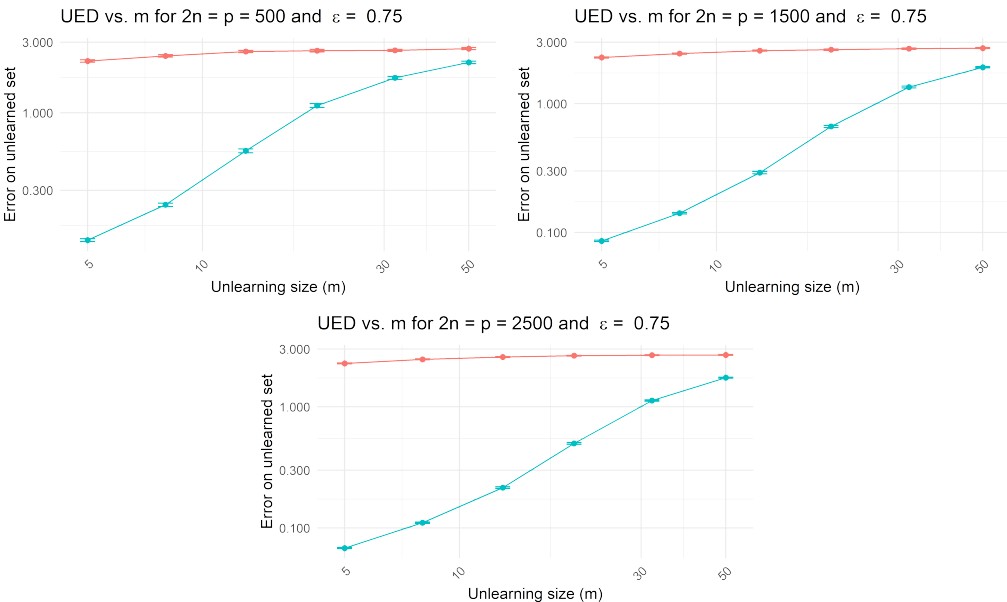

Figure 14: Comparison of UED (plotted in $\log$ scale) across the unlearning size $m$ (plotted in $\log$ scale) for Laplace noise (in red) vs. Gaussian noise (in cyan).

