# OpenReview forum: "Gaussian certified unlearning in high dimensions: A hypothesis testing approach"
_ICLR.cc/2026/Conference — ICLR 2026 Oral_

### Official Review · Reviewer_WvFT · 2025-11-01

**Soundness:** 3
**Presentation:** 3
**Contribution:** 3
**Rating:** 6
**Confidence:** 3

**Summary:**

The paper introduces ε-Gaussian certifiability (GPAR)—a new theoretical notion for analyzing machine unlearning in high-dimensional regimes (p ~ n). It reformulates unlearning guarantees via hypothesis testing and Gaussian trade-off functions, showing that a single Newton step with Gaussian noise suffices to ensure both privacy and accuracy. This contrasts with prior work (notably Zou et al., 2025), which required at least two steps under ε-certifiability. The authors also provide proofs of convergence, theoretical bounds for Generalization Error Divergence (GED), and simulation results validating the high-dimensional behavior.

**Strengths:**

1. Introducing ε-Gaussian certifiability as a canonical high-dimensional analogue to ε-certifiability is a meaningful contribution, bridging DP-inspired hypothesis testing with unlearning theory.

2.  The paper convincingly argues that existing assumptions—especially Ω(1) strong convexity and O(1) smoothness—fail in proportional regimes, motivating a new approach.

3. The contrast with Allouah et al. (2025), Sekhari et al. (2021), and Zou et al. (2025) is detailed—highlighting where previous frameworks break down.

**Weaknesses:**

1. Experiments are confined to synthetic logistic/ridge regression under controlled Gaussian features. There are no tests on real or non-linear models, limiting practical validation.

2.  While the framework is theoretically extendable, the assumptions (convexity, separability, sub-Gaussian features) exclude modern deep models. Discussion on extending to non-convex objectives (e.g., neural networks) remains speculative.

3. Although one-step Newton is theoretically efficient, computational overhead in large-scale settings (Hessian computation, inversion) is not discussed.

**Questions:**

1. How sensitive is the ε-GPAR bound to mis-specification of noise variance? Could small deviations in σ break certifiability guarantees?

2. The theorems assume a single batch of deletions of size m. What happens if deletions arrive sequentially? Is the GED bound sub-additive, or does error accumulate over multiple unlearning rounds?

3. The constants C₁(n), C₂(n) are said to be polylog(n), but they are never specified. How large are they empirically, and do they affect practical implementability?

4. The framework assumes exact access to the Hessian—what happens when it is estimated or approximated in large-scale models? For large $p$, forming and inverting $p \times p$ Hessians is infeasible. Can similar theoretical results be shown for quasi-Newton or stochastic approximations?

---

> ### Author Response · Authors · 2025-11-21
> **Thank you for your positive and careful evaluation. We have addressed your questions, comments in the revised draft, as detailed in the points below. We request that you take a look at the newly added texts in the revised draft for more details.**
>
> Q1.  Sensitivity of ε-GPAR bound to $\sigma$ (see Section F.3 of the appendix of the revised draft (RD))
>
> $(\phi, \varepsilon)$ GPAR (Thm 2) holds for any user-chosen $\sigma$ above the critical threshold  $\sigma=$ $\frac{R}{\varepsilon}$, with R as in Theorem 2, as is clear from its proof. Thus, any positive deviation of $\sigma$ above this choice does not break certifiability of Thm 2. However, with increasing $\sigma$, the GED bound in Theorem 3 worsens, but only linearly in $\sigma$ (see the dominant term of the error involving $\frac{1}{\varepsilon}$ in Theorem 3).
>
> Q2. On sequential deletions (Introduction, conclusion of RD)
>
> Yes,  the GED error can be written as a sum over the GED errors over the multiple rounds of deletion requests, and therefore, the errors do accumulate. But, depending on what information about the original Hessian we keep over time, or only keep the most recent Hessian evaluation, this is an interesting question of practical importance for future study. We have added a brief discussion on how our batch $(\phi,\varepsilon)$-GPAR can serve as a foundation for future online unlearning in high dimensions.
>
> Q3. On constants $C_1(n), C_2(n)$ (Section 4.3, F.3 of RD)
>
> Please see bullet point 6 in section F.3 of RD, where we give two calibration rules for choosing $\sigma$
> in practice without actually requiring  $C_1(n),C_2(n)$. Our main theorems are stated with unspecified constants to simultaneously cover a broad class of losses $\ell, r$ that practitioners may wish to use. Once $\ell$ and $r$ are fixed and estimates for the relevant data parameters are computed, these constants become explicit (see Appendix F.3 of RD for exact expressions, and two examples: linear, logistic regression).
>
> We agree the constants are not optimal up to (1+o(1)) factors. Obtaining sharp constants usually requires delicate asymptotics, whereas our focus was on non-asymptotic guarantees for fixed but large $(n,p)$ that are directly useful in practice, while still covering a substantially wider range of $(\ell,r)$ than prior work (Guo et al., Sekhari et al., Allouah et al.).
>
> Q4. On quasi-Newton procedures? (Conclusion of RD)
>
> We agree that full Hessian computation is expensive in very high dimensions. This motivated us to examine how few Newton steps are needed, and we found that even in high-dimensional settings, a single iteration is sufficient. To test scalability, we have included experiments with large $p=10000$ and $n=p/10$ in RD. We are running even larger $p$ experiments and plan to add $p=25000$ to the appendix of RD. For extremely large-scale models, several works (Koh-Liang 2017, Park et al.) show that related quantities, such as influence functions, formally require the inverse Hessian, which can be computed efficiently via approximations whose accuracy in high dimensions is still not well understood. Investigating these approximations is an important direction for future work.
>
> W1 Synthetic Experiments (Section 5, and I of RD)
>
> We have added more experiments in RD. Figure 6 of RD now reports results on MNIST (low $p$) and IMDb (high $p$) data. On MNIST, the three unlearning notions perform comparably, whereas on IMDb, Gaussian unlearning is clearly superior.
>
> Our primary goal, however, is to develop a rigorous high-dimensional theory of GPAR with relaxed losses $(\ell, r)$, which, to our knowledge, is missing in the literature. So, we focused our experiments on linear ridge-regularised ERM with Gaussian features, exactly matching our theoretical setting.
>
>
> W2a On sub-Gaussian features. (Conclusion, section D of RD)
>
> We agree that assuming sub-Gaussian (SG) features is standard in high-dimensional statistics. However, (1) In practice, features are stored with finite precision, so entries lie in bounded ranges and are often well-modeled as SG. (2) Some control over the tails of the feature is essential, while one can relax SG to slightly more general Orlicz spaces (sub-Weibull in Kuchibhotla and Chakrabortty (2022)), without such assumptions, no efficient, approximate unlearning method can guarantee both accuracy and privacy for arbitrary data-point removals. Under subGaussianity, our work is (to our knowledge) the first to show that a single Newton step suffices to achieve both GPAR and vanishing GED in high dimensions.
>
> W2b. On convex and separable loss. (Conclusion, Appendix D of RD)
>
> We agree that extending our novel GPAR framework to highly non-convex models is an important direction. However, we focus on convex ERM because it offers a clean setting to rigorously track the effect of a Newton update. Even in this convex high-dimensional regime, our result is, to our knowledge, the first to show that a single noisy Newton step simultaneously yields GPAR and accuracy, under weaker conditions on $(\ell, r)$ than prior works (Sekhari et al., Guo et al., and Allouah et al.) whose analysis fails in $p\sim n$ setting. Extending these beyond convex models is a challenging but compelling next step.

---

> ### Author Response · Authors · 2025-12-03
> **Summary of Responses and Revisions for Reviewer WvFT**
>
> We thank the reviewer for their careful and constructive evaluation. Now we summarize the main revisions made during the rebuttal period, including detailed explanations added in the revised draft to address the reviewer's suggestions. In summary, we have   **carefully and thoroughly addressed all the suggestions made by the reviewer**.
>
> **Major points**
>
> 1) **We added real data experiments on MNIST and IMDb datasets** in section 5 and appendix I of the revised draft and showed the superiority of Gaussian unlearning over previous unlearning mechanisms in high-dimensional datasets arising in practice.
>
> 2) **We resolved the practical concern of noise calibration** and provided details of two practical rules to choose the noise level in bullet point 6 in section F.3 of the appendix of the revised draft, as well as the computation of the constants $C_1(n), C_2(n)$ for a given problem.
>
> Responses to reviewer's questions
>
> Q1.  **Sensitivity of ε-GPAR bound to $\sigma$** (see Section F.3 of the appendix of the revised draft)
>
> Yes, $(\phi, \varepsilon)$ GPAR (Thm 2) holds for any user-chosen $\sigma$ above the critical threshold  $\sigma=$ $\frac{R}{\varepsilon}$, with R as in Theorem 2, and small deviations do not break certifiability guarantees.
>
> Q2. **On sequential deletions** (Introduction, section 6, Conclusion of the revised draft)
>
> Yes,  the GED error can be written as a sum over the GED errors over the multiple rounds of deletion requests, and we have added a discussion on how our batch Gaussian certifiability framework can serve as a foundation for future online unlearning in high dimensions with sequential or continuous deletion requests.
>
> Q3. **On practical implementability and computing constants $C_1(n), C_2(n)$** (section 4.3, appendix F.3 of revised draft)
>
> Yes, we give two calibration rules for choosing the noise level $\sigma$ in practice without actually requiring  $C_1(n), C_2(n)$. Moreover, once the losses $\ell$ and $r$ are fixed and estimates for the relevant data parameters are computed, these constants $C_1(n), C_2(n)$ can be explicitly computed.
>
> Q4. **On quasi-Newton procedures** (Conclusion of the revised draft)
>
> Yes, we added a discussion on how our Newton-based unlearning method provides a basis for scalable approximate Newton methods. Moreover, we increased the dimensions of the experiments to p =10000 in the revised version.
>
> Responses to identified Weaknesses
>
> W1. **Synthetic Experiments** (section 5, and appendix I of revised draft)
>
> Yes, we added real data experiments on MNIST and IMDb datasets, as well as experiments with larger dimension p =10K, and showed the superiority of Gaussian unlearning over previous unlearning mechanisms in high-dimensional datasets arising in practice.
>
> W2a. **On sub-Gaussian features** (see Conclusion, and section D of the appendix of the revised draft)
>
> Yes, our theoretical bounds as stated hold for sub-Gaussian features. However, in practice, features are stored with finite precision, so entries lie in bounded ranges and are often well-modeled as sub-Gaussian. Moreover, some control over the tails of the feature is essential, and heavy-tailed high-dimensional features can significantly hurt any efficient, approximate unlearning method.
>
> W2b. **On convex loss functions**. (see Conclusion and appendix D of the revised draft)
>
> Yes, extending our novel Gaussian certifiability framework to highly non-convex models (large neural networks) is an important direction. However,  in this convex high-dimensional regime, our result is, to our knowledge, the first to show that a single noisy Newton step simultaneously yields unlearning and accuracy.
>
> W3. **On the Hessian computation and scalability of the Newton method** (section 6, Conclusion of the revised draft)
>
> Yes, we added a discussion on how our Newton-based unlearning method provides a basis for scalable approximate Newton methods. Moreover, we increased the dimensions of the experiments to p =10000 in the revised version.

---

### Official Review · Reviewer_qmUk · 2025-11-01

**Soundness:** 3
**Presentation:** 2
**Contribution:** 3
**Rating:** 6
**Confidence:** 4

**Summary:**

This paper introduces ε-Gaussian certifiability, a novel and robust framework tailored for high-dimensional machine unlearning. The authors demonstrate that a single noisy Newton step suffices to achieve both privacy and statistical accuracy under this new notion, contrasting with prior work that required at least two steps. Theoretical results are supported by convincing simulations.

**Strengths:**

1. The paper proposes a novel, high-dimensional–friendly notion of certified unlearning (ε-Gaussian certifiability) that tightly captures Gaussian noise mechanisms and subsumes prior privacy definitions.
2. The theoretical analysis is rigorous. The authors theoretically prove that a single Gaussian-Newton step simultaneously achieves (φ,ε)-GPAR privacy and vanishing generalization error in the proportional p∼n regime under relaxed assumptions.

**Weaknesses:**

1. Some writing issues:
   - In the Abstract, the notation $ p \ll n $ should be clarified by specifying what $ p $ and $ n $ represent, respectively.
   - In the Introduction section, the reviewers suggests first discussing the shortcomings or failures of existing works before presenting the performance and contributions of the proposed method.
   -  Using boldface followed by a colon for the first sentence of every paragraph throughout the paper harms the logical flow and readability, making it less reader-friendly.
2. Some typos:
   - $\mathcal D$ shoud be $\mathcal D_{\backslash {\mathcal M}}$ on Line 188.
   - The symbols $\epsilon$ and $\delta$ are not defined in Eq.(8).
   - $ p\uparrow \infty $ on Line 254.
   - The symbol $ r $ is reused in Eq.(16). Earlier in the text, it denotes the regularization term.
3. The paper lacks discussion on online unlearning (i.e., handling continuous unlearning requests).
4. The reviewers suggets that using a table to compare the results of this work with those of existing methods would be clearer and more effective.

**Questions:**

Could you provide experiments with larger values of $p$ (e.g., 10K or 100K)?

---

> ### Author Response · Authors · 2025-11-21
> **Response to your questions and comments**
>
> Thank you for your positive and careful evaluation. We have addressed your questions, comments in the revised draft, as detailed in the points below. We request that you take a look at the newly added texts in the revised draft for more details.
>
>
> Q1. Experiments with larger values of p
>
> Thanks for the suggestion. We have included experiments with large $p=10000$ and $n=p/10$ in the currently revised draft. We are running even larger dimensional experiments currently and plan to add experiments with $p=25000$ to the appendix of the revised version.
>
>
> W1. Some writing issues.
>
> We appreciate the reviewer’s comments on the exposition, and we have incorporated them in the revised draft. First, we have clarified the notation $p\ll n$ in the abstract. Second, we have switched the order of presentation in the introduction. Now failure of existing works comes before presenting our contributions. Finally, we have removed the `boldface followed by a colon’ in the entire text to make it reader-friendly.
>
> W2. Online unlearning.
>
> We thank the reviewer for bringing this important point up. In the revised draft, we have added a brief discussion (in the introduction, second paragraph, and in the conclusion of the paper) on how our batch $(\phi,\varepsilon)$-Gaussian certifiability framework can serve as a foundation for future online unlearning in high dimensions with sequential or continuous deletion requests.
>
> A simple inspection would reveal that once multiple rounds of deletion requests come (consider two-step deletions), using the triangle inequality, the GED error of Theorem 3 can be written as a sum over the GED errors over the deletion rounds, and therefore the errors do accumulate. But, depending on what information about the original Hessian we (the unlearning algorithm designer) keep over time, or only keep the most recent Hessian evaluation (because of certain constraints), this is an extremely interesting question of practical importance for future study.
>
> W3. Table for comparison.
>
> We thank the reviewer for this suggestion. In the revised draft, we have added a table in the introduction on how, on a high level, our contribution fits in comparison to previous works. First, there is the novel introduction of the notion of  $(\phi,\varepsilon)$-Gaussian certifiability, which is the optimally achievable (tightest) framework in high dimensions, in comparison to all such previous notions of certifiability. The second one is the relaxation of the loss functions within the Convex regime for high-dimensional applicability.

---

> ### Author Response · Authors · 2025-12-03
> **Summary of Responses and Revisions for Reviewer qmUk**
>
> We thank the reviewer for their careful and constructive evaluation. Now we summarize the main revisions made during the rebuttal period, including detailed explanations added in the revised draft to address the reviewer's suggestions. In summary, we have   **carefully and thoroughly addressed all the suggestions made by the reviewer**.
>
> **Major point**
>
> **We added real data experiments on MNIST and IMDb datasets, as well as experiments with larger dimension p =10K** in section 5 and appendix I of the revised draft, and showed the superiority of Gaussian unlearning over previous unlearning mechanisms in high-dimensional datasets arising in practice.
>
>
> Responses to reviewer's questions
>
> Q1. **Experiments with larger values of $p$** (section 5 and appendix I of the revised draft)
>
> Yes, we added experiments with real datasets such as IMDb and MNIST as well as experiments with larger dimension p =10K, and showed the superiority of Gaussian unlearning over previous unlearning mechanisms in high-dimensional datasets arising in practice.
>
> Responses to identified Weaknesses
>
> W1 **Some writing issues**
>
> Yes, we have addressed them all in the revised draft.
>
> W2. **Some Typos**
>
> Yes, we have corrected them all in the revised draft.
>
> W3. **Discussion on online unlearning** (Introduction, second paragraph, and Conclusion of the revised draft)
>
> Yes, we have added a discussion on how our batch Gaussian certifiability framework can serve as a foundation for future online unlearning in high dimensions with sequential or continuous deletion requests.
>
> W4 **Table for comparison** (Introduction of the revised draft)
>
> Yes, we have added a table in the introduction on how our contribution fits in comparison to previous works.

---

### Official Review · Reviewer_xe5g · 2025-11-03

**Soundness:** 3
**Presentation:** 2
**Contribution:** 2
**Rating:** 4
**Confidence:** 3

**Summary:**

This paper tackles the problem of certified machine unlearning in high-dimensional settings where the number of parameters p is comparable to the sample size n. It introduces a new privacy notion called (phi, epsilon)-Gaussian certifiability, which is argued to be the canonical and optimal framework for high dimensions. The main theoretical result shows that a single step of Newton's method, followed by calibrated Gaussian noise, is sufficient to achieve both this strong privacy guarantee and maintain model accuracy. This finding directly contrasts with the prior state-of-the-art analysis by Zou et al. (2025), which concluded that at least two Newton steps were necessary.

**Strengths:**

The paper's primary strength is its novel and theoretically sound analysis of unlearning in the challenging high-dimensional proportional regime (p ~ n). The introduction of varepsilon-Gaussian certifiability is well-motivated, leveraging a hypothesis testing perspective and properties of high-dimensional data to create a more natural and tighter privacy notion than previous frameworks like varepsilon-certifiability. The core result—that only one noisy Newton step is needed—is both surprising and significant, as it suggests prior analyses were suboptimal due to an incompatible privacy definition, not a fundamental algorithmic limitation. The theoretical claims are strongly supported by comprehensive experiments showing the clear superiority of the Gaussian mechanism over the Laplace mechanism from Zou et al. (2025).

**Weaknesses:**

The analysis relies on specific technical assumptions, such as the data features being sub-Gaussian and the loss functions being convex. While common in high-dimensional statistics, the practical implications for highly non-convex models (like large neural networks) remain an open and important question. Furthermore, the requirement for Hessian computation in the Newton step, though theoretically elegant, could be a scalability bottleneck for extremely large-scale models. The paper could also benefit from a more detailed discussion on the potential trade-offs or limitations of the Gaussian certifiability notion itself, beyond its superiority over prior work.

**Questions:**

n/a

---

> ### Author Response · Authors · 2025-11-21
> **Thanks for engaging with our work. We have addressed your points in detail below, and in the newly added texts in the revised draft. We hope that, upon reconsideration, you may find the paper to constitute a substantive contribution to the literature on high-dimensional certified machine unlearning.**
>
> W1. On sub-Gaussian features. (see Conclusion, and section D of the appendix of the revised draft (RD))
>
> We agree that assuming sub-Gaussian (SG) features is standard in high-dimensional statistics. However, (1) In practice, features are stored with finite precision, so entries lie in bounded ranges and are often well-modeled as SG. (2) Some control over the tails of the feature is essential, while one can relax SG to slightly more general Orlicz spaces  (sub-Weibull in Kuchibhotla and Chakrabortty (2022)), without such assumptions, no efficient, approximate unlearning method can guarantee both accuracy and privacy for arbitrary data-point removals. For example, consider the Newton update: with heavy tails, a single outlier can dominate the training loss. Removing it causes a large parameter shift, so a single Newton step (or any cheap update) cannot be accurate, and full retraining may be needed. Under subGaussianity, our work is (to our knowledge) the first to show that a single Newton step suffices to achieve both Gaussian certified unlearning and vanishing generalisation error in high dimensions.
>
> W2. On convex loss functions. (Conclusion, Appendix D of RD)
>
> We agree that extending our novel Gaussian certifiability framework to highly non-convex models (large neural networks) is an important direction. In this paper, we focus on convex ERM because it offers a clean setting to rigorously track the effect of a Newton update. Even in this convex high-dimensional regime, our result is, to our knowledge, the first to show that a single noisy Newton step simultaneously yields Gaussian certified unlearning and accuracy, under weaker conditions on the per-example loss than prior works (Sekhari et al., Guo et al., and Allouah et al.) whose analysis fails in the proportional high-dimensional setting. Extending these beyond convex models while retaining a comparable level of sharpness is a challenging but compelling next step.
>
> W3, On Hessian computation and scalability (Conclusion of RD)
>
> We agree that full Hessian computation can be expensive in very high dimensions. This motivated us to examine how few Newton steps are needed, and we found that even in high-dimensional settings, a single iteration is sufficient. To test the scalability of our approach, we increased the dimensions of the experiments to p =10000 in the revised version. We are running even larger dimensional experiments currently and plan to add them to the appendix of the revised draft. For extremely large-scale models, several works (Koh-Liang 2017, Park et al.) show that related quantities, such as influence functions, formally require the inverse Hessian, which can be computed efficiently via approximations whose accuracy in high dimensions is still not well understood. Investigating these approximations is an important direction for future work, and we hope that our framework will shed further light on such methods.
>
> W4. Potential limitations of the Gaussian certifiability. (Appendix B.2 of RD)
>
> We thank the reviewer for raising this point. We would like to clarify that apart from its superiority over all other previously proposed frameworks (in high dimensions), we already discussed the scope and limitations of Gaussian certifiability in the main text and in sufficient detail in appendices C.2, G.1. This includes a discussion of a convex duality perspective of our framework which is also extremely relevant in practice (appendix C.2), whose underlying basis is an intrinsically high-dimensional fact– Gaussian universality of distributions of low dimensional projections of high dimensional symmetric log-concave distributions (appendix C.2), dimension-freeness (Lemma 6 in section G.1).
>
> To make this even clearer, we have also added the following clarification in Appendix B.2 of RD (with much more elaboration), establishing the superiority of our Gaussian certifiability framework and the flexibility of our more general $f$-certifiability framework. Gaussian certifiability serves as the canonical and tightest possible notion of certifiability in high dimensions, as long as the coordinates $\hat{\beta}(i)$ of the released statistic $\hat{\beta}$ $\in$ $\mathbb{R}^p$ are generically continuous, instead of being (structured or discrete) constrained such as binary $\hat{\beta}(i) \in \{0,1\}$.
>
> Even in such highly non-Gaussian and discrete high-dimensional situations, our flexible and more general $f$ certifiability framework (see Definition 2) is applicable, and we believe that our framework is flexible enough to dictate what different baseline $f$ to choose from depending on the class of problems at hand. Examples of such baseline $f$ includes  $f=T(Po(1), Po(\varepsilon))$, where $Po(\varepsilon)$ is the Poisson distribution with mean $\varepsilon$. A particularly interesting direction would be to apply the Poisson framework for sparse networks, including problems of graph unlearning with approximately Poisson degree distributions (Chen et al).

---

> ### Author Response · Authors · 2025-12-03
> **Summary of Responses and Revisions for Reviewer xe5g**
>
> We thank the reviewer for engaging with our work. Now we summarize the main revisions made during the rebuttal period, including detailed explanations added in the revised draft to address the reviewer's comments. In summary, we have   **carefully and thoroughly addressed all the suggestions made by the reviewer**.
>
> Responses to reviewer's questions (N/A)
>
> Responses to identified weaknesses
>
> W1. **On sub-Gaussian features** (see Conclusion, and section D of the appendix of the revised draft)
>
> Yes, our theoretical bounds as stated hold for sub-Gaussian features. However, in practice, features are stored with finite precision, so entries lie in bounded ranges and are often well-modeled as sub-Gaussian. Moreover, some control over the tails of the feature is essential, and heavy-tailed high-dimensional features can significantly hurt any efficient, approximate unlearning method.
>
> W2. **On convex loss functions**. (see Conclusion and appendix D of the revised draft)
>
> Yes, extending our novel Gaussian certifiability framework to highly non-convex models (large neural networks) is an important direction. However,  in this convex high-dimensional regime, our result is, to our knowledge, the first to show that a single noisy Newton step simultaneously yields unlearning and accuracy.
>
>
> W3. **On the Hessian computation and scalability of Newton method** (section 6, Conclusion of revised draft)
>
> Yes, we added a discussion on how our Newton-based unlearning method provides a basis for scalable approximate Newton methods. Moreover, we increased the dimensions of the experiments to p =10000 in the revised version.
>
> W4.**Potential limitations of the Gaussian certifiability**. (appendix B.2 of revised draft)
>
> Yes, we added a discussion that there are no limitations of the Gaussian certifiability framework in high dimensions, except that one should apply the more general and flexible $f$-certifiability framework, depending on the class of problems at hand. Gaussian certifiability serves as the canonical and tightest possible notion of certifiability in high dimensions, as long as the coordinates $\hat{\beta}(i)$ of the released statistic $\hat{\beta}$ $\in$ $\mathbb{R}^p$ are generically continuous, instead of being (structured or discrete) constrained such as binary $\hat{\beta}(i) \in$ {0,1}.

---

### Official Review · Reviewer_3wiW · 2025-11-06

**Soundness:** 3
**Presentation:** 4
**Contribution:** 4
**Rating:** 8
**Confidence:** 2

**Summary:**

This paper introduces **$(\phi, \epsilon)$-Gaussian certifiability**, a new privacy-certification framework for high-dimensional settings ($p \approx n$). It shows that a single Newton step with Gaussian noise can achieve certified unlearning while maintaining model accuracy. The work improves upon prior high-dimensional unlearning methods by reducing the number of required Newton steps and provides empirical validation on synthetic linear models, demonstrating the advantages of Gaussian noise for generalization and unlearning metrics.

**Strengths:**

-**Problem Significance:** This is a timely work in the important field of machine unlearning, addressing key limitations of existing methods. It does so by introducing a canonical privacy notion tailored to high-dimensional settings, bridging differential-privacy hypothesis testing and machine unlearning.

-**Novelty and Theoretical Elegance:** The paper introduces $(\phi, \epsilon)$-Gaussian certifiability and proves that a single Newton step suffices for certified unlearning. This provides a rigorous, theoretically sound approach that balances privacy guarantees and model accuracy.

-**Support for Multiple Deletions:** The method can handle multiple deletion requests simultaneously, as long as the total deletion size remains within the theoretical bound. This feature is valuable for practical batch-deletion scenarios.

**Weaknesses:**

-**On deletion size**. From what I understand, the certified unlearning guarantees hold only for relatively small deletion sets ($m = o(n^{1/4-\alpha})$), which limits applicability when larger batches of records need to be removed. This is a clear theoretical limitation of the method.

-**On noise-scale calibration**. While the paper provides a theoretical formula for Gaussian noise, it involves unspecified constants and poly-logarithmic factors, making practical calibration challenging. Implementing the noise correctly in real-world scenarios may be difficult.

-**On model scope and empirical validation**. Experiments are restricted to linear ridge-regularized ERM models with Gaussian/sub-Gaussian features on synthetic datasets. Non-convex models, deeper architectures, and real-world datasets are not evaluated. Even though this is outside the scope and assumptions of the paper, I think it would be interesting to see a simple extension, such as a two-layer fully connected network on MNIST, with results compared against the frequently cited papers by Sekhari et al. and Guo et al.

-**On scalability of Newton method**. The method requires inversion of the Hessian ($O(p^3)$), which may be difficult to compute in high-dimensional settings, as often encountered in deep neural networks. This can be computationally and memory intensive, and may also be numerically unstable.

**Questions:**

I would appreciate it if you could also answer the following questions:

Q1. **On the high-dimensional regime**. From what I understand, your theoretical framework is developed under the assumption that the number of features $p$ and the number of samples $n$ grow proportionally ($p \sim n$), which allows the use of Random Matrix Theory results. While the framework is elegant in this setting, it is unclear how well it could extend to ultra high-dimensional regimes where $p \gg n$, such as in modern neural networks. Do you think the current framework could provide a solid base for such settings, or would substantial modifications be needed?

Q2. **On the deletion set structure**. Does you framework for more sophisticated/structured unlearning scenarios such as class or concept unlearning?

Q3. **On the choice of $\phi$ (tolerance):** How sensitive are the certified unlearning bounds to the tolerance parameter $\phi_n$? Is there a principled way to choose $\phi_n$ in practice, and how might different choices affect empirical performance or computational stability?

---

> ### Author Response · Authors · 2025-11-21
> **Thank you for your positive and careful evaluation. We have addressed your questions, comments in the revised draft, as detailed in the points below. We request you to take a look at the newly added texts in the revised draft for more details.**
>
> Q1. On the high-dimensional regime. (see Appendix B.2 of the revised draft (RD))
>
> Our $(\phi, \varepsilon)$ Gaussian certifiability framework (GCF) applies in the  $p \gg n$  regime. We believe that GCF provides a solid foundation for any high-dimensional unlearning setting, including neural networks. However, since we showed the superiority of GCF with a noisy Newton-based algorithm, where RMT results fit in well with $p\sim n$, we have used it to simplify the expressions that appear in Theorems 2 and 3.  Validating the GCF with other un-learning algorithms in $p\gg n$ regimes is an interesting future direction.
>
> Q2. On the deletion set structure. (Appendix B.2 of RD)
>
> Our more general $f$-certifiability framework (fCF) applies to sophisticated scenarios, including class or concept unlearning. The corresponding algorithmic question would be an interesting future direction.
>
> Q3. On the choice of  $\phi$ (tolerance). (Appendix C.1 of RD)
>
> In our current framework (Theorem 2), the failure probability $\phi$ explicitly depends only on $n,p$ and the tail of $y$. So, the only practical requirement is that $\phi$ has to be positive and is completely up to the user specification, and it does not affect empirical performance above a minimal threshold $\phi_n$ (Theorem 2 for the value of $\phi_n$). Moreover, since in practice the tail behaviour of the response $y$ is usually good and the dominating term in the threshold $\phi_n$ is $8n^{-3}$ (see Theorem 2), it is effectively negligible in practice. In classification tasks, the response $y$ is bounded, so its tail probability $q_n^{(y)}=0$, and the critical failure probability $\phi_n$ for data size $n\sim 10^5$ is approximately $8\times 10^{-15}$.
>
> W1. On deletion size. (Conclusion of RD)
>
> We agree that the restriction to small deletion sets (still covering a large range from $m=O(1)$ to $m^4=o(n)$) is a limitation of our proof technique. However, our experiments (See Figure 4) indicate that for a single Newton step, the regime $m^3 = o(n)$ is essentially the largest one in which one can simultaneously maintain $\varepsilon$-Gaussian certifiability and ensure that GED $\to 0$. In this sense, our theoretical assumption on $m$ is very close to the empirically best-possible regime for one-step Gaussian certified unlearning.
>
> W2. On noise-scale calibration. (bullet point 6 in section F.3 of RD)
>
> Please see bullet point 6 in section F.3 of the appendix (RD), where we give two calibration rules for choosing the noise level $\sigma$ in practice without actually computing the constants $C_1(n), C_2(n)$. Our main theorems are stated with unspecified constants and poly-logarithmic factors to simultaneously cover a broad class of losses $\ell$ and regularizers $r$ that practitioners may wish to use. Once $\ell$ and $r$ are fixed and standard concentration estimates for the relevant data parameters (e.g., mean, covariance) are computed, these constants become explicit; Appendix F.3 of the revised draft provides the exact expressions and works out two examples: linear and logistic regression with ridge penalty. We agree that our constants are not optimal up to $(1+o(1))$ factors, obtaining sharp constants in high dimensions usually requires very delicate asymptotic analysis, whereas our goal here is to provide non-asymptotic guarantees for fixed but large $(n,p)$ that are directly useful in practice, while still covering a substantially wider range of losses and regularizers than prior work (Guo et al., Sekhari et al., Allouah et al.).
>
> W3. On model scope and empirical validation.
>
> We agree and have added high-dimensional experiments in RD. Figure 4 now reports our unlearning metrics for $n = p/10$ with $p$ from 1000 to 10000, using the same data-generation and evaluation as in the low-dimensional setup, and again showing superior performance of Gaussian unlearning.
>
> Our primary goal, however, is to develop a rigorous high-dimensional theory of Gaussian certified unlearning (with relaxed per-example loss assumptions), which is absent in the current literature. Accordingly, we focus on linear ridge-regularized ERM with Gaussian features, exactly matching the assumptions of our RMT analysis. We agree that extending to non-convex models and real-world datasets is an interesting future direction.
>
> W4. On the scalability of the Newton method. (Conclusion of RD)
>
> We agree that full Hessian computation is expensive in very high dimensions. This motivated us to examine how few Newton steps are needed, and we found that even in high-dimensional settings, a single iteration is sufficient. For extremely large-scale models, prior work (Koh–Liang, 2017; Park et al.) shows that related quantities such as influence functions, which formally require the inverse Hessian, can be computed efficiently via approximations whose accuracy in high dimensions is still not well understood. Investigating these approximations within our theoretical framework is an important direction for future work.

---

> ### Author Response · Authors · 2025-12-03
> **Summary of Responses and Revisions for Reviewer 3wiW**
>
> We thank the reviewer for their careful and constructive evaluation. Now we summarize the main revisions made during the rebuttal period, including detailed explanations added in the revised draft to address the reviewer's suggestions. In summary, we have   **carefully and thoroughly addressed all the suggestions made by the reviewer**.
>
> **Major points**
>
> 1) **We added real data experiments on MNIST and IMDb datasets** in section 5 and appendix I of the revised draft and showed the superiority of Gaussian unlearning over previous unlearning mechanisms in high-dimensional datasets arising in practice.
>
> 2) **We resolved the practical concern of noise calibration** and provided details of two practical rules to choose the noise levels in bullet point 6 in section F.3 of the appendix of the revised draft.
>
> 3) **We added discussions on potential limitations and extension of our proposed certifiability framework** in section 6, Conclusion of revised draft.
>
> Responses to reviewer's questions
>
> Q1. **On the high-dimensional regime** (appendix B.2 of the revised draft)
>
> Yes, we believe our Gaussian certifiability framework provides a solid foundation for any high-dimensional unlearning setting, including neural networks in ultra-high-dimensional regimes.
>
> Q2. **On the deletion set structure**  (appendix B.2 of the revised draft)
>
> Yes, our more general $f$ certifiability framework applies to sophisticated scenarios, including class or concept unlearning.
>
> Q3. **On the choice of $\varphi$ (tolerance)** (appendix C.1 of the revised draft)
>
> Yes, in practice, the only practical requirement is that  $\varphi$ has to be positive and is completely up to the user specification, and it does not affect empirical performance.
>
> Responses to identified Weaknesses
>
> W1. **On deletion size** (section 6 Conclusion of revised draft)
>
> Yes, our theoretical bounds hold up to $m^4 =o(n) $, whereas in the experiments, we find $m^3=o(n)$ is the best possible.
>
> W2. **On noise-scale calibration** (bullet point 6 in section F.3 of the appendix of the revised draft)
>
> Yes, we have addressed the problem of choosing the noise level in practice by providing details for two calibration rules.
>
> W3. **On model scope and empirical validation** (section 5 and appendix I of the revised draft)
>
> Yes, we added experiments with real datasets such as IMDb and MNIST, and showed the superiority of Gaussian unlearning over previous unlearning mechanisms in high-dimensional datasets arising in practice.
>
> W4. **On the scalability of the Newton method** (section 6, Conclusion of revised draft)
>
> Yes, we added a discussion on how our Newton-based unlearning method provides a basis for scalable approximate Newton methods.

---

### Author Response · Authors · 2025-12-03
**Overall response -- Thanking the Area Chairs for their time and consideration and the Reviewers for their constructive feedback.**

We sincerely thank the Area Chairs for their time and effort in evaluating our submission.

We also thank all reviewers for their thoughtful feedback and for recognizing the **motivation**, **timeliness**, **significance**, **novelty and theoretical elegance**, **soundness**, **canonically high-dimensional and fundamental nature** of our study of Gaussian certifiability bridging **hypothesis testing, differential privacy, and machine unlearning** in the **challenging high-dimensional proportional regime** under **relaxed assumptions on the loss**.

Moreover, we appreciate the opportunity to address all the reviewers' concerns and improve our manuscript through this process. Below we summarize the major revisions made during the discussion period, together with clarifications included in the updated draft.

**Major points raised by reviewers during the discussion have been addressed in the revised draft and in comments below**

1) **We added real data experiments on MNIST and IMDb datasets as well as experiments with larger dimension p =10K** in section 5 and appendix I of the revised draft, and showed the superiority of Gaussian unlearning over previous unlearning mechanisms in high-dimensional datasets arising in practice.

2) **We resolved the practical concern of noise calibration and computing the constants $C_1(n), C_2(n)$**. In bullet point 6 in section F.3 of the appendix of the revised draft, we have provided sufficient details of two practical rules to choose the noise level $\sigma$ without requiring the computation of $C_1(n), C_2(n)$, and gave exact expressions of these constants along with working out two examples: linear and logistic regression.

**Discussion on potential limitations, extensions and clarifications of our Gaussian Certifiability framework (GCF)**

1)  In section 6, Conclusion, and appendix B.2 of the revised draft, we have discussed

-- the **applicability of GCF for ultra-high-dimensional modern neural networks**, and how our

-- **Newton-based unlearning method provides a basis for scalable approximate Newton methods**.

2) In appendix B.2 of the revised draft, we have discussed

--**on the deletion set structure** and how our

-- **more general $f$ certifiability framework applies to sophisticated scenarios, including class or concept unlearning**.

3) In appendix C.1 of the revised draft, we have discussed

--**on the choice of $\varphi$ (tolerance)** and the fact that the only practical requirement is that

-- **$\varphi$ has to be positive and is completely up to the user specification, and it does not affect empirical performance**.

4)  In section 6, Conclusion of revised draft, we have discussed that our theoretical bounds in the challenging high-dimensional regime

--hold for **deletion size m** up to $m^4 =o(n) $, whereas in the experiments, we find $m^3=o(n)$ is the best possible.

5) In section 6, Conclusion, and section D of the appendix of the revised draft, we have discussed

-- why our theoretical bounds are naturally stated for **sub-Gaussian features**, since some control over the tails of the feature is essential.

4)  In section 6, Conclusion and appendix D of the revised draft, we have discussed

--**on convex loss functions** how non-convex models are an important direction, although in this convex high-dimensional regime,

-- our result is, to our knowledge, the **first** to show that a **single noisy Newton step simultaneously yields unlearning and accuracy**.

5) In the introduction, second paragraph, and Conclusion of the revised draft, we have discussed

-- how our batch Gaussian certifiability framework can serve as a foundation for future **online unlearning** in high dimensions with sequential or continuous deletion requests.

6)  As discussed in section F.3 of the appendix of the revised draft, that

--  $(\phi, \varepsilon)$ GPAR (Thm 2) holds for any user-chosen $\sigma$ above the critical threshold  $\sigma=$ $\frac{R}{\varepsilon}$, with R as in Theorem 2, and

--small deviations do not break certifiability guarantees and therefore  **ε-GPAR bound is not sensitive to $\sigma$**.

7) Finally, we have added a **table for comparison with previous works**, addressed all the **writing issues**, and corrected all the **typos** in the revised draft.

 **Summary**

 We have carefully and thoroughly addressed **all** reviewer concerns and incorporated the corresponding revisions into the updated manuscript. We hope that the **Gaussian certifiability framework (GCF)** proposed in this paper will serve as **the standard certifiability guarantee** for high-dimensional unlearning going forward, both because of its **strong theoretical soundness** and because of its **practical implementability**.

---

### Meta-Review · Area_Chair_aCcq · 2026-01-07

**Summary:**

This paper proposes a new theoretical framework called Gaussian certifiability for machine unlearning in high-dimensional settings.

The reviewers appreciate the following strengths of the paper:

- S1. The paper studies an important and timely problem in machine unlearning.

- S2. The paper proposes a novel theoretical framework for machine unlearning based on Gaussian Differential Privacy, bridging DP-inspired hypothesis testing with certified machine unlearning. This notion of Gaussian-certifiable machine unlearning yields a new result showing that a single noisy Newton step can achieve both privacy and statistical accuracy under relaxed assumptions.

- S3. The paper provides clear and detailed explanations contrasting the proposed framework with previous certified machine unlearning approaches.

The authors have successfully addressed most of the reviewers’ concerns during the rebuttal. In addition, the authors have provided an updated version of the paper that incorporates new experimental results, further discussions and clarifications, and addresses writing issues and typos. These changes make the required revisions straightforward to incorporate into the camera-ready version. The ratings were mostly positive before the rebuttal. Reviewers 3wiW, qmUk, and WvFT provided positive ratings before rebuttal and are likely to raise or maintain their scores. Reviewer xe5g is likely to raise their rating due to the satisfactory address of the raised concerns.

Overall, the paper has positive evaluations, with almost all raised concerns addressed, and is ready for inclusion in the camera-ready version. Thus, the paper is a strong candidate for acceptance and can be considered for an oral presentation.

**Reviewer Concerns:**

The reviewers also raised the following major concerns:

- W1. Noise calibration may introduce practical difficulties due to many unspecified constants and poly-logarithmic factors.

- W2. The simulations are limited to logistic and ridge regression models on synthetic datasets generated under controlled Gaussian features.

- W3. There is a lack of discussion on the limitations of the Gaussian-certifiability machine unlearning framework, such as the potentially small deletion set size, support for more sophisticated or structured unlearning scenarios, sensitivity to the choice of the tolerance parameter, and extensions beyond assumptions of convexity, separability, and sub-Gaussian features.

- W4. There is a lack of discussion on the limitations of the Newton-based machine unlearning algorithm under the proposed notion of Gaussian certifiability, such as high computational complexity and extensions of the theoretical results to quasi-Newton or stochastic approximations.

- W5. The paper contains several typos and writing issues.

The authors have successfully addressed all of these concerns during the rebuttal period. In particular, the authors have provided new experimental results, additional discussions and clarifications, and an updated manuscript, as detailed below.

- R1. The authors provided real-data experiments on the MNIST and IMDb datasets, as well as experiments in higher-dimensional settings, addressing W1.

- R2. The authors provided clarifications and sufficient detail on two practical rules and exact expressions for key constants, addressing W2.

- R3. The authors added further discussion on deletion set size and structure, clarified that the tolerance parameter is user-specified and theoretically irrelevant, addressing W3.

- R4. The authors clarified that the Newton-based unlearning method serves as a foundation for scalable approximate Newton methods, addressing W4.

- R5. The authors addressed all writing issues and corrected the reported typos in the revised manuscript, addressing W5.

One remaining concern not explicitly addressed is the relaxation of the current assumptions beyond convex loss functions. The authors emphasize that the current theoretical results are novel and significant, and that extending the framework to non-convex loss functions is an important direction for future work.

**Reviewer Scores:**

The ratings were mostly positive before the rebuttal. In detail, Reviewers 3wiW, qmUk, and WvFT provided positive ratings before rebuttal and are likely to raise or maintain their scores. Reviewer xe5g originally provided a negative rating is likely to raise the rating as well due to the satisfactory address of the raised concerns.

---

### Decision · Program_Chairs · 2026-01-26

Accept (Oral)